**Comparison of AOD, AAOD and column single scattering albedo from AERONET**
**retrievals and in-situ profiling measurements**
Elisabeth Andrews[1], John A. Ogren[2], Stefan Kinne[3], Bjorn Samset[4]
[1]CIRES, University of Colorado, Boulder, CO 80309, USA
[2]NOAA/ESRL/GMD, Boulder, CO 80305, USA
[3]Max Planck Institute for Meteorology, 20146 Hamburg, Germany
[4]Center for International Climate and Environmental Research – Oslo (CICERO), 0349 Oslo,
Norway
Abstract
Here we present new results comparing aerosol optical depth (AOD), aerosol absorption optical
depth (AAOD) and column single scattering albedo (SSA) obtained from in-situ vertical profile
measurements with AERONET ground-based remote sensing from two rural, continental sites in
the US. The profiles are closely matched in time (within +/-3 h) and space (within 15 km) with
the AERONET retrievals. We have used Level 1.5 inversion retrievals when there was a valid
Level 2 almucantar retrieval in order to be able to compare AAOD and column SSA below
AERONET's recommended loading constraint (AOD>0.4 at 440 nm). While there is reasonable
agreement for the AOD comparisons, the direct comparisons of in-situ-derived to AERONET-
retrieved AAOD (or SSA) reveal that AERONET retrievals yield higher aerosol absorption than
obtained from the in-situ profiles for the low aerosol optical depth conditions prevalent at the two
study sites. However, it should be noted that the majority of SSA comparisons for $AOD_{440}>0.2$
are, nonetheless, within the reported SSA uncertainty bounds. The observation that, relative to
in-situ measurements, AERONET inversions exhibit increased absorption potential at low AOD
values is generally consistent with other published AERONET/in-situ comparisons across a
range of locations, atmospheric conditions and AOD values. This systematic difference in the
comparisons suggests a bias in one or both of the methods, but we can not assess whether the
AERONET retrievals are biased towards high absorption or the in-situ measurements are
biased low. Based on the discrepancy between the AERONET and in-situ values, we conclude
that scaling modelled black carbon concentrations upwards to match AERONET retrievals of
AAOD should be approached with caution as it may lead to aerosol absorption overestimates in
regions of low AOD. Both AERONET retrievals and in-situ measurements suggest there is a
systematic relationship between SSA and aerosol amount (AOD or aerosol light scattering) –
specifically that SSA decreases at lower aerosol loading. This implies that the fairly common
assumption that AERONET SSA values retrieved at high AOD conditions can be used to obtain
AAOD at low AOD conditions may not be valid.
1. Introduction
The amount and location of absorbing aerosol in the atmosphere is critical for understanding
climate change (e.g., Hansen et al., 1997; Ramanathan and Carmichael, 2008; Bond et al.,
2013; Samset et al., 2013). Ramanathan and Carmichael (2008) note the effects of absorbing
aerosol (which they termed black carbon (BC)) on atmospheric heating rates, precipitation and
weather patterns. (Note: The terminology used to refer to absorbing aerosol is imprecise
(Petzold et al., 2013, Andreae and Gelencsér, 2006) and encompasses the terms describing
chemistry, e.g., 'black carbon' (BC) and terms describing optical effects, e.g., absorption.  The
measurements reported herein all refer to light absorption.) The vertical distribution of BC can
also influence its effect on climate (e.g., Haywood and Ramaswamy, 1998; Samset et al., 2013;
Ramanathan and Carmichael, 2008). Single scattering albedo (SSA) is an indicator of the
absorbing nature of the aerosol; higher SSA values indicate a more reflective (whiter) aerosol
while a more absorbing aerosol will have lower SSA values. SSA is a primary determinant of
whether the aerosol will have a warming or cooling effect (e.g., Haywood and Shine, 1995;
Hansen et al., 1997; Reid et al., 1998).  Uncertainty in the value of SSA due to uncertainties in
the amount of absorbing aerosol can even prevent determination of the sign of aerosol forcing
on local to regional scales. Bond et al. (2013) assessed BC as the second most important
global-average warming species (top-of-atmosphere forcing +1.1 W $m^{-2}$, 90% bounds: +0.17 to
+2.1 W $m^{-2}$) after $CO_2$ (in Bond et al. (2013) the direct effect of BC is 0.71, 90% bounds: +0.09
to 1.26 W $m^{-2}$).
Currently, the only way vertical profiles of aerosol absorption can be obtained is via airborne in-
situ measurements.   Such flights are expensive and tend to primarily occur during intensive
field campaigns, which are usually aimed at studying specific aerosol types (e.g., biomass
burning, African dust, urban/industrial pollution).  This reliance on short-term campaigns results
in profile data sets that are sporadic in both space and time, and not necessarily representative
of typical conditions.  Additional issues with airborne in-situ measurements include adjustment
of measurements to ambient conditions, particle losses in sample lines, and instrument
uncertainties.  Nonetheless, in-situ vertical profiling of absorbing aerosols has provided useful
information to modelers trying to understand climate effects, transport, and lifetimes of these
important atmospheric constituents (e.g., Koch et al., 2009; Schwarz et al., 2010; Skeie et al.,
68    2011).
The limited availability of in-situ vertical profile measurements means modelers must rely on
globally sparse and/or temporally sporadic airborne measurements to evaluate BC vertical
distributions in their models. Alternatively, the column properties retrieved from AERONET
measurements and inversions have been widely used to provide a first constraint on modeled
vertical aerosol properties (e.g., Sato et al., 2003; Koch et al., 2009; Bond et al., 2013; He et al.,
2014; Wang et al., 2014).  Use of the AERONET data as an absorption constraint has
suggested upscaling of modeled AAOD values by a factor of 2-6 depending on location (e.g.,
Bond et al., 2013), although Wang et al. (2016) has shown that better spatial resolution of
models and emission inventories can reduce some of the previously observed model/AERONET
discrepancies.
Ground-based remote sensing of both direct attenuation and sky radiances permit inversions of
atmospheric column averaged absorption. By retrieving the complex refractive indices at
different solar wavelengths as well as the average aerosol size-distribution, absorption related
properties can be determined (e.g., aerosol absorption optical depth (AAOD), single scattering
albedo (SSA) and, absorption Ångström exponent (AAE)). The AERONET network has a fairly
wide spatial coverage on land, with long data records at many sites (Holben et al., 1998;
Dubovik et al., 2000; Dubovik and King, 2000). One obvious limitation of the AERONET
inversion retrievals is that the uncertainty of the derived single scattering albedo (SSA) becomes
very large at low values of AOD (Dubovik et al., 2000).  To minimize the effects of this
uncertainty, the AERONET Level-2 data invalidates all absorption-related values if the AOD at
wavelength 440 nm ($AOD_{440}$) is below 0.4 (Dubovik et al., 2000; Dubovik et al., 2002; Holben et
al., 2006).  Unfortunately, this restriction greatly reduces the spatial and temporal coverage of
absorption-related data that can be obtained from AERONET. Moreover, by excluding low AOD
cases, the climatological statistics of AAOD derived from the AERONET Level-2 data may be
biased high.
Model analysis of global AOD values suggest that 95% of global $AOD_{440}$ values are below 0.4
(Figure 1), while 89% of the $AOD_{440}$ values over land are below the 0.4 threshold.  Five models
in the AeroCom suite (GMI-MERRA-v3, GOCART-v4, LMDZ-INCA, OsloCTM2, and
SPRINTARS-v385) have reported daily-average values of $AOD_{440}$ (for AeroCom Phase II
control experiment), which can be used to develop a cumulative frequency distribution of the
percent of the Earth's surface and days where a Level-2 AERONET retrieval of AAOD might be
possible (ignoring the presence of clouds and absence of sunlight).  Figure 1 indicates that, at
best, Level-2 AERONET AAOD retrievals might represent 5% of the days, globally, and less
than 11% of the days over land. In other words, the AOD constraint on Level-2 AERONET
almucantar inversion retrievals means these retrievals represent only a small fraction of the
Earth's surface and are biased to conditions of high aerosol loading.
The other information that Figure 1 provides is the fractional contribution of regions with different
$AOD_{440}$ amounts to the total aerosol and the fossil fuel black carbon (BCFF) radiative budget.
These values were derived from monthly data from 4 models in the AeroCom suite.  The
fractional contribution to the radiative budget can be mathematically described as follows: for
each model grid box there are three quantities: (i) the radiative forcing (W m$^{-2}$), (ii) the horizontal
area of the box (m$^2$), and (iii) the $AOD_{440}$. The product of the radiative forcing term and area is
the perturbation to Earth's radiative budget due to total aerosol (or BCFF) in the box.  The sum
of this product over all the boxes is the total perturbation.  Figure 1 shows the fraction of the
radiative budget perturbation as a function of $AOD_{440}$.  It suggests that approximately 75% of the
total aerosol forcing and 83% of BCFF forcing is due to regions of the globe where $AOD_{440}$ <0.4.
This highlights the significant contribution of aerosol in these cleaner areas to the total global
radiation budget.
It should be noted that there is significant inter-model variation in the AeroCom cumulative
$AOD_{440}$ and radiative forcing plots shown in Figure 1. In particular the BCFF cumulative forcing
fraction varies with the lifetime of BC predicted by the models. A long BC lifetime results in more
dilute AOD and BCFF radiative forcing distributions. Other issues include the fact that global
models have limited spatial and temporal resolution, and generally simulate less variability in
aerosol properties than is observed in measurements.  However, all models used to generate
Figure 1 follow the same general trend as is shown in Figure 1 with the take-away point being
that $AOD_{440}$ values >0.4 are a relatively rare occurrence.

Because of the potential of the AERONET absorption-related retrievals (e.g., AAOD and SSA)
for understanding global distributions of absorbing aerosol, there have been many studies
comparing AERONET retrieval values with those obtained from in-situ measurements in order
to assess the AERONET retrieval validity. Such comparisons have taken several different
forms. There have been direct comparisons where column SSA or AAOD values calculated
from individual in-situ vertical profiles have been compared with AERONET retrieved values for
retrievals close in time and space (Haywood et al., 2003; Magi et al., 2005; Mallet et al., 2005;
Leahy et al., 2007; Corrigan et al., 2008; Osborne et al., 2008; Johnson et al., 2009; Esteve et
al., 2012; Schafer et al., 2014). In addition to direct comparisons there have been general,
statistical assessments between AERONET and  in-situ measurements for both SSA and AAOD
including: (a) comparing surface in-situ measurements with AERONET retrievals (e.g., Dubovik
et al., 2002; Doran et al., 2007; Mallet et al., 2008; Corr et al., 2009); (b) comparing in-situ SSA
(or AAOD) from a few flight segments to the corresponding column SSA (or AAOD) from
AERONET (e.g., Kelektsoglou et al., 2012; Müller et al., 2012) and (c) comparison of statistical
distributions or averages of AERONET retrievals for a given time period with airborne in-situ
measurements (e.g., Ramanathan et al., 2001; Leahy et al., 2007; Andrews et al., 2011a;
Ferrero et al., 2011; Johnson et al., 2011). Many of these statistical comparisons have shown
good agreement between the AERONET and in-situ values. This increases general confidence
in the AERONET retrievals.   However, such statistical comparisons are not appropriate for the
evaluation of the accuracy of individual retrievals.

The primary scientific question to be addressed in this paper is: *Is there a consistent bias*
*observed between AAOD and column SSA obtained from in-situ profiling flights and AERONET*
*retrievals?* The answer to this question may help determine the validity of adjusting model
estimates of AAOD to agree with AERONET retrievals (e.g., Sato et al., 2003; Bond et al.,
2013).  It should be noted that AERONET does not recommend the use of absorption-related
parameters (e.g., single scattering albedo, absorption aerosol optical depth, and complex index
of refraction) at $AOD_{440}$ below 0.4. Dubovik et al. (2000) suggests the uncertainty of AERONET
SSA values more than doubles for $AOD_{440}$ less than 0.2.
In what follows, we first evaluate how direct AERONET AAOD retrievals compare with those
derived from multi-year, in-situ measurements obtained from vertical profiles over two rural
continental AERONET sites in the U.S.  Second, we create a summary of all direct AAOD or
SSA comparisons between in-situ vs. AERONET data previously presented in the literature in
order to place our results about AERONET aerosol absorption-related retrievals in a wider
context. Finally, we look at the seasonality of in-situ, AERONET, and modelled (AeroCom) SSA
and AAOD values to see if the annual cycles can provide any insight into observed
discrepancies in the direct comparisons.  Because this study focuses on only two low AOD sites
in the continental US which are unlikely to be generally representative of other low loading sites
around the globe, and because other factors (e.g., Wang et al., 2016) may contribute to
reported differences between modelled and AERONET AAOD we do not attempt to suggest
implications for global BC forcing.

## 2. Methods

This study utilizes data from two sites with collocated AERONET measurements and multi-year, in-situ aerosol profiling measurements. The two sites are Bondville (BND, 40.05ºN 88.37ºW, 230 m asl) and Southern Great Plains (SGP, 36.61ºN 97.49ºW, 315 m asl). Surface in-situ measurements and AERONET column measurements have been made at both locations since the mid-1990s (e.g., Delene and Ogren, 2002; Sheridan et al., 2001: Holben et al., 1998). Weekly to twice-weekly flights measuring in-situ vertical profiles of aerosol optical properties over these two sites were made for a subset of the years of ground-based observations. At SGP the in-situ profile flights were centered over the site's central facility where the AERONET sunphotometer is deployed. Due to FAA flight restrictions, the BND in-situ profiling flights took place approximately 15 km to the WNW of the AERONET sunphotometer location at the BND surface site (Sheridan et al., 2012). Additionally, for BND, a low level flight leg (200 m agl) was flown directly over the instrumented BND surface site. The flights at both sites were subject to 'visual flight regulations' which means they took place during daylight hours and the plane did not fly in-cloud.

At BND and SGP, the median $AOD_{440}$ values are 0.14 and 0.11, respectively (based on all AERONET Level-2 data from the start of AERONET measurements at each site). These median values fall right around the 50% mark on the AOD cumulative distribution plot (Figure 1), indicating BND and SGP may be appropriate sites to explore potential discrepancies between AERONET and in-situ AAOD and SSA retrievals at lower AOD conditions.

## 2.1 IN-SITU

The in-situ aerosol profiles were obtained with dedicated Cessna 206 airplanes flying stair-step profiles one to two times per week over the two sites. Between 2006 and 2009, 365 flights were flown over BND (out of a total of 401 flown in the region (Sheridan et al., 2012)), while 171 aerosol profile flights were flown over SGP in the 2005-2007 time period (Andrews et al., 2011a). The profiles consisted of 10 (at BND) or 12 (at SGP) level flight legs between approximately 450 and 4600 m asl (corresponding to approximately 150 and 4200 m agl). The profiles, which were 'stair-step' descents, took approximately 2 hours to complete as the airplane spent set amounts of time at each level (10 min/flight level for flight legs above ~1600 m asl and 5 min/flight level for flight legs below that altitude) in order to improve measurement statistics at the typically cleaner higher altitude flight levels. Airplane speed was approximately 50 m/s, resulting in the 10 min upper level legs being approximately 30 km long and the 5 min lower level legs approximately half that (15 km) length. This flight pattern means the last 30 min of the profile were typically in the boundary layer for these two sites and encompassed the majority of the aerosol contribution to column aerosol loading. Previous work has shown that the airplane measurements appear to capture the variability in aerosol properties observed by the long-term, continuous measurements at the surface (e.g., Figure 3 in Andrews et al., 2004)

Descriptions of the flight profiles and aircraft package have been described in detail in other papers (Andrews et al., 2011a; Sheridan et al., 2012) so only a brief description is provided

here. The pilot flew within the constraints provided (specifically-defined stairstep profile, vary the
time of day, cross wind, over the instrumented field site, during daylight and not within clouds)
but without day-to-day scheduling input from scientists. Here, we utilize the same 10 flight levels
for both profiling sites: 457, 609, 915, 1219, 1829, 2439, 3050, 3659 and 4575 m asl.  Of the
365 flights at BND, 253 flights had complete profiles (all flight levels) with valid scattering,
absorption and relative humidity data; at SGP, 132 flights out of 171 were complete.  Only
complete profiles (all 10 flight levels) were used in this analysis. As is obvious from the vertical
range of the flight levels, complete in-situ profiles do not equate to complete atmospheric
profiles – this is discussed more in the in-situ uncertainties discussion (Section 2.4.1).  The
number of flights that could be compared with AERONET measurements is significantly less
than this, as discussed in Section 2.3 where the merging of the AERONET and in-situ data sets
is described.

The aircraft were equipped with an inlet that sampled particles with aerodynamic diameter $D_p<7$
$\mu$m, and losses in downstream sample lines were estimated to reduce the particle diameter for
50% sampling efficiency to 5 $\mu$m (Sheridan et al., 2012).  Aerosol light absorption ($\sigma_{ap}$) was
measured at three wavelengths (467, 530, 660 nm) using a Radiance Research Particle-Soot
Absorption Photometer (PSAP) and aerosol light scattering ($\sigma_{sp}$) was measured at three similar
wavelengths (450, 550, 700 nm) using an integrating nephelometer (TSI model 3563). The
measurements of absorption and scattering were made at low relative humidity (RH<40%).
Absorption data were corrected for scattering artifacts, flow and spot size calibrations, etc.,
using the Bond et al. (1999) algorithm, with appropriate modifications for wavelength (Ogren,
2010).  The Anderson and Ogren (1998) correction for instrument non-idealities was applied to
the nephelometer data.

Ambient temperature ($T_{amb}$)  and RH ($RH_{amb}$) were measured by a sensor (Vaisala Inc, Model
Humicap 50Y) mounted on the aircraft fuselage inside a counterflow inlet shroud, and the
nephelometer sample pressure was used as a surrogate for ambient pressure.  These
measurements of ambient meteorological parameters were used to adjust the in-situ optical
data to ambient conditions in order to compare with the AERONET measurements and
retrievals, which are made at ambient conditions.  Climatological IMPROVE network surface
aerosol chemistry measurements of sulfate and organic carbon (Malm et al., 1994) were utilized
to determine a value for the hygroscopic growth parameter '$\gamma$' for each site based on the Quinn
et al. (2005) parameterization which relates aerosol hygroscopicity to organic mass fraction.
For BND $\gamma=0.71\pm0.08$, while for SGP $\gamma=0.65\pm0.08$.  At BND the IMPROVE chemistry
measurements are co-located at the profile location, while for SGP the measurements at the
IMPROVE Cherokee Nation site (approximately 56 km southwest of the profile location) were
used.  This $\gamma$ value was then used in conjunction with the airborne $RH_{amb}$ measurements to
adjust the in-situ scattering profiles for both SGP and BND.

The equation used to adjust the dry, in-situ scattering to ambient relative humidity ($RH_{amb}$) is a
commonly used aerosol hygroscopic growth parameterization (e.g., Kasten, 1969; Hanel, 1976;
Kotchenruther et al., 1999; Carrico et al., 2003; Crumreyrolle et al., 2014):

$$\sigma_{sp}(RH_{amb})/\sigma_{sp}(RH_{dry})=a*(1-(RH_{amb}/100))^{-\gamma}. \qquad (1)$$

where $\sigma_{sp}(RH_{amb})$ is the aerosol scattering at ambient RH, $\sigma_{sp}(RH_{dry})$ is the measured scattering
at low RH, and $\gamma$ is the hygroscopic growth parameter derived from the IMPROVE aerosol
chemistry. The value of 'a' can be determined using: a = $(1/(1-RH_{dry}/100))^{-\gamma}$ (e.g., Crumreyrolle
et al., 2014; Quinn et al., 2005).  Here we assume a=0.9 based on the typical RH values
measured inside the nephelometer for both profile locations (BND $RH_{dry}$=12+/-11%; SGP
$RH_{dry}$=14+/-10%). $RH_{amb}$ at BND and SGP averaged 47.4% and 38.6%, respectively, over all
flight levels and seasons (56% (BND) and 43% (SGP) below 1500 m asl). The 95[th] percentile
$RH_{amb}$ values (calculated over all flights and flight levels) were 79.3% and 76.6% at BND and
SGP, respectively. (Note: scattering-weighted column average RH values were 54% at BND
and 43% at SGP).  Applying eq. 1 to the observed $RH_{amb}$ and $\sigma_{sp}(RH_{dry})$ profiles, the average
enhancement of column-average $\sigma_{sp}$ due to hygroscopic growth was 1.52 and 1.36 at BND and
SGP, respectively.  The corresponding 95[th] percentiles of column average enhancement of
scattering were 2.06 and 2.10.  While Equation 1 takes into account differences in hygroscopic
growth due to RH for each segment of each flight, it does not account for compositional
changes that might affect the scattering enhancement due to hygroscopicity.   For aerosol
events such as biomass burning and dust episodes with significantly different composition than
the 'normal' aerosol we would expect to over-predict the aerosol hygroscopicity relative to the
normal aerosol.  Sheridan et al., (2001) showed that the SGP surface aerosol had lower
hygroscopicity when it was influenced by dust or smoke.
The absorption measurements were adjusted to ambient temperature and pressure, but not to
ambient RH because the parameterization of the correction and its magnitude are unknown.  It
is typically assumed that absorbing aerosol is hydrophobic (e.g., Schmid et al., 2003; Reid et al.,
2005; Schaefer et al., 2014), i.e., does not take up water.  The uncertainties associated with this
assumption are discussed in section 2.4.
Both the scattering and absorption in-situ measurements were adjusted to the two nominal
Level-2 AERONET wavelengths in the mid-visible spectrum (440 nm and 675 nm). The 440 nm
wavelength is of interest as that is the wavelength for which the AOD constraint for retrieving
SSA and hence, AAOD, is given; the 675 nm wavelength is also presented because it is less
sensitive to $NO_2$, organics, and dust which could potentially bias the in-situ/AERONET
comparison. Also, evaluating data at both wavelengths helps in attributing aerosol absorption to
BC versus dust, since at 675 nm absorption is almost entirely caused by BC.  The measured
scattering Ångström exponent was used to adjust the in-situ scattering measurements to the
AERONET wavelengths. For the in-situ aerosol absorption wavelength adjustments we used a
constant absorption Ångström exponent of 1.2 to minimize the effects of noise in the
measurement.  Previous studies have shown that for both BND and SGP the absorption
Ångström exponent is ~1.0 in the BL and 1.5 at higher altitudes (Andrews et al., 2011; Sheridan
et al., 2012).  Using the incorrect absorption Ångström exponent will have a negligible effect on
the resulting absorption value because of the small difference between the measured and target
wavelengths; using an absorption Ångström exponent of 1.2 instead of 1.0 will result in a 1%
difference in adjusted wavelength while using an Ångström exponent of 1.2 instead of 1.5 will
result in a 2% difference in adjusted absorption.

Finally, using these in-situ values adjusted to AERONET wavelengths and ambient conditions
the flight profile average properties can be determined.  Aerosol extinction ($\sigma_{ep}= \sigma_{sp} + \sigma_{ap}$) was
calculated and integrated vertically for the profile to obtain the in-situ AOD.   The aerosol
absorption for each profile was integrated vertically to obtain the in-situ AAOD.  As described in
Andrews et al. (2004), the in-situ column SSA (which is compared to the AERONET SSA value
in section 3.1) was calculated for each flight level and then extinction-weighted and integrated to
determine column SSA.  This results in SSA values which are virtually identical to SSA values
calculated using: $SSA_{col,in-situ} = (AOD_{in-situ} - AAOD_{in-situ})/AOD_{in-situ}$) and effectively gives higher
weighting to the SSA values at altitudes that had the highest aerosol concentrations.  Details of
the procedure for calculating the vertical integral are given in Andrews et al. (2004), although, in
this study, the in-situ profiles contained two additional high altitude flight levels (at 3659 and
4575 m asl) and the layer at the highest altitude was assumed to extend 457 m above the
measurement altitude.  Profile statistics for various parameters including SSA are provided in
Andrews et al. (2004, 2011a) and Sheridan et al. (2012).  Individual flight profiles for various
parameters are available online at:  http://www.esrl.noaa.gov/gmd/aero/net/iap/iap_profiles.html
(for SGP) and https://www.esrl.noaa.gov/gmd/aero/net/aao/aao_prof2007.html (for BND).
2.2 AERONET
AERONET measurements have been made at BND since mid-1995 and at SGP since mid-
1994.  The AERONET network makes spectral measurements of aerosol optical depth (AOD)
using CIMEL sun/sky radiometers (Holben et al., 1998). The measurements are typically made
at seven wavelengths, with an eighth wavelength used for water vapor measurements.  The
AERONET website (http://aeronet.gsfc.nasa.gov) provides links to data from more than 500
sites across the globe.   The column extinction Ångström exponent (å) can be directly calculated
from the wavelength-dependent AOD measurements (Eck et al., 1999).  In addition to AOD and
å, algorithms have been developed utilizing both the spectral AOD and the spectral angular
distribution of the sky radiances obtained from almucantar scans, which enable retrieval of other
column aerosol properties including AAOD, SSA, size distribution, complex refractive index, and
fine mode fraction of extinction ($FMF_e$) (Dubovik and King, 2000; Dubovik et al., 2000; O'Neill et
al., 2003; Dubovik et al. 2006). The nominal wavelengths of the almucantar inversion retrievals
are 440, 675, 870 and 1020 nm.  An additional advantage of the AERONET database is that the
retrieval values are obtained consistently – the calibrations, corrections, QC and algorithms are
applied identically for each AERONET location.
For Version 2 AERONET data, there are different levels of AERONET data available for
download from the AERONET website.  Level 1.0 is unscreened data while Level-1.5
undergoes automated cloud-screening (Smirnov et al., 2000).  Level-2 represents data with pre-
field and post-field calibrations applied, manual inspection, and quality assurance (Smirnov et
al., 2000). In addition to the Level-1.5 screening, the criteria for Level-2 almucantar inversion
products include a check of the sky residual error as a function of $AOD_{440}$, solar zenith angle
must be greater than or equal 50 degrees, and almucantars must have a minimum number of
measurements in each of the four designated scattering angle bins.  Further, for Level-2
absorption-related products (including SSA, AAOD, AAE, and the complex refractive index) the
$AOD_{440}$ must be greater than 0.4 to exclude more uncertain aerosol absorption estimates
(Holben et al., 2006).  Version 3 AOD products are now available but the Version 3 inversion
products were not at the time of this writing.
The AAOD values reported in the AERONET almucantar inversion files are obtained using the
relationship:  AAOD=(1-SSA)*AOD. Schafer et al. (2014) has a nice description of how SSA is
obtained from the AERONET measurements.  In the present study, in order to maximize the
number of AERONET data points available for comparison with the in-situ measurements,
Level-1.5 retrievals of AAOD and SSA were included in the analysis if there was a
corresponding valid Level-2 AOD value but $AOD_{440}$<0.4 (i.e., the same primary criterion as was
used in Bond et al. (2013)).  We will refer to these AAOD and SSA values as 1.5* data.
2.3 Merging the IN-SITU and AERONET data sets
Merging of collocated (within 15 km), but temporally disparate data sets can induce
discrepancies in the combined data set.  Lag-autocorrelation analysis (e.g., Anderson et al.,
2003) is used to determine an appropriate time window for comparison of the AERONET and in-
situ profile measurements.  Figure 2 shows that, at the surface, at both BND and SGP,
scattering is well correlated (r(k)>0.8) out to 4-5 hr lag, while absorption is less correlated than
scattering (r(k) for absorption is 0.75 at BND and 0.55 at SGP).  Based on the correlograms,
AERONET retrievals were merged with the in-situ profile data when the retrievals were within
+/-3 h of the end of the in-situ profile. This is the same time range constraint used to compare
AERONET and PARASOL SSA values (Lacagnina et al., 2015).   Additionally, Figure 2
represents the maximum correlation that we can realistically expect to achieve in a comparison
of two different instruments with temporally offset measurements and provides context for the
AERONET/in-situ comparisons presented in Section 3.
Because the profiles are "stair-step" descents from ~4600 m asl down to ~450 m asl (e.g., see
Figure 4 in Sheridan et al., 2012), matching with AERONET retrievals at the end of the profile
means that the matches are more closely aligned with when the airplane is in the boundary
layer and thus, typically, sampling the highest aerosol concentrations. This way the maximum
time difference between the boundary layer portion of the flight and the AERONET retrieval is 3
h; if we'd chosen to match based on the start of the flight the maximum time difference between
the boundary layer measurements and the AERONET retrieval could be as large as 5 h. The
boundary layer portion (<1800 m asl) of the ~2 h profile takes approximately 30 min.  While the
+/- 3 h match window was chosen based on the surface in-situ aerosol lag-autocorrelation
statistics (Figure 2), other time windows were also examined.  For time windows less than +/-3 h
(e.g., 1 h and 2 h) the fit coefficients (slope, intercept) did not change significantly although the
AOD and AAOD correlation coefficients did improve for those smaller time windows.  For time
windows longer than +/- 3 h (e.g., 6 h and 12h) there were changes in AOD and AAOD fit
parameters and the correlation coefficients decreased significantly.  For SSA there appeared to
be no correlation between AERONET retrievals and in-situ calculated values regardless of
match window length (highest SSA correlation coefficient was 0.12, but most were less than
0.05 for both sites).  The poor correlations for SSA are not surprising given the uncertainties at
low loading.  The AERONET/in-situ comparisons for the +/-3 h window are discussed in section
3.1 below.
2.4 Uncertainties in IN-SITU and AERONET data
In any study comparing parameters obtained from different instruments and/or methods, an
understanding of the uncertainties in each of the parameters being compared is critical.  Below
we discuss the uncertainties inherent in both the in-situ and AERONET datasets.
*2.4.1 IN-SITU uncertainties*
Uncertainties for measurements by the in-situ instruments have been described previously (e.g.,
Sheridan et al., 2002; Formenti et al., 2002; Shinozuka et al., 2011; Sherman et al., 2015) so
only an overview is provided here.  Sheridan et al. (2002) calculated uncertainties in aerosol
light scattering for the TSI nephelometer to be 7-13% for 10 min legs depending on amount of
aerosol present – the higher uncertainty value applies to very low aerosol loadings (scattering <
1 $Mm^{-1}$). We assume that uncertainty in the profile scattering measurements is 13%.  13% is
appropriate for the higher altitude flight legs (10 min duration with, typically, low aerosol loading)
and is also reasonable for the lower altitude flight legs which are only 5 min in duration but have
significantly higher loading.  At both BND and SGP the median boundary layer scattering is
typically >10 $Mm^{-1}$ while median scattering for the upper altitude flight legs is typically between
1-10 $Mm^{-1}$ (Andrews et al., 2011; Sheridan et al., 2012).
Unfortunately, because profile-specific aerosol hygroscopicity measurements were not available
for the in-situ aircraft measurements described here, a single hygroscopic growth
parameterization was applied for all profiles at each site as described in Section 2.1 and
equation 1.  To determine the uncertainty in AOD induced by the uncertainty in the scattering
adjustment to ambient RH, AOD values were calculated using different $\gamma$ values representing
the range of hygroscopic growth factors suggested by the aerosol chemistry.  Specifically,
$AOD_{440}$ was calculated for $\gamma\pm1$ standard deviation and $\gamma\pm2$ standard deviations. As described
above, $\gamma$ was calculated from the climatological chemistry measurements made by the
IMPROVE network (14 years of data, ~1700 data points at BND; 10 years of data, ~1000 data
points at SGP) using the Quinn et al. (2005) parameterization.  We calculated the mean and
standard deviation of $\gamma$ based on those climatological chemistry measurements.  Using this
approach, the uncertainty in AOD due to adjustment to ambient RH was determined to be
between 9% and 16%.  This uncertainty might seem to be low, but recall that the 95[th]
percentiles of ambient RH values observed throughout the profiles were ~80% but that more
typically ambient RH in the boundary layer was less than 70% at BND and less than 60% at
SGP.  Sum of squares uncertainty analysis suggests the overall uncertainty in the in-situ AOD is
approximately 30% for higher ambient humidities ($RH_{amb}$>70%) and approximately half that at
$RH_{amb}$<50%.

Both Jeong and Li (2010) and Eck et al. (2014) have noted that the presence of nearby clouds
may influence AOD values.  They've investigated the effect of high RH-halos embedded in
aerosol layers that typically exist in the vicinity of non-precipitating cumulus clouds.  If the
AERONET retrieval went through such a halo it could result in an increased AOD due to the
combined effects of hygroscopic growth, cloud processing of aerosols and rapid gas-to-particle
conversions.  If the aircraft also flew through this RH-halo then the effect would also be
accounted for in the RH-corrected in-situ measurements.  However, if the high RH layer was
between two flight levels then the aircraft measurements would not account for it.  Addressing
this effect is outside the scope of this paper.
The PSAP measurement of aerosol absorption is more uncertain than the aerosol scattering
measurements – PSAP uncertainty is reported to be in the 20-30% range (e.g., Bond et al.,
1999; Sheridan et al., 2002; Sherman et al., 2015).  It should be noted that the PSAP absorption
measurement represents all absorbing aerosol collected on its filter, as opposed to being
specific to 'black carbon' absorption.  That is actually helpful for this particular study as the
AERONET retrieval of AAOD also represents all flavors of absorption (e.g., 'black carbon',
'brown carbon' and dust).  Müller et al. (2011) describe detailed experiments to characterize
filter-based absorption instruments and describe some additional limitations of the instruments.
There is, however, some question of whether the PSAP (or any filter-based measurement) is
able to accurately represent absorption by particles coated with semi-volatile or liquid organics,
due to the possibility of such coatings changing the characteristics of the filter substrate
(oozing!) after impaction (e.g., Subramanian et el., 2007; Lack et al., 2008).  Comparisons of
filter-based absorption measurements for denuded and un-denuded particles (e.g., Kanaya et
al., 2013; Sinha et al., in revisions, 2017) suggest that the un-denuded particles have absorption
enhancements of 5-25% relative to those that have been through a denuder. These
comparisons show that stripping off coatings and evaporating the non-absorbing particles
reduces the measured absorption, i.e., that the effect of coatings is not completely lost in filter-
based measurements. The effect of coatings appears to increase the absorption value reported
by the PSAP relative to that reported by a non-filter-based instrument (Lack et al., 2008); in
other words the aerosol absorption values obtained from PSAP measurements may have a
positive bias.  It is worthwhile to explore the potential magnitude of such a bias. The mean mass
concentrations of organic aerosol determined from the IMPROVE measurements near BND and
SGP (the OCf value in the IMPROVE data set; Malm et al., 1994) are similar for both sites and
less than 2 $\mu g/m^3$, putting them firmly in the rural/remote category identified by Lack et al (2008;
their figure 4). Depending on whether figure 3 or figure 4 in Lack et al. (2008) is used, Lack et
al.'s (2008) results suggest that the PSAP might be overestimating absorption by a factor of 1.1
to 1.5 due to artifacts caused by organic aerosols.  However, in a subsequent study, Lack et al.
(2012) reported a PSAP overestimate by factors of 1.02-1.06 over Los Angeles, considerably
lower then the Lack et al. (2008) results.
The positive bias in absorption related to filter-based measurements is the same order of
magnitude and direction of the absorption enhancement factor found by some lab and
theoretical studies for coated absorbing particles suspended in the atmosphere. Absorption
enhancement values of 1.3-3 have been predicted for coated particles (e.g., Bond et al., 2006;
Lack et al., 2009; Cappa et al., 2012) although enhancements larger than a factor of 2 have not
been measured for ambient aerosol (e.g., Lack et al., 2008; Cappa et al., 2012; McMeeking et
al., 2014). Wang et al. (2014) suggested that an absorption enhancement factor of 1.1 was
appropriate for fossil fuel influenced aerosol and that 1.5 was a more reasonable enhancement
factor for biomass burning affected aerosol.  Biomass burning does not have a consistent
influence on either BND or SGP.  Cappa et al. (2012) suggested that the discrepancies between
ambient and modelled and/or laboratory results, could be a result of differences in particle
morphology and/or chemistry.  We have not made any adjustments for the absorption effects of
coatings or the potential positive bias in PSAP measurements as the science is still unclear.
In addition to the potential absorption enhancement due to organic coatings, it has been
suggested that aerosol water on absorbing particles may also enhance absorption. There have
been very few studies where the hygroscopic growth enhancement of absorption was explicitly
considered.  Redemann et al. (2001) modeled absorption enhancement as a function of RH
based on characteristic atmospheric particles and found absorption enhancement values of up
to 1.35 at 95% RH; for the 95[th] percentile $RH_{amb}$ values encountered at BND (78.9%) and SGP
(76.6%), the Redemann et al. (2001, their figure 2) study would predict absorption
enhancements of ~1.1.  Nessler et al. (2005) and Adam et al. (2012) utilized both ambient
aerosol measurements and Mie theory to calculate absorption enhancement values due to
hygroscopic water uptake.  Nessler et al. (2005) does not provide absorption enhancements as
a function of RH, but Adam et al. (2012) suggest absorption enhancements due to hygroscopic
growth of less than 1.1 at 80% humidity.  Brem et al. (2012) report on laboratory studies that
show that aerosol absorption was enhanced by a factor of 2.2 to 2.7 at 95% relative humidity
relative to absorption at 32% relative humidity, although for RH less than ~80% (i.e., the RH
values observed in this study) they show no absorption enhancement (their figure 9).  Lewis et
al. (2009) actually observe a decrease in absorption with increasing RH for some biomass fuels,
but hypothesize the decrease might have been due to their measurement technique and/or a
change in the morphology of the particles.
In summary, the positive bias in the PSAP measurements of aerosol light absorption might be
as high as a factor of 1.1 to 1.5 due to oozing (e.g., the overestimate of absorption reported by
Lack et al., (2008) for filter-based measurements). Atmospheric absorption may be
underestimated by PSAP measurements by up to a factor of 1.5 due to not accounting for
coating (organic or water) effects.  Without additional laboratory and field measurements to
quantify the net effect of the possible positive and negative biases in PSAP measurements of
aerosol light absorption, it is not possible to estimate the actual uncertainty in the in-situ light
absorption measurements reported here due to coating effects. To address this, we double the
assumed PSAP uncertainty of ~25% to 50% in the calculations of uncertainty.
One aspect of the in-situ system that will affect both the scattering and absorption measurement
is the gentle heating used to dry the particle to RH<40%.  The drying process we use (heating
of 40 C or less) may remove some volatile components but we believe the removal to be
minimal (<10-20%) based on lab and ambient volatility studies in the literature. Thermal
denuder studies suggest little removal of volatile components (<10%) at 40 C (e.g., Mendes et
al., 2016; Huffman et al., 2009, Bergin et al., 1997) although thermal denuders results may be
limited by short residence times (<20s).  However, smog chamber evaporation studies on
ambient aerosol over longer time periods (minutes-hours) at ambient temperature also suggest
ambient aerosol may be less volatile than previously thought – Vaden et al. (2011) showed that
ambient SOA lost just ~20% of its volume after ~4h.
Once the uncertainties in the in-situ aerosol scattering and absorption are known, the
uncertainty in SSA (SSA= $\sigma_{sp}/(\sigma_{sp}+ \sigma_{ap})$ can also be calculated. Formenti et al. (2002, their
equation 5) suggests the uncertainty in single scattering albedo ($\delta$SSA/SSA) can be calculated:
$$\delta\text{SSA/SSA} = (1-\text{SSA})*[(\delta \sigma_{sp} / \sigma_{sp})^2+(\delta \sigma_{ap} / \sigma_{ap})^2)]^{1/2} \qquad (2)$$
As an example, for scattering uncertainties of 30%, (combined nephelometer and f(RH) induced
uncertainty), PSAP absorption uncertainties of 50%, and SSA values of 0.95 (typical of the in-
situ SSA observations), equation 2 results in an in-situ SSA uncertainty of ~3% or approximately
0.03.  For the higher altitude flight segments the loading does tend to be quite a bit lower and
thus has higher uncertainty but those upper-level segments contribute little to the overall AOD
or AAOD. Because the flight column SSA is calculated using extinction-weighted SSA flight
segments, segments with very low aerosol concentrations will have little impact on the column
SSA derived from the flight measurements. Figure 3 shows the calculated SSA uncertainties for
each flight layer as well for the in-situ column SSA for each individual flight.  For high AOD
(AOD>0.3) the SSA uncertainty is quite low (less than 0.01), while for lower loading
(0.005<AOD<0.2) the SSA uncertainty is less than 0.06 (the median uncertainty in this low AOD
range for the in-situ flights in this study is ~0.03).
In addition to instrumental uncertainties there are also uncertainties associated with the aircraft
flight patterns, i.e., the presence of aerosols below, between and above the discrete flight levels.
Missing aerosol above and below an aircraft profile is a potential issue in all aircraft/column
comparisons. Different approaches have been used to assess whether aerosol loading
contributions above the highest flight level (4.6 km asl) are important.   Andrews et al. (2004)
utilized Raman lidar measurements to determine that 80-90% of the aerosol was below 3.7 km
asl at SGP (3.7 km was the maximum altitude flown by the original SGP airplane, although all
the profile flights utilized here occurred after the maximum flight level was increased to 4.6 km
asl). Andrews et al. (2004) also assumed assumed an AOD contribution of 0.005 from
stratospheric aerosol which was not done here. At SGP, Turner et al. (2001) segregated lidar
aerosol extinction profiles by season and AOD.  Their results (their Figure 1) suggest that for the
vast majority of cases observed at SGP, 5% or less of the extinction will be found above 4 km.
For low AOD cases (AOD$_{355}$<0.3) their mean extinction profiles suggest little to no aerosol
extinction between 4-7 km.   At BND, Esteve et al. (2012) noted that CALIPSO data indicated
negligible extinction above 4.6 km asl.  Regionally, seasonal average profiles from CALIPSO
also suggest there is minimal aerosol above the flight's highest level (Ma and Yu, 2014; Yu et
al., 2010).
Although statistical profile results (e.g., Turner et al., 2001; Yu et al., 2010; Ma and Yu, 2014)
suggest little contribution from high altitude aerosol layers in the region of these two sites,
Schutgens et al. (2016) demonstrates the importance of considering the specifics rather than
the statistical.  We used the Raman lidar best estimate data product of extinction profiles at
SGP to evaluate the presence of aerosol above the highest flight level at the site.  For the SGP
in-situ profiles that had matches with AERONET inversion retrievals, we identified three lidar
profiles that exhibited aerosol layers at high altitudes, but in all three cases the presence of
these layers was also hinted at by an increase in the aerosol loading at the highest flight levels
of the in-situ measurement.  Thus, we further screened in-situ/AERONET comparisons by
removing flights at SGP and BND with significant increases in loading at the highest flight levels.
There may still be aerosol layers above the level measured by the Raman lidar, but we have no
means of assessing that.  The AOD comparison presented in Figure 4 suggests we are unlikely
to be missing significant aerosol at high altitudes.
Several papers (Andrews et al., 2004; Esteve et al., 2012; Sheridan et al., 2012) have shown
that there is a high correlation ($R^2 > 0.8$) between scattering measured at the surface site (SGP
or BND) with scattering measured at the corresponding lowest flight leg, although the slopes of
the relationships indicated that the airplane measurements might be missing a fraction (10-20%)
of the aerosol below about 150 m agl.  Additionally, Esteve et al. (2012) found high correlation
(slope=1.01, $R^2$~0.7) between scattering AOD calculated by assuming the lowest leg
represented scattering in the entire layer between surface and that flight leg with scattering AOD
calculated from 1-sec data obtained during descent from the lowest flight leg to landing.  This
result suggested that no consistent bias would result from assuming the lowest flight leg was
representative of the aerosol between surface and that altitude.  We've looked at the
surface/lowest flight leg relationship specifically for the flights with matching AERONET
retrievals studied here.  We found that at BND the surface and lowest level flight aerosol
measurements were virtually identical.  At SGP the lowest level leg actually measured slightly
higher aerosol loading than was observed at the surface, which could lead to an overestimate of
the aerosol optical depth in that layer, depending on the shape of the profile.
Similarly, Esteve et al. (2012) investigated differences in aerosol scattering between and at flight
levels by comparing scattering AOD from the airplane descent between layers with that
calculated from the individual level legs in the profile. Again they were able to confirm that
measurements made during the fixed flight altitudes are representative of the aerosol near
those altitudes.
*2.4.2 AERONET uncertainties*
Uncertainties in AERONET retrievals have been reported in several papers.  Eck et al. (1999)
indicate that the uncertainty in AOD is approximately 0.01 for a field-deployed AERONET
sunphotometer at solar zenith angle = 0 (i.e., sun directly overhead).  For the almucantar
retrievals (solar zenith angle > 50) used here, the AOD uncertainty will be smaller as the
uncertainty in AOD decreases inversely with air mass (Hamonou et al., 1999; their equation 1).
Dubovik et al. (2000) report AERONET retrieved SSA uncertainties in their Table 4.  For water
soluble aerosol (the predominant aerosol type at both BND and SGP) they report that SSA
values are reliable to within ±0.03 when $AOD_{440} > 0.2$, while the uncertainty in SSA increases to
(±0.05-0.07) for $AOD_{440} ≤ 0.2$. The almucantar retrieval of SSA may be biased by errors in the
surface reflectance when the AOD is very low. Another potential issue is that the AERONET
retrievals report only one pair of (real, imaginary) refractive index values for the total size
distribution (for each wavelength). If there are two or more aerosol modes in the column, this
assumption may skew the resulting SSA and AAOD values, although the effect of such skewing
would depend on the aerosol properties and cannot be assessed here. Potential impacts in the
case of uneven mode absorption in the retrieved size distribution have been found to be minor
since the retrieved size distribution is more linked to forward scattering than absorption (pers.
comm., O. Dubovik).
Mallet et al. (2013) reports an AAOD uncertainty of 0.01 but does not indicate whether or how
the AAOD uncertainty would change with $AOD_{440}$.  Using the sum of squares propagation of
errors to calculate the uncertainty in AAOD for both high and low AAOD cases results in an
AAOD uncertainty of approximately ±0.015 for both high and low AOD cases (high $AOD_{440}=0.5$,
$\delta AOD=0.01$, SSA=0.95, $\delta SSA=0.03$, AAOD=0.026;  low $AOD_{440}=0.2$, $\delta AOD=0.01$, SSA=0.95,
$\delta SSA=0.07$, AAOD=0.011). An AAOD uncertainty value of ±0.015 suggests an uncertainty of
about 60% in AAOD for $AOD_{440}=0.5$ and more than 140% uncertainty in AAOD for $AOD_{440}<0.2$.
3. Results
In this section we first present comparisons of AOD, AAOD and SSA from the in-situ
measurements at BND and SGP with AERONET retrievals.   This includes (1) direct
comparisons of each in-situ profile with contemporaneous AERONET retrievals; the BND and
SGP comparisons are then put in the wider context of a literature review of similar direct
comparisons of in-situ and AERONET AAOD and SSA; (2) seasonal comparisons of AOD,
AAOD and SSA from Phase II AeroCom model results, AERONET retrievals and in-situ
measurements for BND and SGP; and finally, (3) we discuss these results in the context of
biases in determination of AAOD.
*3.1.1 BND and SGP: in-situ vs AERONET – Direct Comparisons*
Figures 4, 5 and 6 show the direct comparisons of AOD, AAOD and SSA at both 440 nm and
675 nm.  On all 3 plots, the blue points represent the same data set – each point indicates a
flight for which there was one or more successful AERONET Level-2 almucantar retrievals
within +/-3 hours of the end of the flight profile (if there was more than one retrieval
corresponding to a flight, the retrievals were averaged).  The thin gray lines on the 440 nm plots
indicate the reported (AERONET) or calculated (in-situ) uncertainties in the data.  Table 1
provides a comparison of the statistical values (median, mean and standard deviation) at 440
nm for each of the parameters at both of the sites for these direct comparisons (blue points in
Figures 4, 5, and 6). The low number of flights for which there are comparisons available (~10%
of total number of flights) indicate both the effects of AERONET stringent cloud screening
routine and the constraints imposed by the almucantar retrievals.  In addition to limiting the
number of comparisons available for this study, this limited data availability also has implications
for modellers utilizing AERONET data – for example, Schutgens et al. (2016) has shown the
importance of temporal collocation in measurement-model comparisons. Figure 4 also contains
red points – the red data points represent all direct sun AERONET Level-2 AOD measurements
during the +/-3 hours window around the end of each profile.  Depending on atmospheric
conditions, there may be more than one AERONET measurement within +/-3 hours of the end
of each profile, which is why in Figure 4 there are more red data points plotted than there are
flights.  The red points have not been averaged in order to provide an indication of the variability
in AOD during the in-situ profiling flight.
The comparison between in-situ and AERONET AOD is important because it can be used to
evaluate how well the in-situ and AERONET data can be expected to agree and, thus, set the
context for the AAOD and SSA comparisons.  Many studies have investigated the relationship
between in-situ and remotely sensed AOD (e.g., Crumreyrolle et al., 2014; Schmid et al., 2009,
and references therein). As noted in these studies, the in-situ derived AOD values tend to be
slightly lower than the AOD retrieved from remote sensing measurements. Figure 4 presents the
comparison of Level-2 AOD for AERONET and in-situ measurements at 440 nm and 675 nm for
two sets of AERONET AOD data. The first comparison (red points on plots) is for all direct sun
AERONET Level-2 AOD measurements.  The second comparison (blue points on plots) is for
flight-averaged AERONET Level-2 AOD measurements where all the criteria required for
almucantar retrievals are satisfied. Table 2 summarizes how many points make up each of
these data sets.
In general, Figure 4 shows that AERONET AOD tends to be higher than the in-situ AOD,
although there is good correlation between AERONET and in-situ AOD. The uncertainty bars
tend to overlap the 1:1 line suggesting that in-situ measurements provide a reasonable proxy of
the total column aerosol loading as represented by AERONET AOD.  Student t-test evaluation
suggests that the AERONET and in-situ AODs are the same at the 95% confidence level. The
coefficients of determination ($R^2$) are within the range we would expect based on the lag-
autocorrelation of scattering at these two sites (Figure 2) and the +/-3 h time window.  The $R^2$
values increase when sub-setted for the more restrictive Level-2 almucantar retrievals.  The
lower in-situ AOD values observed at both sites, compared to AERONET, may be due to the
hygroscopicity adjustment from dry in-situ to ambient RH conditions being too low or
undersampling of larger particles (e.g., Esteve et al., 2012). Esteve et al. (2012) found slopes
closer to 1 when they restricted AERONET/in-situ AOD comparison to low ambient RH (<60%)
conditions, although the AERONET AOD values were still larger than the in-situ AOD. The
effect of undersampling larger particles or underestimating aerosol hygroscopicity on the AAOD
and SSA comparisons are discussed in section 3.1.2.  Some of the discrepancy between the in-
situ and the AERONET values may also be due to the limited vertical range covered by the
airplane (150 – 4200 m asl).  We've excluded flights that might have had significant aerosol
above the highest flight level, based on Raman lidar comparisons (at SGP) and profile shapes
(at BND).   The relationships observed between AERONET and in-situ AOD for both sites are
very similar to those observed for the recent DISCOVER-AQ campaign (e.g., Crumreyrolle et
al., 2014, their figure 3).

One thing to note on Figure 4a is the blue point marked BB (the BB stands for biomass
burning).  This measurement occurred on June 28, 2006 and appears to have been strongly
affected by forest fire smoke transported from Canada.  We applied the same hygroscopicity
adjustment to the measurements of this flight as we did to all of the BND flights and, in this BB
case, the hygroscopicity correction was the primary reason the in-situ AOD value is significantly
higher than the AERONET AOD value.  This point would lie much closer to the 1:1 line if the in-
situ BB data were assumed to be hygrophobic. Previous work at the surface site at SGP has
shown that dust and smoke aerosol types tend to exhibit lower hygroscopicity than the
background aerosol normally observed at the site (Sheridan et al., 2001).   This BB point
provides an extreme example of the downside of using a constant hygroscopic growth
parameter as a function of RH, although without additional information about the aerosol for
each profile it is difficult to do otherwise.  The light blue dotted line on Figure 4 represents the
relationship between AERONET and in-situ data if the BB point is excluded.
Figure 5 presents the comparison of AAOD for flight-averaged AERONET and in-situ
measurements. As described above, the AERONET AAOD values shown in Figure 5 are what
we have termed Level-1.5* data – i.e., they are from Level-1.5 almucantar retrievals when there
was a valid Level-2 almucantar retrieval, but the $AOD_{440}$>0.4 constraint was not applied.  In
contrast to the AOD comparison depicted in Figure 4, the AERONET Level-1.5* AAOD values
are significantly higher than the in-situ AAOD values.  Figure 5 also shows that the correlation
between the AERONET and in-situ AAOD is poorer than it was for AOD, particularly at BND ($R^2$
is 0.49 at BND and 0.68 at SGP for the 440 nm comparison).  The lower correlation at BND is
somewhat surprising given the lag-autocorrelation results for aerosol absorption (Figure 2a) at
the BND surface site.  Surprisingly, while the BND site has higher 3-hour autocorrelations for
absorption than SGP (R = 0.75 for BND and R = 0.55 for SGP, per Figure 2), the results for
BND in Figure 5 indicate less correlation than at SGP for absorption. Nonetheless, the
correlation coefficients for BND in Figure 5 ($R^2$=0.49 (blue) and 0.37 (red) correspond to R =
0.70 (blue) and 0.61 (red)) are not that far from the 3 h auto-correlation of r(k=3h)=0.75 for
absorption at BND in Figure 2.  For AAOD the uncertainty bars, while wider, exhibit significantly
less overlap with the 1:1 line (indeed no overlap at SGP) and indeed the student t-test suggests
the AERONET and in-situ AAOD values are different at the 95% level at both sites.
Both Figure 5 and the median values provided in Table 1 indicate that AERONET Level-1.5*
AAOD tends to be larger than the in-situ AAOD, although the scatter in the relationships
(particularly at BND) suggests that a multiplicative factor doesn't represent the relationship very
well. The purple points in Figure 5 indicate AAOD retrievals where the flight-averaged
$AOD_{440}$>0.2.  There is no obvious improvement of the relationship between in-situ and
AERONET AAOD when these points are considered (although there are only 1-4 comparison
points above $AOD_{440}$>0.2 for each site).
The AAOD comparisons at 675 nm at BND (Figure 5c) are quite similar to those at 440 nm,
suggesting that there is little contribution to absorbing aerosol from dust, organic carbon and/or
$NO_2$.  In contrast, at SGP, there is a change in the relationship between AERONET and in-situ
AAOD from 440 to 675 nm indicating that one or more of these components may affect the 440
nm comparisons at that site (Figure 5d). Recent work by Engelbrecht et al. (2016) has
suggested that even at 405 nm most dusts have have SSA values > 0.9 meaning they are not
much more absorbing than the aerosol typically observed at BND and SGP. Further, Ångström
exponent values from the matched AERONET and in-situ profile data do not support the
presence of dust, while the rural nature of the site suggests significant levels of $NO_2$ are
unlikely. Thus the most likely explanation is the presence of organic carbon, although the
IMPROVE sulfate and organic data used to estimate aerosol hygroscopcity do not support this.
The IMPROVE measurements tend to suggest a relatively small contribution of organics to the
aerosol mass with the average mass concentration of organics only 40 to 60% that of sulfate
aerosol mass concentration for BND and SGP, respectively. In contrast, the Aerosol Chemical
Speciation Monitor (ACSM) measurements by Parworth et al. (2015) indicate that, depending on
the month, organic aerosol can contribute up to 70% of the total aerosol mass at SGP.
Figure 6 presents the comparison of column SSA retrieved from flight-averaged AERONET
inversions (Level-1.5* data) with the column SSA calculated from in-situ profile measurements
of aerosol scattering and absorption at BND and SGP. Consistent with the AOD and AAOD
comparisons (Figures 4 and 5) the SSA retrieved from AERONET tends to be much lower than
the SSA calculated from the in-situ profile measurements. As with AAOD, the SSA uncertainty
bars exhibit little overlap with the 1:1 line and a student t-test suggests the AERONET and in-
situ SSA values are different at the 95% level for both BND and SGP. At both sites the range in
AERONET-retrieved SSA is much wider than the range in column SSA obtained from the in-situ
profiles. Long term, in-situ measurements at the BND and SGP surface sites yield mean SSA
values of 0.92 and 0.95 respectively (Delene and Ogren, 2002, based on monthly-averaged
data). Delene and Ogren's (2002) surface SSA values are reported at low RH (RH<40%) and
550 nm; adjusting them to ambient conditions and 440 nm would likely cause them to increase
making them more comparable to the in-situ column SSA depicted in Figure 6 but even less like
the AERONET Level-1.5* SSA values. As with Figure 5, the purple points on Figure 6 indicate
when the flight-averaged $AOD_{440}$>0.2; although there aren't enough points to draw a robust
conclusion, there does not appear to be an improvement in the relationship between in-situ and
AERONET SSA when only these purple points are considered.
Figure 6 also includes a set of 'hybrid SSA' ($SSA_{hybrid}$) points in yellow. These points have been
calculated using the AERONET AOD and the in-situ AAOD:
$$SSA_{hybrid} = (AOD_{AERONET} - AAOD_{PSAP})/AOD_{AERONET} \qquad (3)$$
This hybrid approach to SSA eliminates the uncertainty associated with the empirical
hygroscopic growth factors applied to the in-situ scattering measurements, and also removes
the scattering uncertainty associated with undersampling the coarse mode. It does not,
however, eliminate the uncertainties associated with assuming the absorbing aerosol is
hygrophobic, that there is little absorption in the potentially undersampled coarse mode, or the
unknown contribution from absorption enhancement. $SSA_{hybrid}$ is very similar to the SSA derived
from in-situ measurements, suggesting the primary discrepancy between the AERONET SSA
and the in-situ SSA is due to the determination of the absorbing nature of the aerosol, either due
to issues with the limitations of the filter-based measurements or to the interpretation of the
relative contribution of aerosol absorption from the AERONET inversion retrieval products.

*3.1.2 How might in-situ hygroscopicity assumptions and under-sampling of the aerosol affect*
*SSA and AAOD comparisons?*

Figure 4 shows that the AERONET AOD may be slightly larger than the in-situ AOD, while
Figures 5 and 6 suggest that the AERONET retrievals significantly overestimate the amount of
absorbing aerosol (low SSA, high AAOD) relative to the in-situ measurements. The slight
deviation between in-situ and AERONET AOD may lead to questions about whether directly
comparing other AERONET and in-situ parameters (e.g., SSA, AAOD) is a reasonable thing to
do and whether the AAOD and SSA comparisons shown in Figures 5 and 6 are related to
issues with the AOD comparison. As mentioned above, Esteve et al. (2012) suggested the
AOD difference was most likely due to either underestimating the hygroscopic growth correction
and/or undersampling of supermicron particles by the aircraft inlet. In this section we evaluate
how these two possible causes of the AOD discrepancy might affect the SSA and AAOD
comparisons.
Increasing the hygroscopic growth adjustment of the in-situ measurements would enhance the
in-situ scattering values used to calculate the in-situ AOD, but would not change the in-situ
AAOD because the absorbing particles are assumed to be non-hygroscopic. Consequently, the
comparison depicted in Figure 5 would not change with a different adjustment for hygroscopic
growth. Increasing the in-situ AOD, without affecting the in-situ AAOD, would result in higher in-
situ SSA values and an even greater discrepancy between AERONET and in-situ SSA values
than shown in Figure 6. To evaluate the effect of assuming absorbing particles were non-
hygroscopic, a sensitivity test was performed assuming the absorption enhancement due to RH
was the same as the hygroscopicity scattering enhancement, i.e., $\sigma_{ap}(RH_{amb})/\sigma_{ap}(RH_{dry})=a*(1-$
$(RH_{amb}/100))^{-\gamma}$. While this is likely an extreme assumption, it had minimal effect on the
comparisons of AOD, AAOD and SSA.
The other likely candidate to explain the in-situ AOD being slightly lower than the AERONET
AOD is aircraft under-sampling of super-micron aerosol particles due to the 5 $\mu$m inlet cutoff'.
Esteve et al.'s (2012) comparison of column in-situ and AERONET scattering Ångström
exponents at BND suggested that the airplane measurements might be under-sampling larger
particles. Sheridan et al. (2012) estimated that the aircraft inlet 50% cut-off aerodynamic
diameter is approximately 5 $\mu$m, so particles larger than that are unlikely to be sampled by the
in-situ measurements but will be sensed by the AERONET sunphotometer. If we take into
account that atmospheric particles are likely to have a density greater than 1 g cm$^{-3}$, the actual
cut size would be closer to 3 or 4 $\mu$m. The AERONET volume size distributions were used to
estimate the fraction of column extinction due to particles less than 3 $\mu$m. At BND the mean and
standard deviation of the 3 $\mu$m extinction fraction (extinction(D<3$\mu$m)/extinction(D<30$\mu$m)) was
0.93 ±0.07, while at SGP the extinction fraction value was 0.88 ± 0.09. At the BND and SGP
surface sites, most (80-90%) of the observed sub-10 $\mu$m scattering and absorption is also
attributed to sub-micron aerosol, with absorption more likely to be in the sub-micron size range
than scattering (Delene and Ogren, 2002; Sherman et al., 2015). This is consistent with the
observation that absorbing aerosol tends to be concentrated in sub-micron particles for typical
aged continental air masses (e.g., Hinds, 1982). Based on these observations, larger and
primarily scattering particles are more likely to be under-sampled by the in-situ measurements
than absorbing particles.  This is the opposite of what is needed to explain the discrepancies
between AERONET and in-situ AOD, AAOD, and SSA shown in Figures 4-6. The in-situ
measurements would need to preferentially under-sample absorbing aerosol relative to
scattering aerosol in order to come into line with the AERONET observations.  Additionally,
Sheridan et al. (2012) calculated particle transmission losses from behind the sample inlet on
the airplane to both the nephelometer and PSAP to be similar and to be less than 10% in the
particle diameter range $0.01<D<1$ $\mu$m.  This suggests that preferential losses of absorbing
aerosol are also unlikely to occur downstream of the aerosol inlet.  In summary, we can only see
two ways that the in-situ measurements can sample aerosol efficiently enough to represent
AERONET AOD fairly well but significantly underestimate AAOD and overestimate SSA: (1) not
accounting properly for the effect of coatings (organic or water) on absorption enhancement
which we've discussed in detail in the manuscript and (2) not sampling layers of predominantly
absorbing aerosol below, between, and/or above the flight layers.  We suspect that the SSA
required of such layers in order to explain the AAOD and SSA discrepancies is physically
impossible.
3.2 Literature survey: in-situ vs AERONET – Direct Comparisons
Direct comparisons at BND and SGP suggest that AERONET retrievals underestimate SSA
and, consequently, that AERONET overestimates AAOD relative to in-situ measurements of
AAOD for the low AOD conditions typical at these two sites.  The next question to address is
whether this discrepancy, found for two rural, continental sites in the central US with relatively
low aerosol loading, is more widely observed for direct in-situ/AERONET comparisons at a
variety of sites/conditions. As in section 3.1, the focus in this section is on direct comparisons of
column-averaged SSA (or AAOD) derived from in-situ measurements made during aerosol
profiling flights that were flown in close proximity (temporal and spatial) to an AERONET
retrieval. Tables 3 and 4 summarize literature results describing the direct comparisons of
AERONET retrievals with in-situ aerosol profile measurements for AAOD and column SSA.
Figure 7 provides a graphical overview of the SSA comparisons described in Table 4. Tables 3
and 4 and Figure 7 also include the BND and SGP comparisons described in this study.  With
the possible exception some of the profiles reported by Corrigan et al. (2008), the literature
comparisons cited in Tables 3 and 4 and shown in Figure 7 have been made at higher AOD
conditions ($AOD_{440}>0.3$) to reduce retrieval uncertainty.  In contrast, the SGP and BND
comparisons are more representative of global AOD (Figure 1) with the majority of the
comparisons at BND and SGP occurring for $AOD_{440}<0.2$.  Please note that some of the earlier
studies shown in Figure 7 and described in Table 4 used values from Version 1 AERONET
Level-2.0 data.  Where that was the case, we retrieved Version 2 AERONET Level-2.0  data
from the AERONET website and those Version 2 data are what is reported in Table 4 and
depicted in Figure 7.  The comments section of Table 4 mentions the cases where this was
done.  For some of these references we also retrieved the $AOD_{440}$ values from the AERONET
website as the $AOD_{440}$ values weren't reported in all papers.

Tables 3 and 4 have been restricted to studies with direct comparisons of column-averaged
AAOD or SSA retrieved from full in-situ vertical profiles flown near (within ~100 km) AERONET
sites within a few hours of the AERONET retrieval, i.e., studies that are comparable to the BND
and SGP studies described in Section 3.1.  For non-plume data sets, Anderson et al. (2003)
found autocorrelations $\geq$ 0.8 at 100 km (their figure 6). For plume-influenced data sets they
found autocorrelations ~0.6**.** Included in the tables are the field campaign name (if applicable),
number of AAOD or SSA comparisons, the primary type of aerosol studied, summary of AOD
comparisons (if available), altitude range covered by the airplane, instruments and data
processing (e.g., instrument corrections, treatment of hygroscopicity, wavelength adjustment)
and a summary of the results of the AAOD comparison. The last column in Tables 3 and 4
includes information on the spatial and temporal differences between the in-situ measurements
and AERONET retrievals and comments on treatment of the AERONET and in-situ data.  The
last column also notes how each campaign dealt with aerosol below and above the in-situ
profile if reported. It should be noted that the number of SGP and BND comparisons of AAOD
and SSA in Tables 3 and 4 are only possible because we've utilized AERONET retrievals below
the recommended threshold of $AOD_{440}>0.4$.  The uncertainty for the BND and SGP
comparisons is much higher than for some of the other direct comparisons due to the low AOD
conditions observed at these sites.
For the three AAOD closure studies listed in Table 3 (the BND and SGP results presented here,
plus results from a field campaign over the Indian Ocean) the AERONET retrievals indicate
more absorbing aerosol in the column than is suggested by the corresponding in-situ
measurements.  The Corrigan et al. (2008) paper mentioned in Table 3 is the sole
AERONET/in-situ AAOD comparison cited by Bond et al. (2013), as it was the only published
direct AAOD comparison available.  Corrigan et al. (2008) present no AOD comparisons that
could provide an indication of their sampling system efficiency, and information about the
wavelength of the comparisons and profiles specifics are lacking.  To our knowledge, no other
direct comparisons of in-situ and AERONET AAOD are available in the literature.
The SSA comparison studies listed in Table 4 and visually summarized in Figure 7 indicate that,
even at higher AOD, AERONET retrievals tend to indicate more-absorbing aerosol (lower SSA)
relative to in-situ measurements, although most of the values are within the combined standard
uncertainty of the AERONET and in-situ values indicated by the shading (see BIPM, 2008, their
equation 16 for how the combined standard uncertainty was calculated). Of the 63 cases
depicted in Figure 7, 16 cases (~25%) of the AERONET/in-situ comparisons were within 0.02.
While much of the observed difference between $SSA_{in-situ}$ and $SSA_{AERONET}$ may fall within the
uncertainty of the SSA values, as noted in Schafer et al. (2014), the fact that the difference
($SSA_{AERONET} - SSA_{in-situ}$) is predominately negative across all the direct comparisons found in the
literature is not what would be expected from random error.  Figure 7 also shows the mean and
2*standard deviation of all of the points (black square and vertical lines) and just the literature
value points (black diamond and vertical lines).  Based on the characteristics of a normal
distribution the standard deviation lines suggest ~80% of the points will be negative – random
error would suggest only 50% of the points should be negative.  Figure 7 suggests that
AERONET retrievals of SSA could perhaps be used at $AOD_{440}<0.4$, perhaps down to
$AOD_{440}$~0.25 or ~0.3 – even at those low AOD values the differences in SSA between
AERONET and in-situ still tend to be within the AERONET uncertainty.  However, as Figure 7
shows, there are not a lot of direct comparisons to support such a choice.
Most of the SSA comparisons in Table 4 reported fairly good agreement between AERONET
and in-situ AOD, implying that  the discrepancy is associated with the absorption values rather
than the scattering values (since scattering is typically 90% of extinction .  This is consistent with
the AERONET AAOD values being greater than those obtained from in-situ measurements
presented in Table 3. Out of the 63 profiles compared in Table 4, there are four exceptions,
(three from Leahy et al. (2007) and one from this study for the BND site) where $SSA_{AERONET}$ is
larger than the corresponding $SSA_{in-situ}$.  Interestingly, the three exceptions from Leahy et al.
(2007) were for their high AOD ($AOD_{550}$>0.6) cases; for their two low AOD ($AOD_{550}$<0.3) cases
the opposite was found, i.e., $SSA_{AERONET}$<$SSA_{in-situ}$.
In summary, the literature survey featuring measurements across the globe for many aerosol
types suggests that even at higher AOD conditions, direct comparisons of AERONET with in-
situ aerosol profiles find that AERONET column SSA is consistently lower than the SSA
obtained from in-situ measurements (although mostly within the uncertainty of the AERONET
SSA retrieval and in-situ measurements).  If there was no consistent bias in the AERONET/in-
situ comparison we would expect (AERONET_SSA – INSITU_SSA) to be evenly distributed
around zero.  Instead, Figure 7, which summarizes the literature survey, suggests either that
AERONET retrievals are biased towards too much absorption, or that in-situ, filter-based
measurements of aerosol absorption are biased low.  We note that the results from the literature
(e.g., Figure 7) indicate that the hypothesized low-bias in in-situ absorption is not associated
with a single airplane's measurement system or the atmospheric conditions encountered in a
single experiment. That leaves us with possible bias in the in-situ experimental methods
(instrument issues (nephelometer, PSAP), treatment of f(RH), vertical coverage, sampling
artifacts), all of which we have attempted to address above.
An alternative explanation is that the AERONET SSA uncertainties are non-symmetric.  Dubovik
et al. (2000) suggest that simulated retrievals of SSA for 'water soluble aerosol' are asymmetric
when different 'instrumental offsets' are assumed, particularly at lower AOD values (0.05 and
0.2). Their figure 4 shows a much larger decrease in SSA for some instrumental offsets relative
to the increase in SSA observed for an instrumental offset of the same magnitude but opposite
sign. Asymmetry is also indicated for 'biomass burning' aerosol (their figure 7) although the
asymmetry is in the opposite direction, i.e., the increase in SSA is larger than the decrease for a
given pair of instrumental offset values. It is not obvious from their figure 7 whether the retrievals
are asymmetric for simulated dust aerosol. Interestingly, at least three of the four points in
Figure 7 with AERONET_SSA>INSITU_SSA represent retrievals of biomass burning aerosol.
3.3 BND and SGP: in-situ vs AERONET and AeroCom model output – Statistical Comparisons
Most of the statistical comparisons between AERONET and in-situ profiles (e.g., Ramanathan et
al., 2001; Leahy et al., 2007; Ferrero et al., 2011; Johnson et al., 2011) were for short-term field
campaigns with a limited number of in-situ profiles.  The advantage of the multi-year, in-situ
vertical profiling programs at BND (401 flights) and SGP (302 flights) is that we can compare the
statistics for both in-situ and AERONET values as opposed to comparing individual in-situ
values to remote retrieval statistics. Figure 1 in Andrews et al. (2011) and Figure 9 in Sheridan
et al. (2012) demonstrate that the BND and SGP flight programs captured the multi-year
seasonality in aerosol properties at these two sites. Because of the large number of flights over
an extended period of time, Skeie et al. (2011) was able to compare the seasonally averaged,
in-situ absorbing aerosol profiles from BND and SGP with seasonal vertical profiles of black
carbon generated by the Oslo-CTM2 model.  Skeie et al. (2011) found that the model
underestimated absorbing aerosol relative to the BND and SGP in-situ profiles for most seasons
and altitudes, although agreement between the model and measurements tended to be better at
higher altitudes.
As mentioned in the introduction, AERONET retrievals of AAOD have been used to suggest
upscaling factors for modelled values of absorbing aerosol (e.g., Sato et al., 2003; Bond et al.,
2013).  These model/AERONET comparison studies are typically based on model and
measurement statistics (i.e., properties are averaged over time and region) rather than direct
comparisons due to both computational constraints and the discrete nature of the AERONET
measurements.   Given the statistical nature of some historical AERONET/in-situ comparisons
as well as the typical model/AERONET comparison constraints, in this section we compare
monthly statistics for in-situ measurements, AERONET retrievals and AeroCom model output.
It should be reiterated here that we are comparing asynchronous data  and that there
are some additional differences amongst the data sets that need to be kept in mind:  the
AERONET data are rigorously cloud-screened (although cloud halo effects may persist
(e.g., Jeong and Li, 2010) and only obtained during daytime; the in-situ measurements
are also daytime-only and the airplane did not fly in-cloud due to FAA flight restrictions,
but may have flown near clouds; and the model data include day and night with clouds
and also represent values over a 1x1 degree grid.
Figure 8 shows the 440 nm monthly medians of AOD, AAOD and SSA at BND and SGP based
on the in-situ profile measurements, and two versions of AERONET retrievals as described
below. For the in-situ properties, all profiles were used, regardless of whether there was an
AERONET retrieval corresponding to the flight.  The AERONET monthly medians in Figure 8
use the long-term (1996-2013) AERONET data record for each site.  As described previously,
the lines labeled AERONET 1.5* were calculated from Level-1.5 inversion data with matching
Level-2 almucantar retrievals. The lines labeled AERONET 2.0 utilized only Level-2 almucantar
retrieval data.  In both cases the median AERONET AOD values represent those Level-2 AOD
measurements for which there was also an AAOD and SSA retrieval, ensuring that the
AERONET AOD medians represent the same set of retrievals as the corresponding AAOD and
SSA medians in the figure. The AERONET Level-1.5* AOD monthly medians are representative
of the direct sun AERONET Level-2 AOD climatology at the two sites.  Figure 8 also includes
the AeroCom Phase II model monthly medians for BND and SGP (Kinne et al., 2006, Myhre et
al., 2013) with model emissions, meteorology and other details briefly described in Myhre et al.
(2013).  The AeroCom values, which were provided at 550 nm, have been adjusted to 440 nm
using the reported AeroCom monthly scattering Ångström exponent to adjust AOD wavelength
and assuming an absorption Ångström exponent of 1 for the AAOD wavelength adjustment.  It
should be noted that the three monthly data sets (AERONET, AeroCom, and in-situ) plotted in
Figure 8 are derived from measurements for overlapping, but not identical time periods, i.e.,
these plots represent climatological comparisons rather than direct comparisons of the data
sets.
At both sites, the climatological seasonal patterns for AOD (i.e., high in summer, low in winter)
are similar for the three data sets: in-situ measurements, AERONET Level-1.5* retrievals (recall
that the AERONET 1.5* AOD is representative of the overall AERONET AOD climatology at
each site) and AeroCom model output.  At BND the AeroCom model AOD tends to be larger
than the in-situ and AERONET 1.5* AOD values by up to a factor of two. AERONET 1.5* AOD
is larger than the in-situ AOD in the summer (by up to 50%) but quite close the rest of the year
(typically within 20%).  While a 50% discrepancy between the AERONET and in-situ climatology
may appear significant, it's important to remember that these data sets do not represent the
same period of time or measurement conditions (e.g., time of day, cloud cover, aerosol events,
ambient humidity, etc).  Schutgens et al. (2016) shows there can be large differences when
comparing values obtained with different samplings (more than 100% for AOD), particularly
when there are high levels of variability in the data.  At SGP the AOD monthly medians from in-
situ measurements and AERONET Level-1.5* are almost identical for August-December, with
slightly more discrepancy among the AOD values in summer and early part of the year.  In
contrast, AeroCom model median AOD values tend to agree better with AERONET 1.5* and in-
situ AOD values from January-July but are noticeably higher (up to a factor of 2) in the later half
of the year.  At both sites, the median AERONET Level-2 AOD values (corresponding to AAOD
and SSA retrievals) are much higher (by a factor of 2 or more) than the Level-1.5* and in-situ
climatologies due to the $AOD_{440}>0.4$ constraint.  During the cleanest, lowest humidity, and often
cloudiest months of the year (December-February) there are none to few Level-2 almucantar
retrievals of SSA and AAOD at either BND or SGP – the gray lines in Figures 8ab are lacking
data points for Jan., Feb. and Dec. at BND and Jan. and Dec. at SGP.
For AAOD at BND, the AeroCom model output falls between the AERONET 1.5* and in-situ
values, with AERONET 1.5* AAOD being higher than the in-situ data by up to a factor of 8. As
with AOD, the AERONET AAOD Level-2 values are much higher than the in-situ or modelled
AOD values due to the constraint that they are only retrieved at high loading conditions
($AOD_{440}>0.4$). The three data sets (AeroCom, in-situ and AERONET 1.5*) agree best in the
month of May when the median values of AAOD are within 30%.  At SGP there is fairly good
agreement between AeroCom model and in-situ AAOD for the first 7 months of the year, while
the AERONET 1.5* monthly AAOD values are considerably higher for that same time period.
For the latter part of the year the in-situ AAOD values tend to be lower than both AERONET and
AeroCom AAOD values.
The AERONET 1.5* SSA values tend to be quite a bit lower at BND, and somewhat lower at
SGP, which is why the AERONET 1.5* AAOD values tend to be higher (recall that for
AERONET data AAOD is calculated using AAOD=(1-SSA)*AOD).  Figure 8 also shows that the
AERONET Level-2 SSA values are similar to the monthly in-situ and AeroCom SSA medians
between April and November. There are no AERONET Level-2 almucantar retrievals of SSA in
January or December at either site. For the February and March, median Level-2 almucantar
retrievals of SSA are based on very few data points resulting in bigger discrepancies between
AERONET Level-2 almucantar retrievals of SSA and the in-situ and AeroCom SSA values.
Aside from differences in magnitude, there are also differences in the seasonal patterns of AOD,
AAOD and SSA for the three data sets (in-situ, AERONET 1.5* and AeroCom).  For example, at
BND, the AERONET and in-situ AAOD both have a bi-modal annual distribution with peaks in
late spring and early fall, which is not captured by the AeroCom AAOD and which is not seen in
the AOD seasonality.  The observed seasonal differences may be a result of (a) the different
climatology time ranges for each method and/or (b) very little overlap in the measurement times
for AERONET and in-situ measurements or (c) in the case of the models, not capturing local
emissions near the sites.  This highlights the importance of direct (i.e., near in time and space)
comparisons in order to understand these seasonal differences.  The seasonal cycle plots in
Figure 8 also direct attention to the fact that AOD and AAOD vary independently rather than
exhibiting the same seasonal pattern. This suggests that different emission sources and/or
atmospheric processes control the variability of absorption and scattering aerosol over the
course of the year.
3.4 Discussion
Because AERONET data are readily available and are being widely used as a benchmark data
set for evaluating model output of AAOD (e.g., Chung et al., 2012; Bond et al., 2013; He et al.,
2014; Wang et al., 2014) as well as for comparison with satellite retrievals and development of
AAOD climatologies, we document and discuss some of the previous methods for utilizing
existing AERONET retrievals that have been used to estimate AAOD at low AOD ($AOD_{440}<0.4$)
where Level-2 retrievals do not exist. These approaches fall into several categories (1) use only
Level-2 data; (2) use Level-2 and Level-1.5 data with acknowledgement of greater uncertainty in
the retrievals and potentially additional measurement constraints for the Level-1.5 data; (3)
make climatological assumptions about the representativeness of Level-2 SSA for low AOD
conditions to obtain AAOD.
Clearly the simplest approach to minimize uncertainty in retrieved AERONET AAOD and SSA is
to only use AERONET Level-2 retrievals which include the $AOD_{440}>0.4$ constraint. This
approach has been and continues to be used (e.g., Koch et al., 2009; Bahadur et al., 2010;
Chung et al., 2012; Buchard et al., 2015; Pan et al., 2015; Li et al., 2015).  However, as shown
in Figure 1 the vast majority of the globe has $AOD_{440}<0.4$, meaning few if any AERONET Level-
2 AAOD or SSA retrievals will be available for most locations.  This approach is quite useful in
regions (or for case studies) with high aerosol loading (high AOD). However, excluding low
loading conditions is likely to cause AERONET AAOD statistics to be biased high.  This is
particularly important when evaluating models in clean locations such as the Arctic. The
$AOD_{440}>0.4$ constraint may also affect the SSA statistics.
Some studies have utilized AERONET Level-1.5 retrievals of absorption-related aerosol
properties in order to avoid being limited to the high AOD levels required by Level-2 data (e.g.,
Lacagnina et al., 2015; Mallet et al., 2013).  These studies note that Level-1.5 data include more
relevant AOD values but that there are accompanying higher uncertainties in the retrievals for
absorption related properties.   Mallet et al. (2013) use Level-1.5 data to evaluate the spectral
dependence of aerosol absorption. Lacagnina et al. (2015) utilize both Level-2 and Level-1.5
AERONET data in their comparison with PARASOL satellite retrievals of SSA and AAOD.  For
the Level-1.5 data they apply the additional requirement that the solar zenith angle must be
≥50°. Lacagnina et al. (2015) find quite good agreement (within +/- 0.03) for AAOD and note
that larger differences between PARASOL and AERONET retrieval occur at higher AOD
conditions, possibly due to less homogenous aerosol (i.e., plumes).
A more sophisticated approach to deal with SSA (and hence AAOD uncertainties) at low AOD is
implemented by Wang et al. (2014).  They make the assumption that SSA is independent of
AOD (at least as a function of season) and utilize climatological Level-2 SSA values for each
season with the measured AOD in order to obtain AAOD. The seasonal climatologies of SSA
are based on 12 years of Level-2 AERONET data. For the two US continental sites studied in
this paper, the approach of Wang et al. (2014) would likely minimize the potential AERONET
tendency towards high AAOD at low AOD conditions as the Level-2 monthly climatological SSA
values are quite similar to SSA values obtained by in-situ measurements (Fig. 8).
A similar, though statistical, approach was used in Bond et al.'s (2013) bounding BC paper in
order to reduce the uncertainty and better represent AERONET SSA and AAOD retrievals at
low AOD. Bond et al. (2013) worked with AERONET monthly local statistics for the time period
2000-2010. Monthly values of AAOD and SSA at 550 nm were calculated from size distributions
and refractive index when there were at least 10 valid inversion retrievals for that month at that
site in the 2000-2010 period (most sites had more than 10 retrievals in a given month over the
11 year period). It was assumed in Bond et al. (2013), based on AERONET reported
uncertainties, that the retrieved absorption-related values were more reliable at larger AOD and
so they made some adjustments to account for this. For each site, AAOD and SSA values were
binned as a function of AOD (there were five AOD bins, with each bin corresponding to 20% of
the AOD probability distribution).  For lower AOD conditions, the calculated AAOD and SSA
values were replaced by values obtained during larger AOD conditions for the same month as
follows: (i) the SSA and AAOD values corresponding to $AOD_{550}$ of 0.25 were prescribed for all
SSA and AAOD observations at lower AOD and (ii) for locations where all $AOD_{550}$<0.25, the
average SSA and AAOD of the upper 20$^{th}$ percentile of AOD observations at the site was
prescribed for all lower AOD bins.  Finally, the average of all five bins was used to determine the
overall monthly average.  In the case of AAOD the bin averages were simply averaged to get
the monthly value while for SSA the AOD-weighted bin averages were averaged to get the
monthly value.   Note: the $AOD_{550}$=0.25 cutoff point used in Bond et al. (2013) corresponds
(approximately) to $AOD_{440}$=0.35 for smaller particles and $AOD_{440}$=0.25 when large particles are
present.  Thus it is less strict than the AERONET recommended constraint of $AOD_{440}$>0.4, but it
had been suggested that the recommended constraint might be too restrictive (pers. comm., O.
Dubovik).
One drawback affecting approaches using climatological values of SSA (e.g., Wang et al., 2014;
Bond et al., 2013) is that they may not account for the systematic variability that has been
observed between SSA and loading at many sites, although AOD is usually more variable than
the composition (or SSA). Still some studies with in-situ data (e.g., Delene and Ogren, 2002;
Andrews et al., 2013; Pandolfi et al., 2014; Sherman et al., 2015) indicate that SSA
systematically decreases with decreasing aerosol loading. A similar SSA/AOD systematic
variability relationship is also observed at some North American AERONET sites. Schafer et al.
(2014; their figure 6) show SSA decreasing at lower loading for the GSFC site near Washington
D.C. during the period of their field campaign; they also show similar relationships between SSA
and AOD based on the long-term data for three mid-Atlantic AERONET sites.  Additionally, a
quick survey (not shown) of other long-term North American AERONET sites with good
statistics (i.e., lots of points) for Level-1.5 SSA retrievals (e.g., Billerica (Massachusetts), Bratts
Lake (Saskatchewan, Canada), COVE (Virginia), Egbert (Ontario, Canada), Fresno (California),
Konza (Kansas), SERC (Maryland), and University of Houston (Texas)) indicates this
systematic relationship may be observed at a wide range of locations in North America. Such
climatological analyses may mask short-lived and/or infrequent aerosol events (e.g., dust or
smoke incursions) that may have significantly different optical properties.
Figure 9 shows the systematic relationships between $SSA_{440}$ and $AOD_{440}$ for BND and SGP for
both the AERONET retrievals and in-situ profile measurements. Consistent with previous
figures, we have utilized SSA values for $AOD_{440}<0.4$ when there was a valid Level-2 AOD
inversion retrieval, i.e., what we call AERONET Level-1.5*.  Also included on the figure is a line
showing the $SSA_{550}$ versus scattering ($\sigma_{sp,550}$) relationships for the surface measurements at
BND and SGP.  The surface measurements are made at low RH conditions (RH<40%) and
adjusted to ambient RH using the available meteorological measurements at the site (ambient
RH at 2 m at SGP and ambient RH at 10 m at BND); adjustment of the surface measurements
from dry to ambient conditions shifts the $SSA_{550}$ values upward (assuming absorption is not
affected) and the scattering values to the right.
Figure 9 suggests that for all three sets of measurements at both sites, there is a consistent
decrease in SSA as aerosol loading decreases below $AOD_{440}=0.2$.   This relationship implies
that a climatology based on SSA values measured at high AOD may underestimate the AAOD
climatology. The AERONET SSA values are lower than the in-situ profile values as would be
expected from the results presented in sections 3.1 and 3.3. The AERONET SSA values are
also lower than the surface in-situ SSA values – the surface in-situ SSA values adjusted to
ambient conditions are quite similar to those obtained from the in-situ vertical profiles. It should
however be noted that despite the discrepancy between in-situ and AERONET SSA values,
Figure 9 shows that the SSA values for all three sets of measurements at SGP are within the
reported AERONET SSA uncertainty range of 0.05-0.07 for $AOD_{440}<0.2$ across the narrow and
low AOD range shown in the figure.  At BND the SSA values are within the AERONET SSA
uncertainty range down to $AOD_{440}\sim0.1$.  At the lowest AOD values ($AOD_{440}<\sim0.05$) the
AERONET SSA values diverge, consistent with very large uncertainties expected in the
AERONET SSA retrievals in the cleanest conditions.  Uncertainty in the AERONET AOD
retrieval may begin to affect the AERONET SSA retrieval where +/- 0.01 AOD uncertainty is
equivalent to a 20% change in AOD for AOD of 0.05. In addition, at such low AOD values, the
surface reflectance uncertainties may influence AERONET's retrieval of SSA. Figure 9
suggests that, in terms of the shape of the systematic variability plot, there are no obvious
retrieval issues for AERONET SSA retrievals in the range $0.05<AOD_{440}<0.2$, although this is in
the AOD range where high uncertainty in the SSA retrieval is expected (Dubovik et al., 2000).
There are large differences (orders of magnitude) in the number of data points in each of the
data sets; the number of points in each bin is indicated by the color-coded histograms shown on
Figure 9. The mean standard error (MSE) in SSA (MSE=(standard deviation)/(number of
points)$^{1/2}$) is indicated by the shading surrounding the solid colored lines. The MSE is quite
similar for the AERONET 1.5* and in-situ profile measurements across the AOD range plotted in
Figure 9, suggesting the observed systematic variability is not merely due to small numbers of
data points in each bin, particularly at lower loading. However, the fact that the AERONET MSE
is approximately the same as the in-situ profile MSE, despite having approximately an order of
magnitude larger number of points/bin, indicates that variability in the retrieved AERONET SSA
is larger than the variability in SSA derived from in-situ profile measurements.
This study has utilized a valuable but spatially limited (i.e., two rural continental North American
sites) climatological vertical profile dataset to explore AERONET retrievals of AAOD and SSA.
Clearly, one way to address the observed discrepancy between in-situ and AERONET AAOD is
to pursue a focused measurement program designed to acquire statistically robust in-situ
vertical profiles over AERONET sites representing a wide range of conditions and aerosol types.
This type of measurement program has been proposed to evaluate satellite retrievals and better
characterize atmospheric aerosol (R. Kahn, SAM-CAAM, pers. comm.). Further evaluation and
development of in-situ instrumentation for measuring aerosol absorption is also necessary,
particularly in assessing the effects of coatings and hygroscopy on the resulting absorption
values. Additional evaluation of the AERONET retrieval algorithm may provide insight into a
potential SSA and, thus, AAOD bias (e.g., Hashimoto et al., 2012). The discrepancies reported
here between in-situ and AERONET values of AAOD and SSA suggest that caution should be
used in upscaling model results to match AERONET retrievals of absorbing aerosol as this will
have a significant impact on global radiative forcing estimates. The work of Wang et al. (2016)
has shown that other factors (e.g., the spatial resolution of models and emissions) may also
contribute to the differences observed between model and AERONET retrievals of AAOD.
Thus, to really be able to understand and simulate the influence of absorbing aerosol on
radiative forcing will require expanded effort on both the measurement and modeling fronts.
4. Conclusion
AERONET retrievals of SSA at low AOD conditions (below the recommended $AOD_{440}<0.4$
constraint) are consistently lower than coincident and co-located in-situ vertical profile
observations of SSA (based on detailed comparisons at two rural sites in the US).
Correspondingly, AERONET retrievals of AAOD at low AOD are consistently higher than those
obtained from in-situ profiles. A survey of the literature suggests that even at higher loading
($AOD_{440}>0.4$) AERONET SSA retrievals tend to be lower than SSA values obtained from vertical
profiling flights, although discrepancies are within the reported uncertainty bounds down to ~
$AOD_{440}>0.3$. The tendency of AERONET SSA to be lower suggests either that AERONET
retrievals over-estimate absorbing aerosol or that the in-situ measurements under-estimate
aerosol absorption. Since the observed discrepancy in SSA can not definitively be attributed to
either technique, the idea of scaling modelled black carbon concentrations upwards to match
AERONET retrievals of AAOD should be approached with caution.  If the AERONET SSA and
AAOD retrievals are indeed biased towards higher absorption, such an upscaling may lead to
aerosol absorption overestimates, particularly in regions of low AOD.  If the discrepancy
between the in-situ and AERONET AAOD is due to issues with the in-situ measurements of
absorption, the only way we see to increase the in-situ absorption values is a significant
enhancement (on the order of a factor of 2 or more) in absorption due to a coating effect.  While
that level of absorption enhancement factor is within the range suggested by modelling studies,
it is significantly higher than many observations of absorption enhancement for ambient aerosol
reported in the literature.
The AERONET retrievals of SSA and AAOD have been used as a primary constraint on global
model simulations of aerosol absorption.  Using only Level-2 retrievals of AAOD (i.e., for
$AOD_{440}>0.4$) on a global scale (e.g., Koch et al., 2009; Bahadur et al., 2010; Chung et al., 2012;
Buchard et al., 2015; Pan et al., 2015; Li et al., 2015) is likely to lead to significant over-
estimates of absorption in cleaner regions although it may be appropriate for conditions of high
loading.  Several different approaches of varying complexity have been developed to better
represent absorbing aerosol for cleaner conditions. Some of these approaches utilize SSA at
high AOD to estimate AAOD at lower AOD conditions (e.g., Bond et al., 2013; Wang et al.,
2014), while others utilize Level-1.5 retrievals with the added uncertainty that entails (e.g.,
Lacagnina et al., 2015; Mallet et al., 2013). Based on the analysis presented here, we cannot
say how to best estimate SSA or AAOD from AERONET retrievals for the low AOD conditions
prevalent around much of the globe.
Some in-situ measurements suggest that a systematic relationship exists between SSA and
AOD, but these measurements are spatially sparse and typically not made at ambient
conditions. Nonetheless, systematic relationships between SSA and AOD, similar to those seen
in the in-situ data at the two sites, are also observed for multiple North American AERONET
sites.   The existence of such a systematic relationship may limit the accuracy of AAOD
estimates when climatological values for SSA from high AOD retrievals are assumed to apply at
low loading conditions.  However, for the two mid-continental rural sites studied here, the
statistically-based monthly medians of SSA from Level-2.0 inversions (i.e., SSA values derived
for $AOD_{440}>0.4$) appear to be quite consistent with monthly SSA values obtained from in-situ
measurements and AEROCOM model simulations.  This suggests that, at these two sites, using
the Level-2.0 inversion SSA to retrieve monthly AAOD at lower AOD conditions (e.g.,
AAOD=AOD*SSA) would not bias the resulting monthly AAOD high, as would occur if only
AAOD values for high AOD cases are included in the AAOD statistics. .  This may not be true
for other locations or averaging times.  Further, for these two sites, a more complex approach to
retrieve monthly AAOD is needed for very clean months when no Level-2.0 inversions are
available.
This study points to several areas where additional research would be useful in resolving the
observed AERONET/in-situ absorption-related discrepancies. First, continued laboratory, field
and modelling efforts are needed to elucidate and unify the current inconsistencies in the
literature on the effects of coatings on absorption enhancement reported for field and lab
measurements and for model simulations. Second, a more extensive evaluation of the
hygroscopicity of ambient (not lab-generated!) absorbing particles would be helpful. Third,
better characterization of how filter-based measurements of absorption respond to coated
particles would be useful, not just in the context of this study, but also for improving our
understanding of the in-situ absorption data acquired by long-term, surface aerosol monitoring
networks (e.g., GAW). Finally, the development of a focused measurement program designed to
acquire statistically robust in-situ vertical profiles over AERONET sites representing a wide
range of conditions and aerosol types could be used to explore the relationships between
retrievals of column properties and variable aerosol profiles and to provide further validation of
the inversion retrieval data products.
**Acknowledgements**
Funding for the SGP airplane measurements was provided by DOE/ARM, while the BND
airplane measurements were funded by NOAA's Climate Program Office. Greenwood Aviation
and FlightSTAR provided fabulous pilots and maintenance of the aircraft. Derek Hageman
(University of Colorado) is owed much for his coding genius and Patrick Sheridan
(NOAA/ESRL/GMD) for making instruments on both airplanes work and ensuring the quality of
the BND aircraft data set. We thank the modelling groups for providing AeroCom Phase II
results. We greatly appreciate the ease of access to and use of IMPROVE aerosol chemistry
data for the Bondville and Cherokee Nation IMPROVE sites. IMPROVE data were downloaded
from: http://views.cira.colostate.edu/web/DataWizard/. This study was supported by NOAA
Climate Program Office's Atmospheric Chemistry, Carbon Cycle and Climate (AC4) program.
BHS acknowledges funding by the Research Council of Norway through the grants AC/BC
(240372) and NetBC (244141). Last, but not least, we also gratefully acknowledge the very
helpful discussions from David Giles and Brent Holben from AERONET and the very useful and
extensive comments from our reviewers.

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

**Tables**
**Table 1a** Statistical values (medians, means and standard deviations) of AERONET versus in-
situ comparison where there was an AERONET retrieval within +/-3 h of the end of a 2 h flight
profile. AERONET values are for Level-1.5 data when there was a Level-2 AOD value and an
almucantar retrieval. (First value in each cell is median; second set of values in each cell are
mean± Std.Dev, third row is number of AERONET retrievals corresponding to flights (in
AERONET columns) or number of flights (In-situ columns)).  These numbers represent the blue
points in Figures 4-6.

| | BND | | SGP | |
|---|---|---|---|---|
| | AERONET | In-situ | AERONET | In-situ |
| AOD | 0.118; 0.146±0.099 | 0.114; 0.135±0.139 | 0.138; 0.146±0.099 | 0.137; 0.147±0.077 |
| AAOD | 0.013; 0.013±0.007 | 0.003; 0.005±0.006 | 0.019; 0.023±0.008 | 0.004; 0.004±0.003 |
| SSA | 0.895; 0.898±0.034 | 0.961; 0.964±0.020 | 0.847; 0.839±0.038 | 0.971; 0.973±0.011 |
| # | 51 retrievals[1] | 21 flights | 23 retrievals[1] | 11 flights |

[1]retrievals are flight-averaged prior to calculating statistics.

**Table 1b** Statistical values (medians, means and standard deviations) of AERONET versus in-
situ comparison where there was an AERONET retrieval within +/-3 h of the end of a 2 h flight
profile and AERONET $AOD_{440}>0.2$. AERONET values are for Level-1.5 data when there was a
Level-2 AOD value and an almucantar retrieval. (First value in each cell is median; second set
of values in each cell are mean± Std.Dev, third row is number of AERONET retrievals
corresponding to flights (in AERONET columns) or number of flights (In-situ columns)).  These
numbers represent the purple points in Figures 5-6.

| | BND | | SGP | |
|---|---|---|---|---|
| | AERONET | In-situ | AERONET | In-situ |
| AOD | 0.306; 0.304±0.125 | 0.299; 0.331±0.230 | 0.269 | 0.238 |
| AAOD | 0.025; 0.019±0.012 | 0.010; 0.013±0.012 | 0.034 | 0.009 |
| SSA | 0.941; 0.942±0.023 | 0.971; 0.966±0.010 | 0.875 | 0.964 |
| # | 6 retrievals[1] | 4 flights | 2 retrievals | 1 flights |

[1]retrievals are flight-averaged prior to calculating statistics.


**Table 2** Number of AERONET/IN-SITU AOD and AAOD flight matches as a function of various
AERONET constraints and the +/- 3h time window.

| | BND (2006-2009) | SGP(2005-2007) |
|---|---|---|
| Total profile flights | 402 | 171 |
| Level-2 AOD | 73 | 37 |
| Level-2 AOD+almucantar retrieval[1] | 21 | 11 |
| Level-2 AOD+almucantar retrieval+$AOD_{440}$>0.20 | 2 | 1 |
| Level-1.5* AAOD | 21 | 11 |
| Level-1.5* AAOD + $AOD_{440}$>0.20 | 4 | 1 |
| Level-2 AAOD | 1 | 0 |

[1]an almucantar retrieval does not necessarily imply an AAOD retrieval

Table 3 Direct AAOD comparisons – AERONET ("RS") vs In-Situ ("IS")

| Study, # profiles Citation(s) | Location, aerosol type AOD comments Alt. range | Instruments corrections size cut | AAOD comparison information | Comments |
|---|---|---|---|---|
| BND 24 profiles<br><br>This study | Central US Rural, continental<br><br>$AOD_{440}$ range: 0.04-0.55<br><br>AOD comparison See Fig. 4a<br><br>150-4200 m agl | PSAP-3wave TSI neph-3wave<br><br>B1999, O2010, AO1998 f(RH) adjust<br><br>Dp<5-7 $\mu$m | Wavelength=440, 670 nm Ångström interpolation<br><br>RS AAOD>IS AAOD<br><br>$AAOD_{440}$ range: 0.001-0.042 | Profiles matched within 3 hours of AERONET measurement. Profiles within 15 km of AERONET measurement.<br><br>Used V2 AERONET Level 1.5 AAOD values for cases with valid V2 AERONET Level 2.0 AOD value.<br><br>Extrapolated from lowest altitude range to ground to account for aerosol below plane |
| SGP 14 profiles<br><br>This study | Central US Rural, continental<br><br>$AOD_{440}$ range: 0.06-0.43<br><br>AOD comparison See Fig. 4b<br><br>150-4200 m agl | PSAP-3wave TSI neph-3wave<br><br>B1999, O2010, AO1998 f(RH) adjust<br><br>Dp<5-7 $\mu$m | Wavelength=440, 670 nm Ångström interpolation<br><br>RS AAOD>IS AAOD<br><br>$AAOD_{440}$ range: 0.012-0.052 | Profiles matched within 3 hours of AERONET measurement. Profiles within 1 km of AERONET measurement<br><br>Used V2 AERONET Level 1.5 AAOD values for cases with valid V2 AERONET Level 2.0 AOD value.<br><br>Extrapolated from lowest altitude range to ground to account for aerosol below plane |
| MAC 13 profiles<br><br>Corrigan et al., 2008 | Indian Ocean Pollution<br><br>$AOD_{440}$ range: 0.1-0.6 | Aethalometer 3-wave OPC +Mie for scattering<br><br>A2005 | Wavelength not provided<br><br>RS AAOD>IS AAOD<br><br>$AAOD_{440}$ range: 0.005-0.033 | No details on how profiles matched with retrievals in terms of time or distance. No details on version of AERONET data used; this is relevant, given low AODs in first half of study – not sure if there were comparisons for low AODs. |

| | | | | |
|---|---|---|---|---|
| | No AOD comparison<br><br>0-3200 m asl | Dp<5 µm | | Note: this study is the one cited by Bond et al. (2013) to support the use of AERONET to scale modeled BC values |

In-situ instrument corrections: B1999=Bond et al., 1999; O2010=Ogren, 2010, AO1998=Anderson and Ogren, 1998; A2005=Arnott et al., 2005; Ångström interpolation – indicates in-situ wavelength adjusted to AERONET wavelength using Ångström interpolation; f(RH) adjust – indicates the in-situ measurements were adjusted to ambient humidity conditions for the AOD comparison.  IS=In-situ measurements, RS=Remote sensing (AERONET) measurements

Table 4  SSA comparisons – AERONET vs In-situ

| Study,<br># profiles<br>Citation(s) | Location, aerosol type<br>AOD comments<br>Alt. range | Instruments,<br>Corrections,<br>Inlet size cut | SSA comparison information | Comments |
|---|---|---|---|---|
| BND<br>24 profiles<br><br>This study | Central US<br>Rural, continental<br><br>$AOD_{440}$ range: 0.04-0.55<br><br>AOD comparison: See Fig. 4a<br><br>150-4200 m agl | PSAP-3wave<br>TSI neph-3wave<br><br>B1999, O2010, AO1998<br>f(RH) adjust<br><br>Dp<5-7 µm | Wavelength=440, 670 nm<br>Ångström interpolation<br><br>RS SSA<IS SSA | Profiles matched within 3 hours of AERONET measurement.  Profiles within 15 km of AERONET measurement.<br><br>Used V2 AERONET Level 1.5 AAOD values for cases with valid V2 AERONET Level 2.0 AOD value.<br><br>Extrapolated from lowest altitude range to ground to account for aerosol below plane |
| SGP<br>14 profiles<br><br>This study | Central US<br>Rural, continental<br><br>$AOD_{440}$ range: 0.06-0.43<br><br>AOD comparison: | PSAP-3wave<br>TSI neph-3wave<br><br>B1999, O2010, AO1998<br>f(RH) adjust | Wavelength=440, 670 nm<br>Ångström interpolation<br><br>RS SSA<IS SSA | Profiles matched within 3 hours of AERONET measurement.  Profiles within 1 km of AERONET measurement.<br><br>Used V2 AERONET Level 1.5 AAOD values for cases with valid V2 AERONET Level 2.0 AOD value. |

| | | | | |
|---|---|---|---|---|
| | See Fig. 4b<br><br>150-4200 m agl | Dp<5-7 μm | | Extrapolated from lowest altitude range to ground to account for aerosol below plane |
| AAO (BND)<br>1 profile<br><br>Esteve et al., 2012 | Central US<br>Rural, continental<br><br>AOD$_{550}$ = 0.65<br><br>AOD comparison:<br>RS AOD>IS AOD<br><br>150-4200 m agll | PSAP-3wave<br>TSI neph-3wave<br><br>B1999, O2010, AO1998<br>f(RH) adjust<br><br>Dp<5-7 μm | Wavelength=550 nm<br>Power law interpolation<br><br>RS SSA < IS SSA | Profiles matched within 2 hours of AERONET measurement. Profiles within 15 km of AERONET measurement.<br><br>Used V2 AERONET Level 2.0 AOD value.<br><br>Extrapolated from lowest altitude range to ground to account for aerosol below plane |
| DISCOVER-AQ<br>12 profiles<br><br>Schafer et al., 2014 | East Coast US<br>Polluted air<br><br>AOD$_{440}$>0.2<br><br>AOD compare:<br>RS AOD > IS AOD<br>(by 23%)[*]<br><br>367-3339 m | PSAP-3wave<br>TSI neph-3wave<br><br>V2010, AO1998[*]<br>f(RH) adjust<br><br>Dp<4 μm[*] | Wavelength=550 nm<br>AERONET "interpolated" to 550 (no detail provided)<br>In-situ absorption interpolated to 550 using Ångström interpolation<br><br>RS SSA < IS SSA | Profile matched within 45 min of AERONET measurement. Profile within 1 km of AERONET measurement.<br><br>Used V2 AERONET Level 2.0 values in paper<br><br>Altitude range: at least <500 m and >1500 m for column comparisons, min and max altitudes: 367 m and 3339 m<br>Did not specify agl or asl but those are similar for the location. |
| CLAMS<br>1 profile<br><br>Magi et al., 2005 | East Coast US<br>Polluted air<br><br>AOD$_{440}$=0.60<br><br>AOD comparison:<br>RS AOD > IS AOD<br>(by 15%) | PSAP-1wave<br>MSE neph-3wave<br><br>B1999, AO1998<br>f(RH) adjust<br><br>Inlet size cut not reported, Sinha, | Wavelength=550 nm<br>Wave_adj =quadratic polynomial interpolation<br><br>RS SSA < IS SSA | Profile matched within 1 hour of AERONET measurement. Profile within 3 km of AERONET measurement.<br><br>Retrieved V2 AERONET Level 2.0 AOD$_{440}$ from http://aeronet.gsfc.nasa.gov/<br><br>Also compared campaign AERONET average |

| | | | | |
|---|---|---|---|---|
| | 170-1500 m agl | 2003 suggests Dp<4 μm | | with profile average: SSA's much closer, but profiles weren't necessarily close in time or space to AERONET site |
| ESCOMPTE 1 profile Mallet et al., 2005 | Avignon, France Pollution AOD$_{440}$>0.55 No AOD comparison 100-2900 m | PSAP-1wave TSI neph-3wave B1999, A1999 No f(RH) adj Inlet Dp not given | Wavelength=550 nm Wave_adj = estimated from visual inspection (spectral dependence is relatively flat) RS SSA < IS SSA | Profile matched within 1 hour of AERONET measurement. Profile within 10 km of AERONET measurement. Used V2 AERONET Level 2.0 AOD$_{440}$ from http://aeronet.gsfc.nasa.gov/, not stated in paper. Did not adjust in-situ measurements for f(RH), so presumably IS SSA would increase so it was even larger than RS SSA. Did not specify agl or asl |
| SAFARI 5 profiles Leahy et al., 2007 UW plane | Southern Africa Biomass burning AOD$_{550}$>0.28-1.12 AOD comparison: RS AOD > IS AOD RS=1.12*IS-0.05 R$^2$=0.99 100-5320 m asl | PSAP-1wave MSE neph-3wave B1999;H2000 f(RH) adjust Dp<4 um | Wavelength=550 nm Wave_adj= 2nd order polynomial For AOD$_{550}$>0.6 (3 profiles) RS SSA > IS SSA For AOD$_{550}$<0.3 (2 profiles) RS SSA $\leq$ IS SSA | Profiles matched within 1-4 hours of AERONET measurement. Profiles within 20 km of AERONET measurement. Used V2 AERONETLevel 2.0 Also found: AEROCOM model>insitu Altitude range is min and max over 5 flights – no flights covered that entire range). They used AATS to account for aerosol above plane and extrapolated down to acct for aerosol below plane. (Altitude range from flight info in Magi et al., 2003) |

| | | | | |
|---|---|---|---|---|
| SAFARI<br>1 profile<br><br>Haywood et al, 2003<br>C-130 | Southern Africa<br>Biomass burning<br><br>$AOD_{440}$=0.71<br><br>AOD comparison:<br>RS AOD < IS AOD<br><br>330-3420 m agl | PSAP-1wave<br>TSI neph-3wave<br><br>B1999,AO1998<br>No f(RH) adj<br><br>Dp<2-4 $\mu$m | Wavelength=native<br>Wave_adj = none<br><br>RS SSA < IS SSA | Profile matched within 2 hours of AERONET measurement.  Profiles within 10 km of AERONET measurement.<br>Used V2 AERONET Level 2.0 data from http://aeronet.gsfc.nasa.gov/<br><br>They defend the lack of f(RH) correction because (a) ambient RH values < 56% and (b) previous measurements of f(RH) of BB aerosol suggest minimal hygroscopicity<br><br>Paper mostly focused on size dist comparison; SSA comparison seems like afterthought.<br><br>Extrapolated from lowest altitude range to ground to account for aerosol below plane |
| DABEX<br>3 profiles<br><br>Osborne et al., 2008 | Africa<br>Dust/BB<br><br>AOD comparison<br>RS AOD < IS AOD<br>(by up to 40%)<br><br>$AOD_{550}$~0.3-0.6<br><br>100-5000 m | PSAP-1wave<br>TSI neph-3wave<br><br>B1999,AO1998<br>No f(RH) adj<br><br>Dp<2-4 $\mu$m | Wavelength=550 nm<br>Wave_adj=log interpolation<br><br>RS SSA < IS SSA | No details on how profiles matched with retrievals in terms of time.   Profiles within 100 km of AERONET measurement<br><br>Used V2 AERONET Level 2.0<br><br>They defend the lack of f(RH) correction because (a) ambient RH values are mostly low (<60%) and (b) previous measurements of f(RH) of BB aerosol suggest minimal hygroscopicity<br><br>Jan 21, 23 and 30 profiles<br>IS overpredicts AOD so IS SSA is greater |

| | | | | than RS SSA |
| --- | --- | --- | --- | --- |
| | | | | Suggest it could be due to large particle correction to IS measurements using PCASP. McConnell et al., (2008) suggests problems with nephelometer sensitivity |
| | | | | Did not specify agl or asl |
| | | | | Altitude range is min and max over 4 flights – no flights covered that entire range |
| DABEX<br>1 profile<br><br>Johnson et al., 2009 | Africa<br>DUST/BB<br><br>AOD comparison:<br>RS AOD < IS AOD (by ~10%)<br><br>$AOD_{550} > 0.7$<br><br>150-3000 m | PSAP-1wave<br>TSI neph-3wave<br><br>B1999,AO1998<br>No f(RH) adj<br><br>Dp<2-4 $\mu$m | Wavelength=550 nm<br>Wave_adj=log interpolation<br><br>RS SSA < IS SSA | Profile matched within 1 hour of AERONET measurement.  Profile within 100 km of AERONET measurement<br><br>Used V2 AERONET Level 2.0<br><br>They defend the lack of f(RH) correction because ambient RH values are mostly low (<40% with a max of 70%)<br><br>Jan 19 profile<br><br>Incorrectly used Mie to adjust $\sigma_{ap}$ to 550 after B1999 applied<br><br>Did not specify agl or asl |

IS=In-situ measurements, RS=Remote sensing (AERONET) measurements.  In-situ instrument corrections: B1999=Bond et al., 1999; V2010=Virkula et al., 2010;O2010=Ogren, 2010; AO1998=Anderson and Ogren, 1998; H2000=Hartley et al., 2000; A2005=Arnott et al., 2005; Ångström interpolation – indicates wavelength adjustment using Ångström exponent interpolation; f(RH) adjust – indicates the in-situ measurements were adjusted to ambient humidity conditions for the AOD and SSA comparison.
[*]Information about Discover-AQ flights from Crumreyrolle et al. (2014)

**Figures**

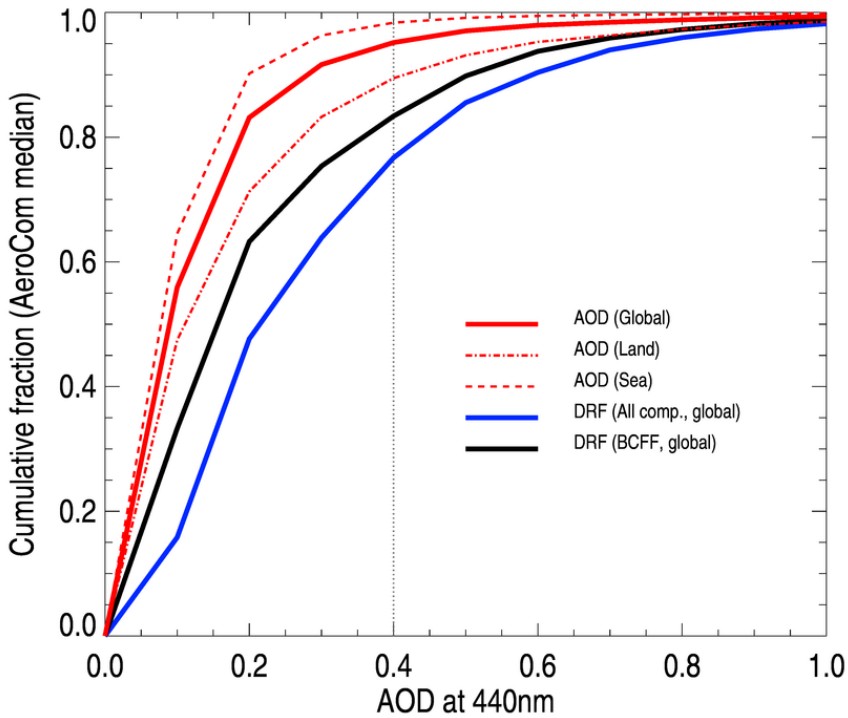

Figure 1. Cumulative AOD$_{440}$ frequency distribution (red lines) based on output from five AeroCom models. Blue and black lines show contribution of total aerosol and fossil fuel black carbon, respectively, to the global radiation budget as a function of AOD$_{440}$. See text for details. Models used to generate the AOD lines include: GMI-MERRA-v3, GOCART-v4, LMDZ-INCA, OsloCTM2, and SPRINTARS-v385. Models used to generate the radiative forcing lines include all but the GMI-MERRA-v3 model. Model information and references can be found in Myhre et al. (2013).

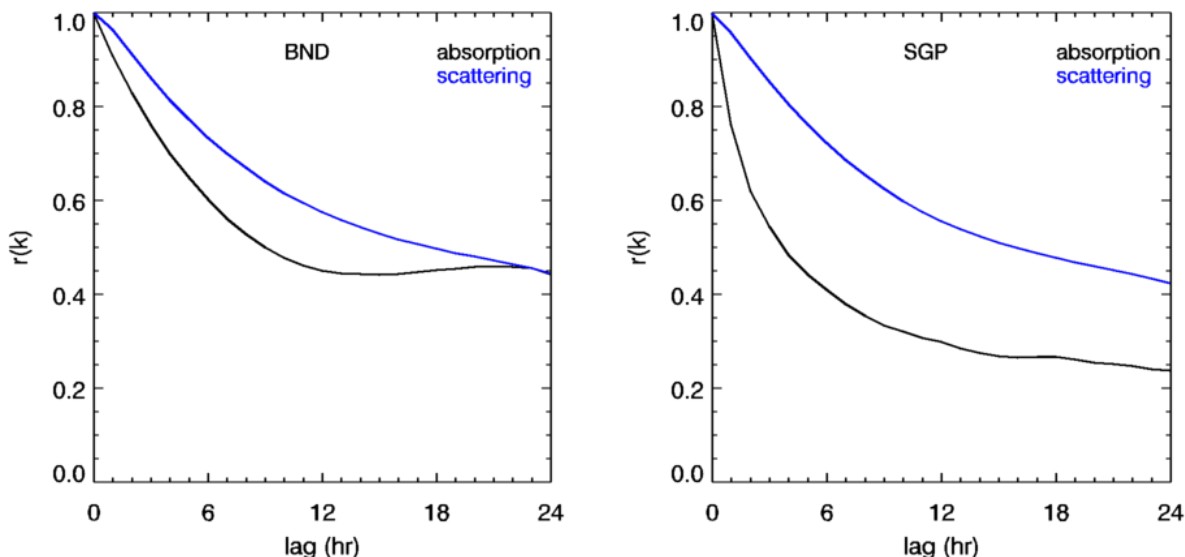

Figure 2. Correlograms for BND and SGP; wavelength = 550 nm, $D_p$<10 $\mu$m, based on hourly averaged surface in-situ data between 1995-2013 (BND) and 1996-2013 (SGP). The value r(k) on the y-axis represents the autocorrelation at lag time 'k'.

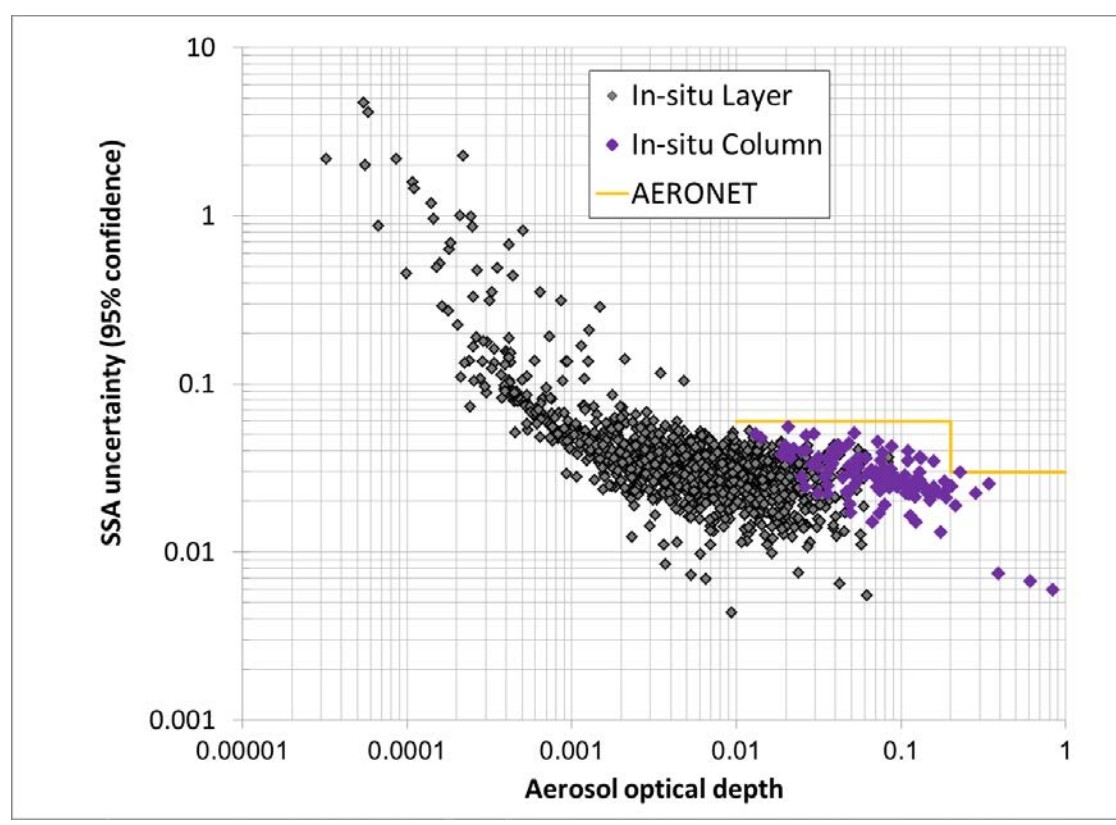

Figure 3. SSA uncertainty as a function of aerosol optical depth.  Black points represent the SSA uncertainty (95% confidence) in each flight layer (all three visible wavelengths measured by the PSAP and nephelometer) as a function of the AOD of that flight layer. Purple points represent the in-situ column SSA uncertainty for each flight (again points include all three wavelengths measured by the PSAP and nephelometer) as a function of the in-situ AOD. Orange line represents the uncertainty in SSA reported by Dubovik et al. (2000, their table 2) for two AOD ranges.

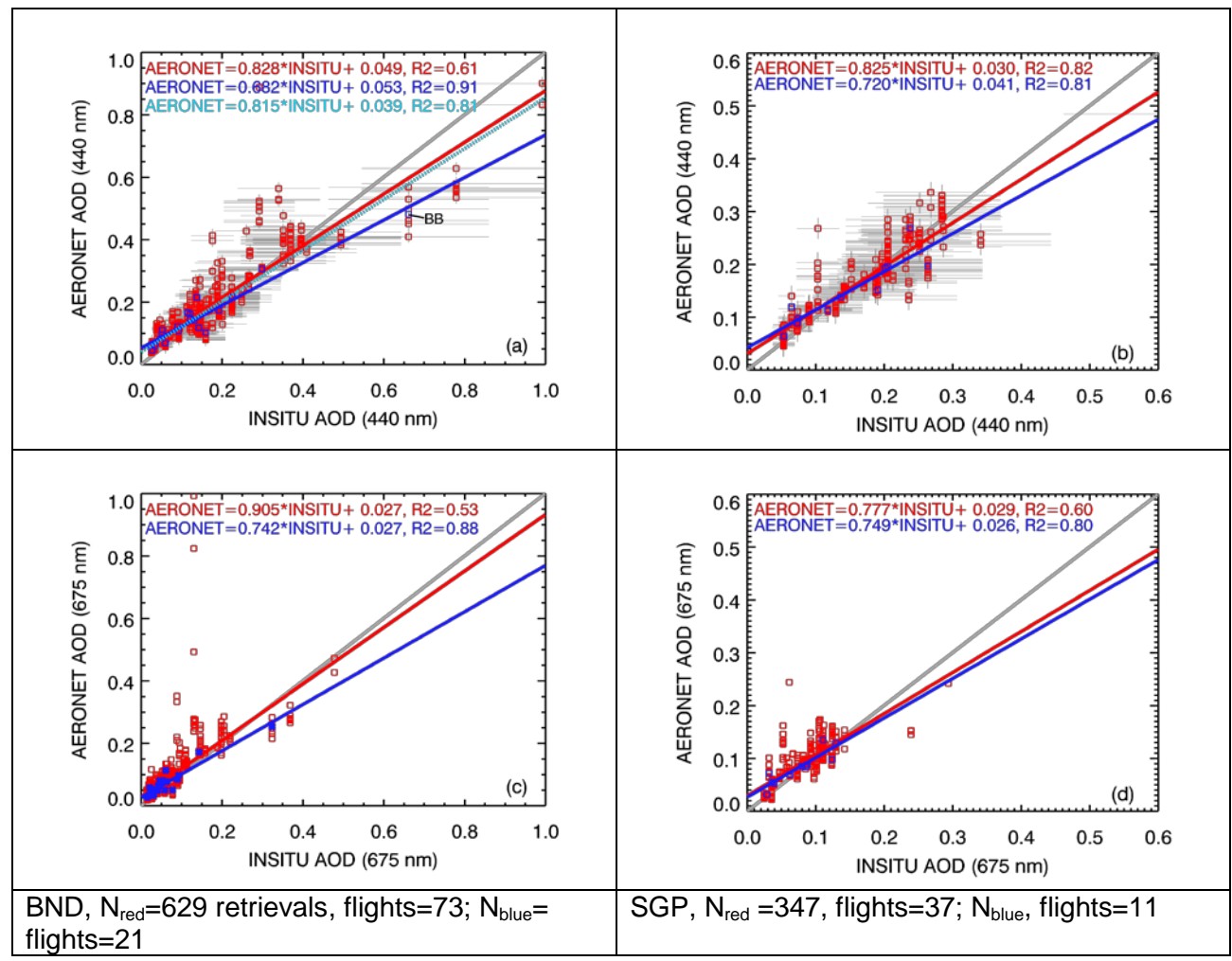

| | |
|---|---|
| BND, N$_{red}$=629 retrievals, flights=73; N$_{blue}$= flights=21 | SGP, N$_{red}$ =347, flights=37; N$_{blue}$, flights=11 |

Figure 4. AOD comparison  (a) BND at 440 nm;  (b) SGP at 440 nm; (c) BND at 675 nm; and (d) SGP at 675 nm; thick gray line is 1 to 1 line. Thin gray lines associated with each data point represent measurement uncertainties.  Red points and fit line represent all AERONET direct sun Level-2 AOD measurements within +/-3 hours of end of profile.  Blue points and fit line represent the average of AERONET Level-2 AOD measurements with successful almucantar retrievals within +/-3 hours of end of profile.  The light blue dashed line is the fit if the BB point is excluded.  Note: two BND direct sun AOD440 points corresponding to the two highest AOD675 points in the figure below are off the scale of the plot and not shown.  The third high AOD440 point is partly obscured by the legend.

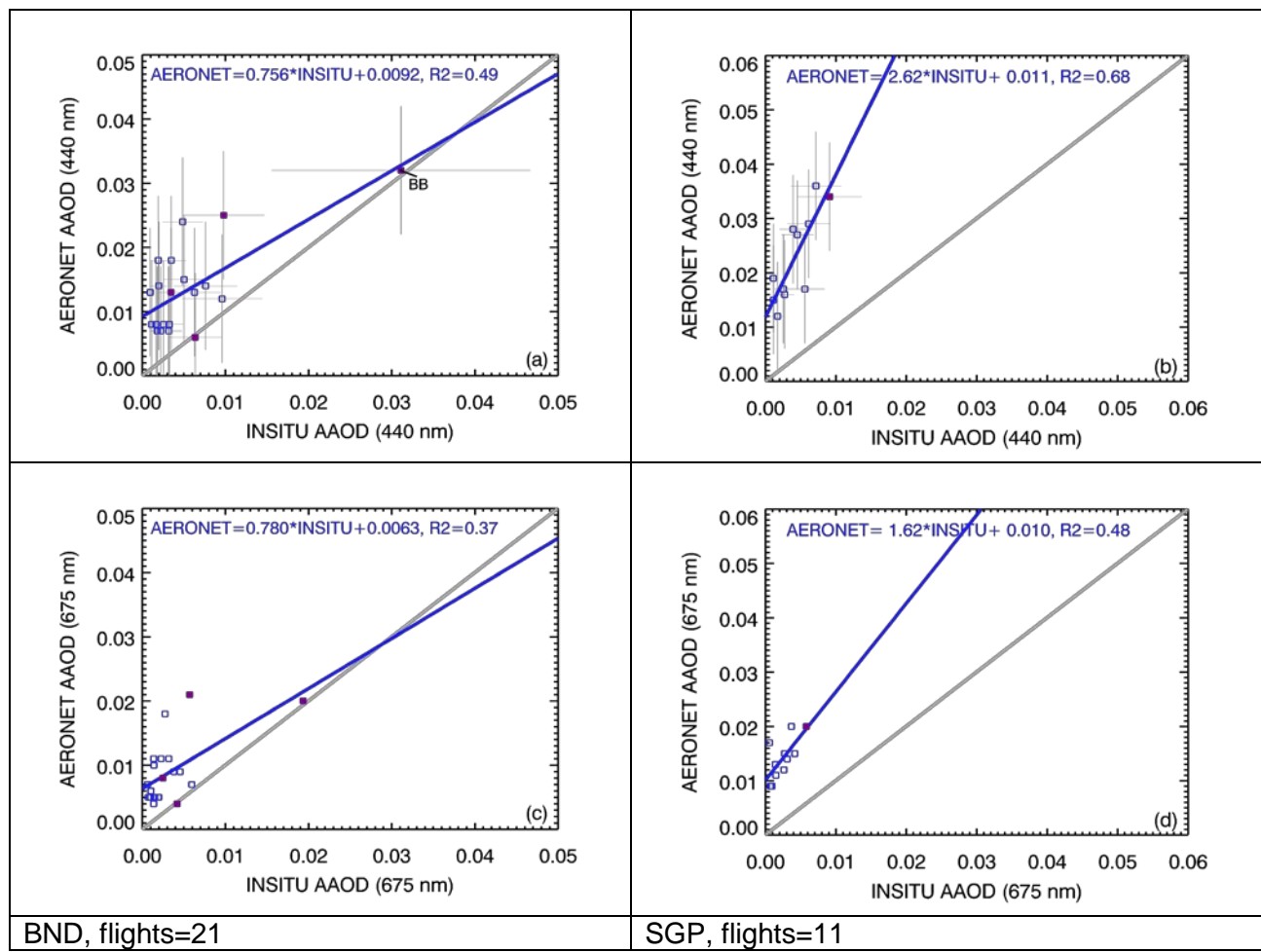

Figure 5. AAOD comparison, (a) BND at 440 nm; (b) SGP at 440 nm ; (c) BND at 675 nm; and (d) SGP at 675 nm. Blue line is linear fit for all points shown; gray line is 1 to 1 line. Thin gray lines associated with each data point represent measurement uncertainties.    Points show the average of AERONET Level-1.5 AAOD retrievals for which there was a successful AERONET Level-2 almucantar retrieval within +/-3 hours of end of profile.  Purple points indicate the few comparisons points for which there are AERONET  Level-2 almucantar retrievals and where the average AERONET $AOD_{440}$ for those retrievals was great than 0.2.

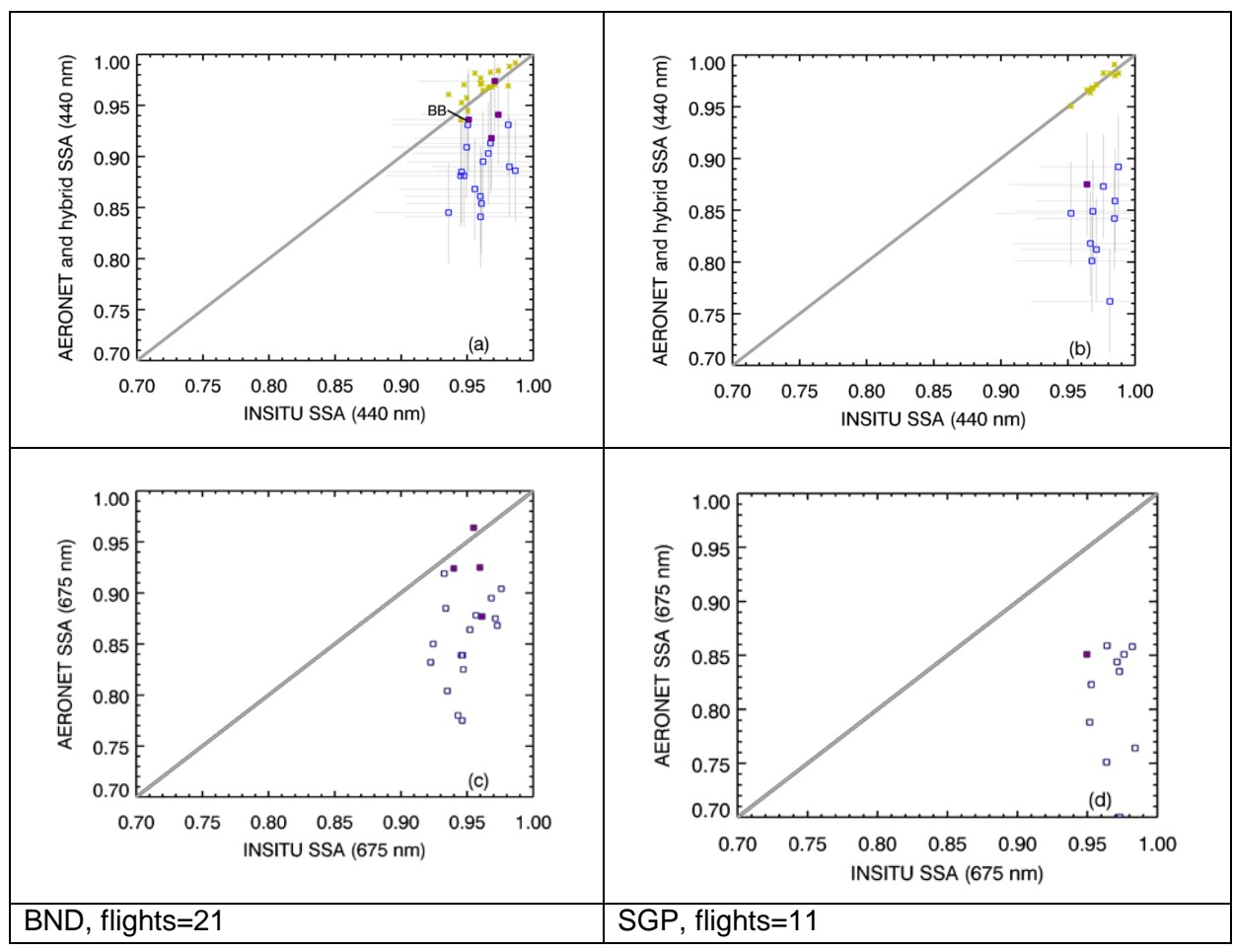

Figure 6. SSA comparison, (a) BND at 440 nm (b) SGP at 440 nm; (c) BND at 675 nm; and (d) SGP at 675 nm. Blue line is linear fit for all points shown; gray line is 1 to 1 line. Thin gray lines associated with each data point represent measurement uncertainties. Blue points show the average of all AERONET Level-1.5 AAOD retrievals for which there was a successful AERONET Level-2 almucantar retrieval within +/-3 hours of end of profile. Purple points indicate the few points for which there are AERONET Level-2 almucantar retrievals and where where the average AERONET $AOD_{440}$ for those retrievals was great than 0.2. The yellow points represent the 'hybrid SSA' which utilizes the AERONET AOD and the in-situ AAOD to derive SSA as described in the text. The in-situ uncertainty lines here represent SSA uncertainty of 0.06 which is the worst case uncertainty.

.

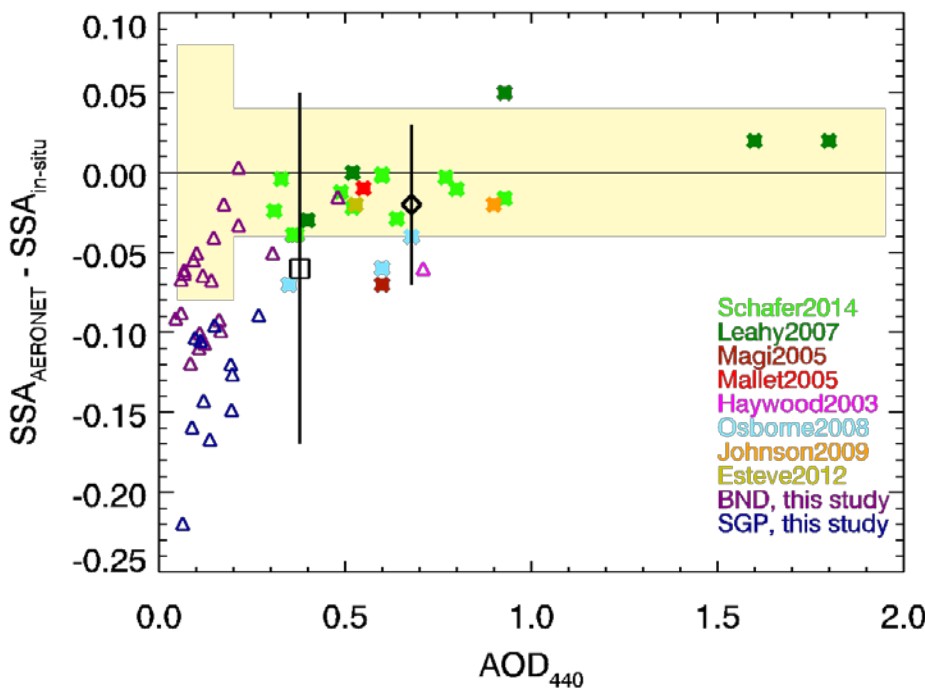

Figure 7. AOD$_{440}$ vs [SSA$_{AERONET}$-SSA$_{INSITU}$] for direct comparisons studies listed in Table 4. Open symbols are for SSA$_{440}$ difference; filled symbols are for SSA$_{550}$ difference. AOD$_{440}$ values for Leahy2007, Osborne2008, Johnson2009 use the Level-2 values reported on the AERONET webpage for the locations and dates of the specific profile.  Shading indicates the combined standard uncertainty of AERONET SSA values as function of AOD as reported in Table 4 of Dubovik et al. (2000) and uncertainty in the in-situ SSA based on Figure 2.  The black square and black diamond with vertical black lines represent, respectively, the mean and 2*standard deviation for all direct comparisons (including BND and SGP) and for literature direct comparisons only.

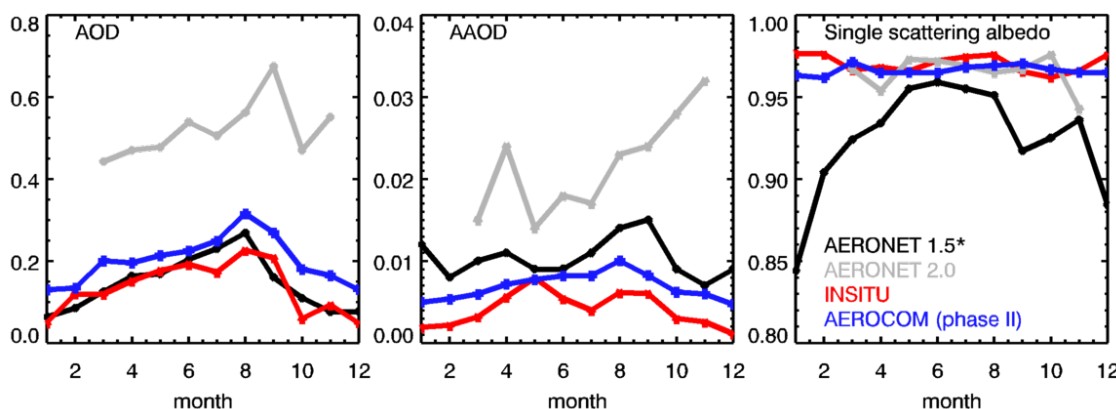

Figure 8a. Monthly medians of BND aerosol optical properties at 440 nm. AERONET medians are for 1996-2011. AERONET AOD medians are for observations with Level-2 almucantar retrievals, with corresponding AAOD and SSA retrievals at Level-1.5 (black) or Level-2 (gray). In-situ data are for June 2006-September 2009. AERONET Level-2.0 almucantar AOD and AAOD values are biased high by definition, because of the $AOD_{440}>0.4$ constraint. AERONET 2.0 direct sun retrievals (not shown) are similar to the AERONET 1.5 AOD values. In-situ values are derived from 365 flights over BND. AeroCom Phase II median model results cover various time periods (depending on the model) and are reported at 550 nm.

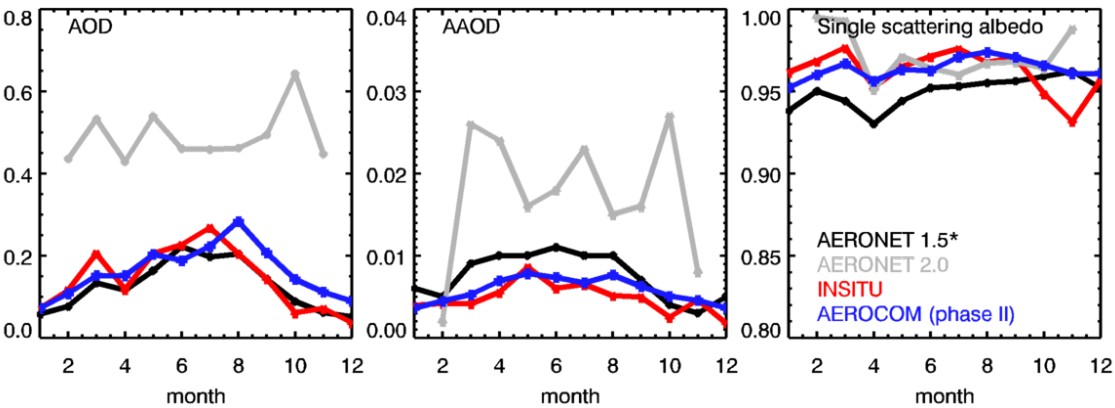

Figure 8b. Monthly medians of SGP aerosol optical properties at 440 nm. AERONET medians are for 1996-2011. In-situ data are for September 2005-December 2007. AERONET AOD medians are for observations with Level-2 almucantar retrievals, with corresponding AAOD and SSA retrievals at Level-1.5 (black) or Level-2 (gray). AERONET Level-2.0 almucantar AOD and AAOD values are biased high by definition, because of the $AOD_{440}>0.4$ constraint. AERONET 2.0 direct sun retrievals (not shown) are similar to the AERONET 1.5 AOD values. In-situ values are derived from 322 flights over SGP. AeroCom Phase II median model results cover various time periods (depending on the model) and are reported at 550 nm.

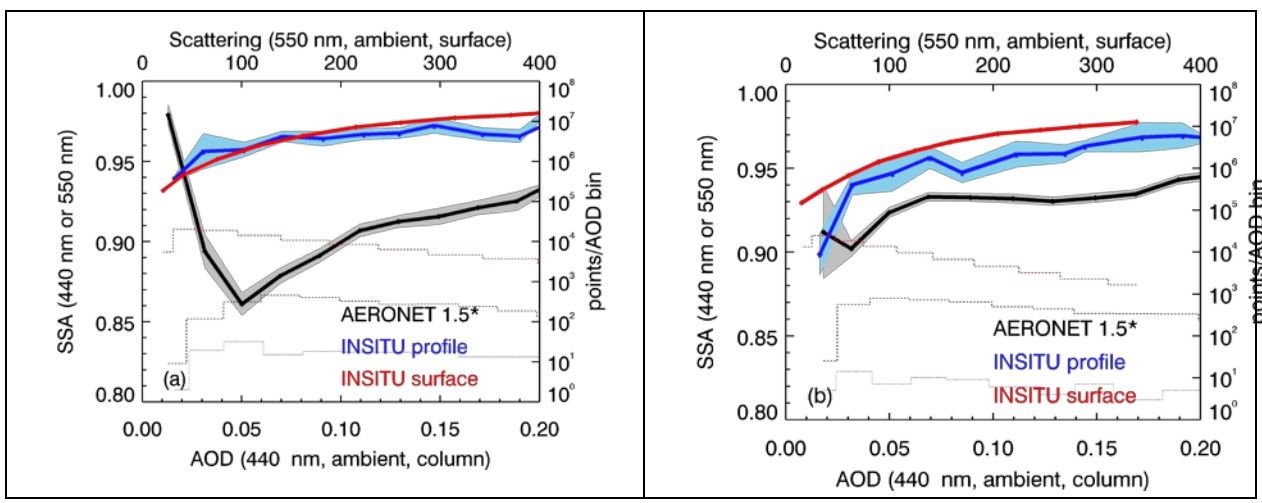

Figure 9. Systematic variability of SSA as a function of loading for (a) BND and (b) SGP for AERONET 1.5* AOD and SSA (black lines), AOD and SSA from in-situ profiles (blue lines) and in-situ scattering and SSA from surface measurements (red lines). Solid lines indicate mean values of SSA and AOD for each 0.05 AOD bin (10 Mm$^{-1}$ scattering bin). Shaded areas represent mean standard error (mean standard error for surface data is within thickness of red line).  Histograms indicate the number of points in each AOD (or scattering) bin.  Plot based on BND and SGP AERONET data (date range: 1996-2012) and BND INSITU profile data (date range: 2006-2012); SGP INSITU profile data (date range: 2006-2007).  Surface data (orange lines) are for 550 nm, low RH, hourly in-situ data from the surface sites at BND (date range: 1997-2013) and SGP (date range: 1998-2013). AERONET 1.5* is from Level-1.5 retrievals with a corresponding Level-2 almucantar retrieval.