# Peer review of "Comparison of AOD, AAOD and column single scattering albedo from AERONET"

_Atmospheric Chemistry and Physics, 2016_

## Referee Comment (RC1) · Anonymous Referee #2 · 20 Oct 2016

This paper deals with an important and challenging issue, certainly acceptable for ACPD. The authors argue that the aerosol absorption data most widely used in climate modeling is likely biased high at low AOD, based on coincident and climatological in situ data at two rural sites. I've included some notes below; in summary, there is a lot of good data presented here, but I think the estimates of uncertainty need to be tightened up in order to reach a strong conclusion. Also, evaluating AERONET SSA at AOD below the value they state as the lower limit of quality results is a key caveat, though I agree that the AERONET results are widely applied beyond their stated validity range. Note that this is actually my full review, so it can be considered as part of the formal review process rather than just as a "quick" review for ACPD posting.

Lines 73-79. This essentially makes the case for selection bias in the AERONET SSA and AAOD values by itself, though I don't think it negates the value of going further and

comparing with in situ observations. If the in situ data can show that in general, SSA is lower when AOD is higher, that could make a useful contribution to the argument.

Line 112. This is supported by the AERONET data themselves. AERONET does not offer global spatial coverage, but it does provide overwhelming evidence AOD_440 is generally <0.4 via direct-sun AOD measurements, which don't suffer from the uncertainties entailed in the model estimates.

Lines 242 to 258. As you know, in addition to collocation, the big challenges for this study are probably getting the total column data from the aircraft sampling right, and accounting for the difference between the properties of the ambient particle observed by AERONET and the dessicated ones measured in situ. Assuming that absorbing aerosol is hygrophobic seems a bit risky, especially for an SSA calculation, though this would be less of an issue for cases where the ambient RH is also low. (Do Lines 332-334 raise another question about getting SSA right?)

Ok. I see that you deal with these issues in Section 2.4.1. I'm thinking that the hygroscopicity issue might need a bit more consideration; there does not seem to be a conclusion about the uncertainty in SSA from the in situ observations, and it is not clear whether the general discussion derived from the literature is applicable to the aerosols observed over the AERONET sites in the current study. For the column AOD question, again the discussion does not seem to come to a real conclusion about the uncertainties. Having coincident lidar would help, and this might be available for at least some cases at one or both sites.

Section 2.4.2. There are other possible factors to consider here. For example, the AERONET retrievals report only one pair of (real, imaginary) refractive index values. If there are two or more modes in the column, this assumption will skew the result. You mention the possible surface reflectance contribution to the AERONET AOD uncertainty; there is a paper assessing this which might be worth considering (Sinyuk et al., Remote Sensing Environment 2007, doi:10.1016/j.rse.2006.07.022). Then there is

a question about whether the direct sun AOD measurements are used to obtain the extinction in the determination of aerosol absorption properties, or whether the scattering and extinction are both determined from the almucantar scan. In the latter case, the measurement uncertainty will be larger than 0.01 or 0.02, whereas in the former case, heterogeneity could affect the result, as the extinction and scattering data would be taken in different parts of the sky. Either way, the SSA result in most cases would be the small difference between two larger numbers, so accuracy could be an issue.

Lines 503-505. Perhaps the AOD comparisons address the total-column sampling question for the aircraft measurements, in addition to the uncertainty related to the hygroscopicity adjustment and possible large-particle under-sampling. Note that in general, a high correlation does not assure good quantitative values, as might be required for SSA assessment. So, quantitatively, how does this affect the uncertainty in subsequent SSA estimation?

Line 561. Again, it is not clear how much the measurement uncertainty contributes to the discrepancies between in situ and AERONET AAOD.

Line 567-568. Does this mean the in situ measurements are missing the extremes, either due to sampling, or to perhaps to conservative estimates of the hygroscopicity effect?

Lines 594-596. Right. But this does not address whether the underlying assumption that absorbing particles are non-hygroscopic is valid. If the absorbing species are OC rather than entirely BC, one might expect at least some hygroscipic growth is possible. And I think you concluded earlier that there must be something like OC, at least at one site.

Lines 614 to 617. Does this call into question whether the in situ measurements adequately sample the entire column observed by AERONET? I'm thinking Section 3.1.2 does not put to rest the question in the title of this section. So I'm uncertain whether you have established the conclusion stated in Lines 625-627, though I think AERONET

might overestimate absorption in many cases, due to the way they relate the measured extinction and scattering in order to derive absorption.

Lines 689-690. This might be stated differently, as it assumes no systematic underestimation of absorption for the in situ measurements.

---

## Short Comment (SC1) · 21 Oct 2016

A very useful paper, and it will be interesting to see how to reconcile various measurements.

Here I just want to mention the issue of temporal sampling, see also: http://www.atmos-chem-phys.net/16/1065/2016/

While this temporal sampling issue is important for model evaluation, it is equally important in comparing different observational datasets. I see two issues relevant to the current paper:

Page 18, Line 732-735: Fig 7 was apparently made with different samplings of the in-situ and AERONET measurements. Is that the case for other figures as well? How

might that affect results?

Page 22, Line 925 - 934: The authors suggest better estimates of AAOD may be obtained by using SSA measured at high AOD and applying it to low AOD cases. They mention possible sampling impacts but seem to feel those may not be that important. I'd like to caution against that.

I attach a figure of the difference in yearly SSA, when that SSA is taken at high AOD or at any AOD, for three different models. At least two models allow differences of more than 0.05. (In general, the MIROC-SPRINTARS model agrees best with AERONET Lev 2 SSA while HadGEM-UKCA is often too high and ECHAM-HAM too low.)

Finally, it would be useful if the authors made a suggestion under what conditions AERONET SSA conditions may be used. Is AOD > 0.4 sufficient?

regards, Nick
* * *
[Figure]

**Fig. 1.**

---

## Referee Comment (RC2) · Anonymous Referee #3 · 15 Nov 2016

ACPD Review:

Is there a bias in AERONET retrievals of aerosol light absorption at low AOD conditions?

E. Andrews, J.A. Ogren, S. Kinne, B. Samset

General comments: The authors present data of situ measurements from aircraft profile flights from which calculations of AOD, single scattering albedo (SSA), and Absorption Aerosol Optical Depth (AAOD) are compared to remote sensing measurements (of AOD) and retrievals of SSA and AAOD from AERONET sun-sky radiometers. These comparisons are made for two sites in the USA and for primarily low AOD levels, mostly less than ∼0.25 at 440 nm. This is well below the AERONET recommendations for use of absorption parameters from their retrievals (>0.4 at 440 nm is recommended), and

although the authors discuss this in the text this recommended low AOD threshold is conspicuously absent from both the Abstract and Conclusions sections (and this needs to be remedied). The authors state that in prior publications "... the in-situ derived AOD values tend to be slightly lower than the AOD retrieved from remote sensing measurements." They fail to point out that the in situ measurements rarely if ever measure the total column AOD, which includes both mid- to upper-tropospheric aerosol plus stratospheric AOD. The authors should include some references and discussion on the AOD that is not measured by in situ instruments in the upper troposphere and stratosphere since the aircraft do not fly complete profiles from the surface on upward into the stratosphere. Discussion of the fact that the aircraft profiles presented (with 4.2 km above ground level as the maximum in situ sampling altitude) do not actually measure the total atmospheric column AOD needs to be included in this manuscript. Therefore differences in in situ versus AERONET AOD are indeed expected and the AOD would be expected to be somewhat higher for sunphotometer total column AOD than for in situ in most aircraft sampling strategies. Moreover lidar measurements sometimes show mid to upper altitude aerosol layers that this aircraft sampling strategy (max at 4.2 km agl) would not measure. Additionally, Sunphotometers in general and AERONET instruments in particular measure AOD more directly than any other technique and as such these data are considered by the scientific community to be the gold standard of accurate AOD measurement for the total atmospheric column. AERONET measured AOD represent the ambient aerosol optical properties and do not have to be corrected for RH humidification growth effects, loss of large particle sampling, etc. as is required and/or discussed for in situ data utilized in this paper. Nyeki et al. (2012) found that AERONET measured AOD agrees very well with other well-calibrated sunphotometers. At the Davos, Switzerland site the comparison of the time co-located and matched 500 nm AOD differences between AERONET and GAW-PFR from 2007 through 2010 resulted in a mean AOD difference of -0.0024 and a root-mean square error of 0.0071. These issues should be included in the discussion on AERONET data, and in the section on comparison of AOD from AERONET measurements to in situ

estimates. Accuracy of AOD is very important in this paper as AAOD is derived from AOD values and the AOD values derived from aircraft profiles (after corrections to make ambient estimations) and also from models (such as within AEROCOM) can be either biased or have significant uncertainties. Furthermore, in order to better understand the comparisons of aircraft profiles to AERONET measurements a scatterplot of computed Extinction Angstrom Exponent (EAE; 440 - 675 nm) estimated from the aircraft data versus AERONET measured EAE needs to be added to Figure 3. This comparison of EAE is pertinent to the discussion in the current text of aircraft inlet sampling issues regarding possible large-sized particle losses.

There is a lack of discussion in the paper of how the uncertainties of the in situ measurements change as aerosol concentrations decrease. All measurement methodologies suffer from issues related to a decrease in signal at low concentrations (relative to potential instrumental noise and offsets), therefore I think that discussion of how the in situ measurement uncertainty changes with aerosol concentration is a very important aspect that needs to be included in the manuscript. Since the paper focuses primarily on low AOD cases, this is a critical issue that is surprisingly neglected in the current manuscript. Additionally it is necessary to summarize in the text a description of the methodology used for computation of profile weighting of the in situ SSA estimates during each aircraft flight. Are these SSA values at each altitude weighted by the extinction coefficient at that altitude, thereby effectively giving higher weighting to the measurements at altitudes that had the highest aerosol concentrations? The AERONET retrievals of SSA are effective optical extinction weighted values for the total atmospheric column, therefore extinction weighting of the in situ data would be the most rigorous way to compare similar quantities. The authors also need to show plots of the in situ aircraft measured/computed SSA altitude profiles to show how SSA varies as a function of altitude for several days of varying AOD magnitude. This is important as it can provide some needed information on how the in situ SSA measurement data look at very low concentrations, especially higher than 3 km above ground level on very low AOD days and also on some moderately high AOD days.

In the abstract you state: "The tendency of AERONET inversions to overestimate absorption at low AOD values is generally consistent with other published comparisons." However the published comparisons between AERONET retrievals and in situ measured SSA shown in Figure 6 are not for low AOD (the AOD are moderate to high in the Figure) and also the SSA differences are generally within the combined uncertainty estimates of the two different techniques (see numerous additional comments on Figure 6 data below in 'Specific Comments'). Since it has never been established that the in situ measurements of SSA have no bias of their own then it is not possible to say that the AERONET measurements of SSA at moderate to high AOD are biased since there is no absolute benchmark for comparison purposes. Additionally since you state that the science is unclear on absorption enhancement due to coated absorbing particles (Section 2.4.1, line 404-407) you need to give a detailed explanation as to how this unknown factor was incorporated into the uncertainty estimates you made for the in situ measured single scattering albedo (it seems to have been ignored in your estimates). The aircraft sampled aerosols are dried first and therefore true atmospheric ambient state aerosol optical properties are actually not measured directly during the profiles. This is important regarding your claim of relative bias of single scattering albedo from one measurement type versus another since you cannot rigorously state (or indirectly suggest) that the aircraft measurements of single scattering albedo are unbiased given that the ambient state optical properties of the aerosols were not directly measured by your in-situ instruments.

The climatological comparison of in situ and AERONET values in Figure 7 is a very important figure in this paper. This figure suggests that if AERONET data are wisely utilized (as done by Bond et al. (2013), for example) then the SSA differences between the two methodologies can be relatively small. The large differences in the time matched data at Bondville and SGP sites from in situ flights and the AERONET retrievals shown in previous figures (Figs. 4-6) are not nearly as evident in Figure 7, especially for the SGP site. There is a surprising lack of discussion of this apparent discrepancy between the matched aircraft profile/AERONET data and the 'climatological' comparisons and the reasons for it. There is also a surprising lack of emphasis on the SSA comparison results shown in Figure 7 in the Abstract and Conclusions given the importance of this result. Additionally it is very interesting that in Figure 8 the in situ surface measurements of SSA agree quite well with AERONET retrieved SSA for both sites, with excellent agreement at SGP site and within uncertainty bounds for the BND site except for extremely low AOD of less than 0.05. The authors have stated that the RH for the surface measurements are all <40%, although this is somewhat surprising given that the surface RH is typically >40% at this location, or perhaps measurements are never made when RH exceeds 40%? Or maybe the surface data that are shown are for the dried aerosol only? If so, you should apply the same humidification factors to the surface in situ data that you have applied to the aircraft profile data in this analysis to make the comparisons in Figure 8 consistent. A look at climatological data for Enid, Oklahoma and Ponca City, Oklahoma (same region as the SGP site) show daily average surface minimum RH of >40% and average Maximum RH of >75-80% for almost all days of the year. The surface in situ measured SSA to AERONET retrieval comparison result may be particularly interesting since the aerosol concentrations are often highest near the surface and therefore the in situ measurements made at the surface should have less uncertainty than those made at high altitudes where the concentrations may be very low. The authors should also present a comparison of the SSA and Extinction Angstrom Exponent measurements made at the lowest flight altitudes during profiles to those made at the surface by similar in situ instrumentation to show how good the agreement is between these measurements and to prove that the aircraft inlet sampling issues mentioned in the manuscript do not result in significant measurement uncertainties.

Specific Comments:

Abstract: You also state: "We conclude that scaling modeled black carbon concentrations upwards to match AERONET retrievals of AAOD may lead to aerosol absorption overestimates in regions of low AOD." This statement is somewhat simplistic and misleading since it does not reflect the much better comparisons shown in Figure 7 for 'climatological' analyses. It also ignores the well thought out application of the use of AERONET retrieved SSA values as weighted by higher AOD observations and then applied to highly accurate AOD measurements at all AOD levels from AERONET, similar to the approach of Bond et al. (2013).

Introduction (lines 77-79): You state: "Moreover, by invalidating low AOD cases, the AAOD values that are retained in AERONET Level-2 data may be biased high." Again, it is misleading and simplistic to suggest that careful investigators would take the AAOD values from only Level 2 data and assume that they can be utilized as is. Many researchers have already utilized a much more intelligent approach: first estimate SSA at higher AOD from AERONET, and then apply those values to ALL levels of AOD (see Bond et al., 2013). I suggest that you remove or modify this sentence.

Introduction (lines 142-143): I assume you mean Dubovik et al. (2000). Dubovik et al. (2002) is not in the reference list.

Section 2.1 (lines 185-187): Please elaborate what you mean by improving measurement statistics here. It would seem that the aircraft instruments 10-minute sampling rate at higher altitudes is an attempt to overcome issues associated with low aerosol concentrations and associated limits of instrumental sensitivity. Therefore on very low AOD days it would seem that an even longer time interval than 10 minutes would be justified. Please elaborate on the sampling strategy and state whether it was modified for very low aerosol concentrations (very low AOD days).

Section 2.1 (line 194): "Only complete profiles were used in this analysis." Please state here that complete profiles as made by the aircraft do not equate to complete atmospheric profiles. None of the aerosol from 4.2 km agl through the stratosphere is sampled in the flights. Especially for very some low AOD days and (and also for some moderate-high AOD days in summer with strong convection) it is expected that a significant amount of the AOD actually occurs above 4.2 km agl. These upper aerosol

layers that are often seen in lidar data may have different optical properties than lower altitude aerosols.

Section 2.1 (line 216-219): Discuss how assuming a constant hygroscopic growth parameter would cause uncertainties when seasonal variation in aerosol type exits. Especially in spring, aerosol type may include biomass burning (crop waste or grass burning) and also dust from the Great Plains region (see Ginoux et al. (2010)), plus pollen from grass and trees.

Section 2.1 (line 267): Please change "column average properties" here to "flight profile average properties" to accurately reflect the fact that the aircraft does not measure the total atmospheric column, as AERONET does.

Section 2.2 (line 290-292): Please add 'calibrations' before 'corrections' as the consistent high accuracy calibration of AOD and sky radiances are the basis for what makes AERONET data so valuable.

Section 2.2 (line 294): Please add that the AERONET data is Version 2 data, since the Version 3 database will be available in the near future.

Section 2.4.1 (line 356-358): Please discuss whether you accounted for soil dust and biomass burning aerosols in the 'aerosol chemistry' mentioned here.

Section 2.4.1 (line 361-364): In your discussion of RH levels during the profile flights please include some mention of the higher RH (RH halos) that typically exist in the vicinity of non-precipitating cumulus clouds that are imbedded within the aerosol layer < 4 km. Higher RH near cumulus clouds and higher AOD in the near Cu cloud environment (likely due to combined humidification, cloud processing of aerosols and rapid gas-to-particle conversions) were observed by Jeong and Li (2010) at the SGP site utilizing both AERONET data and in situ aircraft data. If you only flew aircraft profiles on cloudless time periods or avoided flying near clouds then this needs to be documented in the manuscript, as the sampling could possibly be skewed to specific meteorological

and/or cloudiness conditions.

Section 2.4.1 (line 401-402): Although biomass burning does not have a consistent influence at these sites it is episodic, therefore did you exclude these biomass burning aerosol episodes from your data analysis? If so how did you identify the biomass burning episodes?

Section 2.4.1 (line 443-445): Please state here that the uncertainty estimate for the in situ SSA of ∼0.04 is a lower bound since it does not take into account the effect of particle coatings on aerosols since the aerosols are modified (dried) before the measurements are taken, plus some fraction of the coarse mode particles are not sampled.

Section 2.4.1 (line 451-454): You state here that ∼15% of the aerosol in the column is not sampled below the lowest flight altitude (150 m agl) of the aircraft for in situ measurements, and it can also be inferred that possibly another 15% or more is not sampled above the highest flight altitude on very low AOD days or high AOD days with layering from convective vertical aerosol transport. Therefore it is likely that ∼30% of the aerosol in the total atmospheric column is not sampled by your aircraft vertical profiles. This issue needs some discussion in the text and also should be factored into your uncertainties of in situ measured SSA (or clearly state that it has been ignored).

Section 2.4.1 (line 464-467): Please elaborate here on whether the estimates of the percentage of aerosols above the highest flight altitudes as analyzed by Andrews et al. (2004) were comparisons made for all AOD levels and seasons. It would not be surprising for a greater percentage of AOD above flight altitudes to occur in summer when convection is stronger (transporting boundary layer aerosols upwards), or also in all seasons when AOD is very low since there is always some background mid-tropospheric to stratospheric AOD present which constitutes a greater percentage of total AOD when AOD magnitudes are very low.

Figure 3: In Figure 3, please explain how you can have 3 observation points of AERONET measured AOD at 675 nm ranging from ∼0.5 to ∼1.0 at INSITU AOD of

~0.15 when there does not seem to be any corresponding data at 440 nm in the plot above it. This does not seem possible, and should be explained in the text.

Section 3 (line 528-530): Please also add to this paragraph that fact that the in situ under sampling of the total atmospheric column AOD is due to the restricted altitudes of the flight profile measurements (150 meters to 4200 meters).

Section 3.1.2 (line 604-607): Please note that the in situ instrument known cutoff of 5 micron for particle diameter for the aircraft sampling would also contribute to an under sampling of total column AOD, in addition to the incomplete altitudinal atmospheric profile for the total column AOD.

Figure 6 (numerous comments follow regarding some of the referenced data sets plotted in the Figure, especially note the issues regarding aircraft sampling and also the fact that some papers published Version 1 data that were biased due to inaccurate surface albedo assumptions, versus current Version 2 data that became available in 2006): Osborne et al. [2008] compared three cases of aircraft flights (on three different days) over the same site during the same experiment with the same instruments and aircraft but found that the aircraft in situ measured SSA values ranged from 0.04 to 0.07 higher than the AERONET version 2 retrievals. However, for all three of these cases the aircraft measured Angstrom exponents were found to be about 0.40 lower than the AERONET measured values. This discrepancy in AE suggests that the aircraft may have sampled a different fine and coarse mode fraction mixture than the column integrated value measured by AERONET, and the higher SSA in conjunction with lower AE measured by the aircraft is consistent with this possibility. In fact, for the linear fit of SSA versus AE for all aircraft data from DABEX, reported in the work of Johnson et al. [2008], a difference of 0.40 in AE corresponds to a difference in SSA of about 0.06, almost the same value of the bias reported in Osborne et al. [2008].

Johnson et al. [2009] compared in situ measured aerosol optical properties from an aircraft vertical profile flight over the Banizoumbou (Niger) AERONET site on 19 January 2006. This was a mixed aerosol case with Angstrom exponent (450–700 nm) of approx. 0.8–0.9 and high 550 nm AOD of approx. 0.75, where a shallow dust layer up to 1 km altitude was overlain by a layer of predominantly fine mode smoke. Both aircraft and AERONET measurements of column integrated AOD at 550 nm and of AE were in good agreement for this case, with dAOD = 0.08 and dAE = 0.06, suggesting that both were sampling the same aerosol mixture. The aircraft measured column mean SSA at 550 nm (from PSAP and nephelometer) was 0.87, in good agreement with the AERONET retrieval of 0.85 (interpolated to 550 nm).

Magi et al. (2005; JAS) Note: Version 1 retrievals were 0.015 lower than V2 retrievals on this day at 1310 UTC at COVE site: From the paper: "Ground-based retrievals of SSA were obtained by the Aerosol Robotic Network (AERONET) sun photometers (e.g., Dubovik et al. 2000) during the CLAMS field campaign from a site known as the Clouds and the Earth's Radiant Energy System (CERES) Ocean Validation Experiment (COVE; 36.98 N, 75.78 W). The vertical profiles were often spatially located close to COVE. The mean value of SSA at 550 nm from AERONET retrieval data (processed to remove clouds and manually quality assured) is 0.94 +- 0.03. Therefore, the mean value of SSA retrieved from AERONET agrees with mean value of SSA derived from our in situ airborne measurements (0.96 +- 0.03) to within one standard deviation. On 17 July 2001, measurements were made from the UW aircraft and the COVE site that were both temporally (the aircraft vertical profile was from 1304–1337 UTC and the AERONET retrieval was at 1310 UTC) and spatially (the aircraft was ; 2.5 km from COVE) collocated. The mean value of SSA calculated from the airborne in situ measurements made in polluted layers during this vertical profile was 0.97 +- 0.02; the corresponding column-averaged value of SSA for accumulation mode particles retrieved from the AERONET data was 0.90 +- 0.03 (VERSION 1 data). Particle losses in the sampling system for the in situ instruments could have contributed to an underestimate of the absorbing component of the aerosol. Spatial variability may have played a role as well."

Mallet et al. (2005): V2 almost same as V1 at 0.932 at 550 nm at 6 UTC: AERONET retrieval (at 0600 UTC ; on June 25, 2001) of the single scattering albedo at Avignon indicated a coherent value (SSA 0.93 at 550 nm) compared to the one obtained from optical measurements for flight 41 (0515–0537 UTC, SSA 0.94 in the PBL).

Haywood et al., (2003): Comparison of aerosol size distributions, radiative properties, and optical depths determined by aircraft observations and Sun photometers during SAFARI 2000 V2 SSA = 0.84, 0.83, 0.81, 0.80 for 440, 675, 870 and 1020 nm V1 SSA = 0.88, 0.87, 0.84, 0.82 for 440, 675, 870 and 1020 nm The corresponding SSA for the mean size distribution used in the calculations derived from the PCASP distributions and from the nephelometer and PSAP on the C-130 is 0.90, 0.87, 0.85, and 0.82 (Table 1).

Figure 7 caption: One detail regarding the caption of Figure 7 that is misleading is the Level 2 AOD shown in the plot. Please note that Level 2 AOD exists for all AOD levels not just for AERONET almucantar retrievals for which AOD is >0.4 at 440 nm. Level 2 almucantar retrievals of size distributions are made for all AOD levels, but refractive indices are only given for AOD>0.4 at 440 nm. Therefore the authors need to clearly describe and accurately label in the figure caption that this data as only associated with AERONET almucantar retrievals for which AOD>0.4 and therefore have error bars on SSA of ∼0.03. The actual complete Level 2 AOD data set (for all AOD levels) shows monthly means that are significantly higher in summer with many more days of data sampled and many more partly cloudy to mostly cloudy days sampled also (see Jeong et al., 2010; JGR for a discussion of higher AOD in the near cloud environments at the SGP site).

Section 3.2 (line 692-695): Your statements here assume that the in situ determinations of SSA are un-biased (despite the fact that ambient aerosol properties are not actually measured). This has not been proven in the paper, especially since the in situ data have to be corrected for humidification effects, the total column aerosol is not sampled, and the effects of aerosol coatings are not accounted for (therefore blindly assumed

have no effect). Please revise or eliminate these sentences.

Section 3.3 (line 754-768): Please note that AOD sampled by AERONET in the Level 2 dataset (not just for the subset that have L2 retrievals) includes many more days of data than the in situ flights, and is therefore a much more statistically robust data sample. Please note that the Level 2 AOD climatology for the SGP site (average of 13-19 years per month) shows significantly higher AOD (440 nm) than shown for L1.5* in Figure 7, For example for the SGP site the August monthly mean AOD is 0.272 and the September monthly mean is 0.215 at 440 nm. Similarly for the BND site the L2 monthly means of AOD(440 nm) for June, July, August, and September are 0.282, 0.329, 0.343 and 0.283 respectively (computed from 15-17 years of data per month). These monthly means are significantly higher than the AERONET values shown in Figure 7, since the data in Fig 7 are only AOD associated with the Dubovik and King algorithm retrievals.

Section 3.3 (line 781-783): You state: "The AERONET 1.5* SSA values tend to be quite a bit lower than the other data sets at both sites, which is why the AERONET 1.5* AAOD values tend to be higher (recall that for AERONET data AAOD is calculated using AAOD=(1-SSA)*AOD)." No, this is not really accurate, since as shown on Figure 7, at the SGP site the agreement between the AERONET L1.5* data and in situ measurements of SSA are well within the uncertainty of the measurements for all months (and you have not proved that the in situ is not biased). Please revise this sentence to reflect this fact as presented by the data shown in Figure 7b.

Section 3.4 (line 875-879): Again, you have omitted the fact that for the SGP site the agreement between the in situ estimations of SSA and the AERONET retrievals of SSA are within the uncertainty levels of these data sets over the entire range of AOD shown in Figure 8. The way the paper is written there seems to be a consistent attempt to steer the reader to the conclusion that the AERONET retrievals are biased low despite significant uncertainties in the in situ determinations and despite the fact that the in situ instruments do not measure ambient aerosol properties directly without corrections. Therefore it is not proven that the in situ determinations of SSA are unbiased

themselves, so the text and title require rewriting to acknowledge this.

---

## Referee Comment (RC3) · Anonymous Referee #1 · 17 Nov 2016

This work compares AERONET column retrievals of aerosol single-scatter albedo to column single-scatter albedos obtained from in situ flight profiles. This is an important topic, because modeling groups regularly use the AERONET absorption products to infer or constrain atmospheric black carbon concentrations. This work focuses especially on assessing the capability of the Level 1.5 AERONET retrievals that are obtained at low aerosol optical depth (AOD(440) $<$ 0.4) common to many locations throughout the world, which has received little or no attention thus far. The paper is clear and well written, scientifically solid, and thoroughly covers nearly all issues associated with this topic. I have a few topical comments and a laundry list of small details that will make the paper stronger, in my opinion.

[Figure]

Column Single-scatter Albedos

Once concern is the hygroscopic growth corrections for the nephelometer and PSAP measurements. Since particles are dried prior to entering the aircraft instruments, the authors employ an empirical correction to the scattering measurements and no correction to the absorption measurements. The scattering correction is an exponential function of RH (Eq. 1), and the exponents that they choose for the corrections are based upon aerosol chemistry measurements at nearby IMPROVE sites. This is all very standard in the business, and the pitfalls of empirical scattering corrections at high RH are well documented. The authors argue that they are mostly operating at low RH, so that their column AODs are not very sensitive to the hygroscopic growth corrections.

This is all fine, except that Figure 3 shows a significant amount of scatter in the author's AOD comparisons that undoubtedly works its way into the SSA comparisons. This is unavoidable when applying climatological hygroscopic growth corrections to in situ scattering profiles, but why not use AERONET AODs to compute column SSAs? Unlike AERONET's absorption and size distribution *retrievals*, AERONET AOD is a robust *measurement* that is widely accepted by the community. Thus, the authors could compute a "hybrid" SSA, whereby

$$SSA_{hybrid} = \frac{AOD_{aeronet} - AAOD_{psap}}{AOD_{aeronet}}.$$

This will isolate the differences in SSA caused by absorption, which is of primary interest to the community. This approach will also remove the uncertainty associated with the empirical hygroscopic growth factors applied to the scattering measurements, and it will remove the scattering uncertainty associated with undersampling the coarse mode. Since AAOD is very well-correlated at SGP (per Fig 4), I suspect that this approach will tighten up the SSA scatter at that site in Figure 5.

Now, some readers may find issues with this approach as well, so I suggest including this approach as an additional alternative to the material already presented in the paper. For some of us, this approach will remove doubts about hygroscopic growth corrections and inlet size cutoffs for the scattering measurements.

Merging of in situ and AERONET data

Figure 3-6: The authors use multiple AERONET AODs and AAODs for each flight profile (i.e., the number of AERONET AODs at BND is 662, but there are only 72 flight profiles). This is not appropriate, in my opinion. The flight profiles occur over a 2-hour period, so I argue that they are comparing quasi 2-hour averages to AERONET's quasi-instantaneous retrievals. The atmosphere can easily change during this timeframe, especially since the authors allow an additional 1 hour before the flight and 3 hours after the end of the flight for including additional AERONET retrievals. Apparently the spread of AOD for a given flight profile can be quite large over this 6 hour period, too, as seen in the upper left panel of Figure 3 (for instance the AERONET AOD(440) ranged from 0.45 to 0.55 when in situ AOD(440) = 0.3). Additionally, the authors have argued that aerosols sampled within a 3-4 hour period have high autocorrelations, per their Figure 2. Since the premise of their comparison is that they are essentially sampling homogeneous air masses on average, why not use averaged AERONET values for comparison to the flight profiles? (After all, there is a heck of a lot of averaging going on for each in situ flight leg.) This will reduce the number of points in figures 3,4,5, and 6, but I believe that this will provide a more accurate presentation of the data. Having said all that, I am OK with multiple AODs per in situ measurement (red points in Fig 3), as this provides an indication of the variability in aerosol optical properties that occur in the atmosphere during the flight profiles. A comment from the authors pointing this out would be helpful, though (if I've got this correct). Multiple AERONET AAODs and SSAs per flight in Figures 4 and 5 serve no purpose, though.

The authors present autocorrelations in Figure 2, which is a robust way to identify acceptable collocation lag times. They conclude that a 3-hour lag is acceptable for their analysis. They mention that the auto-correlations of scattering is greater than 0.8 for

up to a 4-5 hr time lag (lines 316-317), but they do not specify that the auto-correlation of absorption is only 0.75 at BND and 0.55 at SGP. This should be explicitly stated here, because it is important for putting Figure 4 into context. If the auto-correlations in figure 2 are robust values computed with single instruments, then Figure 2 represents the maximum correlation that we can realistically expect to achieve in a comparison of two different instruments. Thus, $R^2 = 0.77$ at SGP in Figure 4 (or, $R = 0.87$) is actually an excellent result for AERONET, as the correlation is substantially greater than the auto-correlation in Figure 2. It is also an excellent result for the authors, as it demonstrates that they have put great care into producing a clean analysis. This is something worth mentioning in the article. Surprisingly, the BND site has higher 3-hour autocorrelations for absorption than SGP (r = 0.75 for BND and r = 0.55 for SGP, per figure 2), but the results for BND in Fig 4 are not as clean as SGP. Nonetheless, the correlation coefficients for BND in Figure 4 (R = 0.53-0.58) are not that far from the auto-correlation of 0.75 for absorption at BND in Fig 2.

This brings up another confusing point: The authors define $R^2$ as the correlation coefficient on lines 518, but $R^2$ is usually reserved for the coefficient of determination, and $R$ is the correlation coefficient. The authors also use "r" for the auto-correlation in Figure 2, which adds to the confusion. Thus, I am unsure if the $R^2$ values in Figures 3-5 indicate correlation coefficients or coefficients of determinations. Additionally, how does one determine either parameter (correlation coefficient or coefficient of determination) when the number of points on the y-axis is different than the number of points on the x-axis (e.g., R2 = 0.84 for 56 AERONET retrievals and only 24 flights at BND for the upper left panel of Figure 3, etc.)? The authors need to explain how they were able to do this (mathematically) in both the text and in the figure captions.

**Mixing State**

It is easy to demonstrate (with publicly available Mie codes) that an internally-mixed absorbing aerosol particle has a much higher absorption coefficient than an identical externally-mixed counterpart. The amount of absorption enhancement varies with the

particle size and coating thickness, but factors of two or more absorption enhance-ments are reasonable. Recent measurements utilizing the SP2 instrument indicate that roughly half of biomass burning and background aerosols are internally mixed (e.g., Schwarz, GRL, 2008).

It is not clear to me that a filter-based instrument can differentiate between internally and externally mixed particles because the EM field on the particle changes when it touches the substrate. Therefore, any enhanced absorption associated with internal mixing of atmospheric carbon particles might not be detectable with a filter measure-ment. The altering of aerosol absorption is further aggravated by the drying process, which can modify aerosols by removing coatings that are composed of semi-volatile compounds, inorganics, and certainly water.

The authors state on line 406 that "measured absorption Ångström exponents are quite low (close to 1) suggesting little influence of coatings." This is a weak argument, how-ever, because the absorption Ångström exponent is measured *after* the drying process, which could have easily removed semi-volatile coatings. Additionally, coated carbona-ceous particles of about 0.1 $\mu$m diameter and a wide range of coating thicknesses can easily have absorption Ångström exponents close to 1, per Gyawali (ACP, 2009, Figure 8).

Finally, the authors conclude on line 434 that "...it is not possible to estimate the actual uncertainty in the in-situ light absorption measurements reported here due to coating effects." Since accurately estimating this uncertainty is not possible, the authors seem to be choosing an uncertainty of zero; I believe that many readers will not be satisfied with this approach.

Hence, I suggest that the authors include a more conservative uncertainty of 50% for the PSAP absorption coefficient (rather than 25%) to account for all possible PSAP errors (you've already included the 50% value in much of your discussion on pages 10-11). This number still comes out of thin air (just like zero), but you can point to the

50% enhancement factor recommended by Bond and Bergstrom (AST, 2006). Thus, I recommend including a comment on line 445 that if the PSAP absorption uncertainty is 50%, the SSA uncertainty at SSA=0.9 is approximately 0.06.

Optical and Aerodynamic sizes

The authors state that their inlet samples particles with aerodynamic diameters less than 7 $\mu$m, and that the particle diameter for 50% sampling efficiency is 5 $\mu$m (lines 196-198). They go on to address this issue on lines 601-617 by using the AERONET size distributions to estimate the fraction of aerosol extinction that occurs at AERONET diameters of less than 5 $\mu$m. By doing this, the authors are assuming that aerodynamic diameter is equal to volume equivalent diameter, which is not generally the case (unless the aerosols are spherical and the aerosol density equals 1). Since $D_{vol} \propto D_{aer}/\sqrt{\rho}$, the authors should use a much smaller cutoff diameter than 5 $\mu$m in their equation on line 606. For instance, if one assumes $\rho = 2$, then $D_{vol}$ = 3.53 for spheical aerosols; if $\rho = 2.7$, then $D_{vol}$ = 3.04. These smaller cutoff values will decrease their estimated extinction fractions of the particles entering their inlets. However, the authors correctly argue on lines 612-617 that under-sampling large particles probably lowers the in situ SSA, and that correcting this artifact will probably make the discrepancy between AERONET and in situ worse. Nonetheless, some recognition of the optical/aerodynamic size difference will indicate to readers that you considered this issue.

Tables and Figures

I like Table 1, but the authors need to include the number of AERONET retrievals and the number of in situ flights that are used to compute these statistics. Standard deviations of the means (SDOM) would be nice, too, so that readers can quickly see that the difference between the averages are statistically significant.

I am having difficulty understanding Table 2. You list 56 retrievals with Level 2 AOD + almucantar at BND, but only 6 retrievals with Level 2 + almucantar + $AOD_{440} > 0.2$.

The 2nd number should be greater than the first number, right? (The first number only includes AOD > 0.4, but the 2nd number includes all retrievals with AOD > 0.2). Unless... do you mean "size distribution retrieval" instead of "almucantar retrieval?" The almucantar products include size, AAOD, SSA, and complex refractive index; size distributions are the only Level 2 almucantar products available at AOD(440) < 0.4.

The legend in Figure 1 is confusing. The legend contains two solid black lines (Direct RF BCFF and Global), but there is only one solid black line in the figure. The legend also contains two variations of dashed black lines (Land and Sea), but there are no black dashed lines in the figure. The figure contains two red dashed lines that are not shown in the legend.

The authors need to explain that "r" represents auto-correlation in Figure 2, as it is not obvious from the caption. Also, presumably the correlograms correspond to scattering and absorption coefficients measured with the airborne nephelometer and PSAP; that should also be stated in the caption.

Line-by-line Details

The authors lament on line 27 that the terminology for absorbing aerosols are imprecise, but Andreae and Galencsér (2006) provide precise definitions for soot, soot carbon, brown carbon, light absorbing carbon, elemental carbon, apparent elemental carbon, black carbon, and equivalent black carbon. It would be nice if the community embraced this paper as the "go-to" article for carbonaceous aerosol definitions.

Line 77-78: the authors state "...by invalidating low AOD cases, the AAOD values that are retained in the AERONET Level-2 data may be biased high." This is not quite correct. It would be more accurate to state that "...the averaged AAODs" or "...the climatological AAODs" are biased high.

Lines 82, 93: Authors mention several Aerocom models, but do not tell us which ones or which Aerocom experiments they are pulling the data from.

Line 185: How much horizontal distance is covered in these 5- and 10-minute flight legs? This is important for readers to understand the auto-correlation within the profiles.

Lines 192-194 states that there are 253 complete profiles at BND and 132 complete profiles at SGP. However, only 24 and 14 flights were used in the analysis of Figures 3-6. I think that it is important to mention this here, because this description sounds like you have a much larger dataset available than you really do. This might be a good opportunity to mention that the stringent AERONET cloud screening procedure drastically reduces the number of possible comparisons, and this must also be considered by modelers when they utilize AERONET retrievals.

Lines 322-325: The authors state:
"Because the profiles are "stair-step" descents from 4600 m asl down to 450 m asl (e.g., see Figure 4 in Sheridan et al., 2012), matching with AERONET retrievals at the end of the profile means that the matches are more closely aligned with when the airplane is in the boundary layer and thus, typically, sampling the highest aerosol concentrations." This is inconsistent with other statements in the paper. The flight takes 2 hours and the authors are matching AERONET retrievals within +/- 3 hours of the end of the profile. Hence, the actual retrieval could occur up to 1 hour before the flight profile begins, and anytime during the flight. Thus, the AERONET retrievals are not necessarily "more closely aligned when the airplane is in the boundary layer.

Line 359: I don't understand how the authors can test $\gamma \pm 1$ standard deviation when they do not measure $\gamma$.

Lines 360-361: Authors estimate the uncertainty in AOD by varying $\gamma$ by up to 2 S.D., but there are hundreds of sunphotometer AODs available during their flights (per figure 3). Why not just compare their in situ AODs to AERONET AODs?

Line 379: The Lack 2008 citation is not included in the references.

Lines 434-435: Authors state that it is not possible to estimate the uncertainty in measured coatings due to coating effects. Consequently, they seem to be choosing zero uncertainty associated with coatings. Many readers will not agree with this choice.

Line 516: Authors state that AERONET AOD tends to be higher than in situ AOD in Figure 3; however, the slope is only greater than 1 in one of those figures, although the offset is always positive. So AERONET $>$ in situ AOD is not obvious, especially since many of the regression lines in Figure 3 fall below the 1:1 line. Additionally, AERONET median and average AODs are almost always less than the corresponding in situ AODs in Table 1, indicating that line 516 is not correct for the medians or the averages.

Line 519: Should be changed to "The correlations improve **when subsetted** for the more restrictive Level-2 almucantar retrievals." Otherwise, it sounds like the almucantar scans are incorporated into the AOD measurements, which is not the case.

Lines 541-543:
Authors state: "although the scatter in the relationships (particularly at BND) suggests that a multiplicative factor doesn't represent the relationship very well." I agree that the BND data looks quite scattered, but the SGP values do not look scattered. The R2 values are 0.76-0.77, which is excellent compared to the autocorrelation values for absorption provided in Fig 2 ($r_{abs}$ = 0.45-0.55). Additionally, the scatter shown for AERONET AAOD is deceiving. There are approximately 2 AERONET retrievals for each in situ flight. The in situ flight represents a 2-hour average, but each AERONET retrieval occurs over the period of a 10-minute almucantar scan. The AERONET scans could occur up to 6 hours apart, and up to 3 hours after the flight is complete. Finally, $R^2$ = 0.3 implies that R = 0.55. Thus, the correlation in Figs 4 and Fig 2 at BND are not very different.

Lines 574-576: Well... you don't really have enough data to draw this conclusion, in my opinion. Why not include the averages for the purple points in Table 1, though?

Line 579: "Figure 3 shows that the AERONET AOD may be slightly larger than the

in-situ AOD..." Again, this is inconsistent with Table 1.

Line 644: It should be noted that 100 km is a long ways away! What kind of auto-correlations did Anderson (1998) find at those distances? That value should be noted here.

Line 673: Authors state that SSA differences are greater than would be expected from random error, even when AOD > 0.4. Why not quantify that? That is, compute the mean and 2xSDOM of all independent measurements in that figure to see if the null hypothesis is valid; add this point with the resulting errorbar to the figure. It is important to isolate the independent measurements, though – multiple values that are obtained within 4-5 hours are not independent measurements (as you argued in Fig 2).

Page 10+: Since the authors are discussing non-synchronized data here, I think that they should note that the models and the in situ flights include cloudy periods, whereas the AERONET data are stringently cloud-screened.

Lines 830-837: This is a nice approach in some regions, but it is virtually impossible to capture seasonal variability over North America with this approach. For instance, from 4/1994 through 10/2012 the Cart Site had 3 lev2 retrievals in DJF, 22 Lev2 retrievals in MAM, 122 lev2 retrievals in JJA, and 51 retrievals in SON. Thus, results would be skewed to the high humidity summer periods.

Lines 840-844: This passage needs some massaging, as it makes no sense to me.

Line 872: Emphasize that this comment only pertains to AOD < 0.2.

Line 1125: Müller (2012) is not included in the text. This is an important citation that needs to be presented in the text, as it discusses the difficulties associated with using in situ instruments to measure absorption.

**References**

Andreae, M. O., and A. Gelencsér (2006), Black carbon or brown carbon? The nature of light-absorbing carbonaceous aerosols, Atmos. Chem. Phys., 6, 3131-3148.

---

## Author Response (AR1)

**AAOD Response to Reviewers and other Comments**

**Nick Schutgens' comment:**
A very useful paper, and it will be interesting to see how to reconcile various measurements. Here I just want to mention the issue of temporal sampling, see also: http://www.atmoschem-phys.net/16/1065/2016/
While this temporal sampling issue is important for model evaluation, it is equally important in comparing different observational datasets. I see two issues relevant to the current paper:

Page 18, Line 732-735: Fig 7 was apparently made with different samplings of the in-situ and AERONET measurements. Is that the case for other figures as well? How might that affect results?
Figures 3-5 were made for matched samplings of the in-situ and AERONET measurements where matched means that the measurements were made within 3 h and 15 km of each other. Given the high correlation in AOD we are reasonably confident that with these sampling constraints the in-situ and remote sensing instruments were measuring in the same atmospheric column. The literature studies cited in Table 4 and included in Figure 6 used similar temporal and spatial matching criteria (see comments column in Table 4), with the exception of the DABEX dust/biomass burning flights (Osborne et al 2008; Johnson et al 2009) which were matched in time but less so in distance (flight profiles were within 100 km of AERONET retrievals).

In contrast, Figures 7 and 8 utilized the multi-year climatological data sets available for each measurement which have different samplings. Your work (e.g., Schutgens et al., 2016) shows there can be large differences when comparing values obtained with different samplings (more than 100% for AOD), particularly when there are high levels of variability in the data. In our manuscript the different temporal samplings are likely one contributor to the relatively small differences observed between the in-situ (red line) and AERONET 1.5 AOD (black line) although other things (e.g., assumptions about aerosol hygroscopicity, missed aerosol (i.e., due to size cut or flight limitations)) will also contribute. The relatively small differences between the in-situ and AERONET 1.5 AOD suggest there may not be much year-to-year variability at these two sites. The long term surface measurements at the site also suggest there is not much year-to year variability. The effects of different sampling are definitely the primary reason for the difference between the AERONET level 2 almucantar values (AOD and AAOD) and the in-situ measurements.

Page 22, Line 925 - 934: The authors suggest better estimates of AAOD may be obtained by using SSA measured at high AOD and applying it to low AOD cases. They mention possible sampling impacts but seem to feel those may not be that important. I'd like to caution against that.
I attach a figure of the difference in yearly SSA, when that SSA is taken at high AOD or at any AOD, for three different models. At least two models allow differences of more than 0.05. (In general, the MIROC-SPRINTARS model agrees best with AERONET Lev 2 SSA while HadGEM-UKCA is often too high and ECHAM-HAM too low.)

We agree that this is an approach to be cautioned against, particularly as systematic variability between loading and SSA has been observed by both in-situ and AERONET measurements at BND, SGP and many other sites (Delene and Ogren, 2002; Andrews et al., 2011b; Schaefer et al., 2014 and our Figure 8).  Current work by our group shows this systematic variability is also simulated by many global models.  We've re-written the abstract, discussion of Fig 8 and the conclusions  to highlight the importance of the systematic variability we've observed and to note that such systematic variability cautions against the use of applying SSA obtained from high loading to obtain AAOD at low loading conditions via the relationship AAOD=SSA*AOD.  That said – for the specific case of these two sites, we note that using the monthly median SSA from the high loading retrievals would result in a reasonable monthly median AAOD if the high loading SSA was applied to all AOD values.

Finally, it would be useful if the authors made a suggestion under what conditions AERONET SSA conditions may be used. Is AOD > 0.4 sufficient?
I don't think we can say definitively.  Our Figure 6 comparing many field campaign measurements suggests that AOD440>0.25 or 0.3 may be reasonable.  Oleg Dubovik (pers. comm. with co-author Stefan Kinne) thinks AOD440>0.4 may be too restrictive but did not suggest a lower alternative.  We've added the following text to the discussion of Figure 6: "Figure 6 suggests that AERONET retrievals of SSA could perhaps be used at $AOD_{440}$<0.4, perhaps down to $AOD_{440}$~0.25 or ~0.3 – even at those low AOD values the differences in SSA between AERONET and in-situ still tend to be within the AERONET uncertainty.  However, as Figure 6 shows, there are not a lot of direct comparisons to support such a choice."

**Reviewer #1**
This work compares AERONET column retrievals of aerosol single-scatter albedo to column single-scatter albedos obtained from in situ flight profiles. This is an important topic, because modeling groups regularly use the AERONET absorption products to infer or constrain atmospheric black carbon concentrations. This work focuses especially on assessing the capability of the Level 1.5 AERONET retrievals that are obtained at low aerosol optical depth (AOD(440) < 0.4) common to many locations throughout the world, which has received little or no attention thus far. The paper is clear and well written, scientifically solid, and thoroughly covers nearly all issues associated with this topic. I have a few topical comments and a laundry list of small details that will make the paper stronger, in my opinion.

We'd like to thank reviewer#1 for the thoughtful and well-organized review. Below we describe how we've addressed specific comments and suggestions.

*Column Single-scatter Albedos*
Once concern is the hygroscopic growth corrections for the nephelometer and PSAP measurements. Since particles are dried prior to entering the aircraft instruments, the authors employ an empirical correction to the scattering measurements and no correction to the absorption measurements. The scattering correction is an exponential function of RH (Eq. 1), and the exponents that they choose for the corrections are based upon aerosol chemistry measurements at nearby IMPROVE sites. This is all very standard in the business, and the pitfalls of empirical scattering corrections at high RH are well documented. The authors argue that they are mostly operating at low RH, so that their column AODs are not very sensitive to the hygroscopic growth corrections.

This is all fine, except that Figure 3 shows a significant amount of scatter in the author's AOD comparisons that undoubtedly works its way into the SSA comparisons. This is unavoidable when applying climatological hygroscopic growth corrections to in situ scattering profiles, but why not use AERONET AODs to compute column SSAs? Unlike AERONET's absorption and size distribution retrievals, AERONET AOD is a robust measurement that is widely accepted by the community. Thus, the authors could compute a "hybrid" SSA, whereby
SSAhybrid = (AODaeronet – AAODpsap)/AODaeronet

This will isolate the differences in SSA caused by absorption, which is of primary interest to the community. This approach will also remove the uncertainty associated with the empirical hygroscopic growth factors applied to the scattering measurements, and it will remove the scattering uncertainty associated with undersampling the coarse mode. Since AAOD is very well-correlated at SGP (per Fig 4), I suspect that this approach will tighten up the SSA scatter at that site in Figure 5.

Now, some readers may find issues with this approach as well, so I suggest including this approach as an additional alternative to the material already presented in the paper. For some of us, this approach will remove doubts about hygroscopic growth corrections and inlet size cutoffs for the scattering measurements.

This is an interesting suggestion. We've done this analysis and found that, as the reviewer suggested, it tightens up the SSA scatter at SGP and actually even at BND.
We've added this paragraph in section 3.1.1:
"Figure 5 also includes a set of 'hybrid SSA' ($SSA_{hybrid}$) points in yellow. These points have been calculated using the AERONET AOD and the in-situ AAOD:

$$SSA_{hybrid} = (AOD_{AERONET} - AAOD_{PSAP})/AOD_{AERONET} \qquad (3)$$

This hybrid approach to SSA eliminates the uncertainty associated with the empirical hygroscopic growth factors applied to the in-situ scattering measurements, and also removes the scattering uncertainty associated with undersampling the coarse mode. It does not, however, eliminate the uncertainties associated with assuming the absorbing aerosol is hygrophobic, that there is little absorption in the potentially undersampled coarse mode, or the unknown contribution from absorption enhancement. $SSA_{hybrid}$ is very similar to the SSA derived from in-situ measurements, suggesting the primary discrepancy between the AERONET SSA and the in-situ SSA is due to the determination of the absorbing nature of the aerosol, either due to issues with the limitations of the filter-based measurements or to the interpretation of the relative contribution of aerosol absorption from the AERONET inversion retrieval products."

[Figure]

[Figure]

*Merging of in situ and AERONET data*
Figure 3-6: The authors use multiple AERONET AODs and AAODs for each flight profile (i.e., the number of AERONET AODs at BND is 662, but there are only 72 flight profiles). This is not appropriate, in my opinion. The flight profiles occur over a 2-hour period, so I argue that they are comparing quasi 2-hour averages to AERONET's quasi-instantaneous retrievals. The atmosphere can easily change during this timeframe, especially since the authors allow an additional 1 hour before the flight and 3 hours after the end of the flight for including additional AERONET retrievals. Apparently the spread of AOD for a given flight profile can be quite large over this 6 hour period, too, as seen in the upper left panel of Figure 3 (for instance the AERONET AOD(440) ranged from 0.45 to 0.55 when in situ AOD(440) = 0.3). Additionally, the authors have argued that aerosols sampled within a 3-4 hour period have high autocorrelations, per their Figure 2. Since the premise of their comparison is that they are essentially sampling homogeneous air masses on average, why not use averaged AERONET values for comparison to the flight profiles? (After all, there is a heck of a lot of averaging going on for each in situ flight leg.) This will reduce the number of points in figures 3,4,5, and 6, but I believe that this will provide a more accurate presentation of the data. Having said all that, I am OK with multiple AODs per in situ measurement (red points in Fig 3), as this provides an indication of the variability in aerosol optical properties that occur in the atmosphere during the flight profiles. A comment from the authors pointing this out would be helpful, though (if I've got this correct). Multiple AERONET AAODs and SSAs per flight in Figures 4 and 5 serve no purpose, though.

We've remade the plots as suggested:
- keep all the red points in Fig 3, to show variability in AOD during the flights
- change the blue points in Fig 3 to average AERONET AOD during the flights
- change Figs 4&5 to use average AERONET AAOD and SSA during the flights
- Figure 5 now includes the 'hybrid SSA as well'
- The plots reflect the new uncertainty values as described below.

In terms of new text, we've rewritten the first paragraph of section 3.1.1 to reflect the new plots. "The red points represent all direct sun AERONET Level-2 AOD retrievals during the +/-3 hours window around the end of each profile– this provides an indication of the variability in AOD during the in-situ profiling flight."

We also refer to the blue points in the discussion of Figures 3-4-5 as 'flight-averaged'

*Autocorrelation*
The authors present autocorrelations in Figure 2, which is a robust way to identify acceptable collocation lag times. They conclude that a 3-hour lag is acceptable for their analysis. They mention that the auto-correlations of scattering is greater than 0.8 for up to a 4-5 hr time lag (lines 316-317), but they do not specify that the auto-correlation of absorption is only 0.75 at BND and 0.55 at SGP. This should be explicitly stated here, because it is important for putting Figure 4 into context.

DONE

If the auto-correlations in figure 2 are robust values computed with single instruments, then Figure 2 represents the maximum correlation that we can realistically expect to achieve in a comparison of two different instruments. Thus, R2 = 0:77 at SGP in Figure 4 (or, R = 0:87) is actually an excellent result for AERONET, as the correlation is substantially greater than the auto-correlation in Figure 2. It is also an excellent result for the authors, as it demonstrates that they have put great care into producing a clean analysis. This is something worth mentioning in the article. Surprisingly, the BND site has higher 3-hour autocorrelations for absorption than SGP (r = 0.75 for BND and r = 0.55 for SGP, per figure 2), but the results for BND in Fig 4 are not as clean as SGP. Nonetheless, the correlation coefficients for BND in Figure 4 (R = 0.53-0.58) are not that far from the auto-correlation of 0.75 for absorption at BND in Fig 2.

We've added the following sentence to the end of the first paragraph of Section 2.3 (thank you reviewer for some lovely text!):

"Additionally, Figure 2 represents the maximum correlation that we can realistically expect to achieve in a comparison of two different instruments with temporally offset measurements and provides context for the AERONET/in-situ comparisons presented in Section 3."

We've also added these sentences to the discussion of Figure 4 in Section 3.1.1:

"Surprisingly, while the BND site has higher 3-hour autocorrelations for absorption than SGP (R = 0.75 for BND and R = 0.55 for SGP, per Figure 2), the results for BND in Figure 4 indicate less correlation than at SGP for absorption. Nonetheless, the correlation coefficients for BND in Figure 4 ($R^2$=0.49 (blue) and 0.37 (red) correspond to R = 0.70 (blue) and 0.61 (red)) are not that far from the 3 h auto-correlation of R=0.75 for absorption at BND in Figure 2."

This brings up another confusing point: The authors define R2 as the correlation coefficient on lines 518, but R2 is usually reserved for the coefficient of determination, and R is the correlation coefficient. The authors also use "r" for the auto-correlation in Figure 2, which adds to the confusion. Thus, I am unsure if the R2 values in Figures 3-5 indicate correlation coefficients or coefficients of determinations.
Our mistake! R2 values in Figures 3-5 represent coefficient of determination –we've fixed this and the sentence now reads:
"The coefficients of determination ($R^2$) are within the …"

Additionally, how does one determine either parameter (correlation coefficient or coefficient of determination) when the number of points on the y-axis is different than the number of points on the x-axis (e.g., R2 = 0.84 for 56 AERONET retrievals and only 24 flights at BND for the upper left panel of Figure 3, etc.)? The authors need to explain how they were able to do this (mathematically) in both the text and in the figure captions.
In the original submission we determined the R2 values by matching the single in-situ value with each AERONET retrieval that fell within the +/-3h time window, i.e., if there were 3 AERONET retrievals for a given flight they would all correspond to the same x-value to create 3 xy pairs. Since we're now using flight-averaged AERONET retrievals for the comparisons (blue points in Figures 3-5), this is no longer an issue. We still provide a fit for the red points in figure 3.

*Mixing State*
It is easy to demonstrate (with publicly available Mie codes) that an internally-mixed absorbing aerosol particle has a much higher absorption coefficient than an identical externally-mixed counterpart. The amount of absorption enhancement varies with the particle size and coating thickness, but factors of two or more absorption enhancements are reasonable. Recent measurements utilizing the SP2 instrument indicate that roughly half of biomass burning and background aerosols are internally mixed (e.g., Schwarz, GRL, 2008).
It is not clear to me that a filter-based instrument can differentiate between internally and externally mixed particles because the EM field on the particle changes when it touches the substrate. Therefore, any enhanced absorption associated with internal mixing of atmospheric carbon particles might not be detectable with a filter measurement.
We agree that filter-based measurements may have issues with coated particles and may not report the absorption properly and we mention this in the text (original submission, lines 376-379).  Coatings appear to increase the absorption measured by the PSAP, but we don't know by how much – i.e., whether it is more or less than the actual absorption enhancement observed in the atmosphere. The problem is that the effect of the interaction of coated-particles with filters on the measured absorption is unknown – the literature we've seen suggests that filter-based measurements of absorption tend to be *higher* than those made by non-filter based methods. As we already discuss in the text, Lack et al. (2008; 2012) show absorption enhancements for PSAP measurements relative to photo-acoustic (PAS) measurements which are not filter-based.

In the Lack studies, the fact that the PAS observed less absorption than the PSAP is unsettling as the PAS absorption measurements should also include the effects of coatings. Lack et al., (2008) also showed that the discrepancy between the PSAP and PAS increased as the organic aerosol concentration increased. We should also note that, thermally-denuded, filter-based measurements of absorption are lower than non-denuded absorption measurements suggesting the filter-based measurements are capturing at least some of the contribution of the coating. Comparisons of filter-based absorption measurements for denuded and un-denuded particles (e.g., Kanaya et al., 2013; Sinha et al., in revisions, 2017) suggest the un-denuded particles have absorption enhancements of 5-25% relative to those that have been through a denuder. These comparisons show that stripping off coatings and evaporating the non-absorbing particles reduces the measured absorption, i.e., that the effects of coatings is not completely lost in filter-based measurements.

We're left with considerable uncertainty and a need for more research to really understand the effects of coatings on absorption measured with filter-based instruments. We've added the information about the denuded vs. non-denuded filter-based measurements to the text in section 2.4.1 and have doubled the uncertainty of the PSAP measurements to account for this (see our response to your comment about the coating uncertainty below).

The altering of aerosol absorption is further aggravated by the drying process, which can modify aerosols by removing coatings that are composed of semi-volatile compounds, inorganics, and certainly water.

We also agree that the drying process we use (gentle heating of 40 C or less) may remove some volatile components which could affect both the scattering and absorption measurement. We've added the following text:

"One aspect of the in-situ system that will affect both the scattering and absorption measurement is the gentle heating used to dry the particle to RH<40%. The drying process we use (heating of 40 C or less) may remove some volatile components but we believe the removal to be minimal (<10-20%) based on lab and ambient volatility studies in the literature. Thermal denuder studies suggest little removal of volatile components (<10%) at 40 C (e.g., Mendes et al., 2016; Hakkinen et al., 2012; Huffman et al., 2009, Bergin et al., 1997) although thermal denuders results may be limited by short residence times (<20s). However, smog chamber evaporation studies on ambient aerosol over longer time periods (minutes-hours) at ambient temperature also suggest ambient aerosol may be less volatile than previously thought – Vaden et al. (2011) showed that ambient SOA lost just ~20% of its volume after ~4h."

The authors state on line 406 that "measured absorption Ångström exponents are quite low (close to 1) suggesting little influence of coatings." This is a weak argument, however, because the absorption Ångström exponent is measured after the drying process, which could have easily removed semi-volatile coatings.

Additionally, coated carbonaceous particles of about 0.1 um diameter and a wide range of coating thicknesses can easily have absorption Ångström exponents close to 1, per Gyawali (ACP, 2009, Figure 8).

Figure 8 in Gyawali shows that for a given core size, a wide range of non-absorbing coating thicknesses (from no coating to 4x the size of the core) can have the same AAE (within a factor of 0.25 which is the contour thickness).  This suggests if some (even a lot) of the non-absorbing coating is removed the AAE won't change significantly.  Even in the case of a slightly absorbing coating (Gyawali's Figure 9),  50% of the coating mass would have to be removed to see a significant change in AAE (we estimate that removing 50% of the mass of coating would change the AAE by ~0.25 which is one contour line in the Gyawali's plots).  But we agree with the reviewer that the statement is weak since we know nothing about the coating (thickness, composition, geometry, level of mixing, etc).  With the limited measurements available to us we can't say much more than we already have about the effects of coatings. We've deleted the offending phrase!

Finally, the authors conclude on line 434 that ". . . it is not possible to estimate the actual uncertainty in the in-situ light absorption measurements reported here due to coating effects." Since accurately estimating this uncertainty is not possible, the authors seem to be choosing an uncertainty of zero; I believe that many readers will not be satisfied with this approach.  Hence, I suggest that the authors include a more conservative uncertainty of 50% for the PSAP absorption coefficient (rather than 25%) to account for all possible PSAP errors (you've already included the 50% value in much of your discussion on pages 10-11). This number still comes out of thin air (just like zero), but you can point to the 50% enhancement factor recommended by Bond and Bergstrom (AST, 2006). Thus, I recommend including a comment on line 445 that if the PSAP absorption uncertainty is 50%, the SSA uncertainty at SSA=0.9 is approximately 0.06.

We've redone the uncertainty calculation as suggested.  We've also added this sentence to the discussion of the PSAP uncertainty due to coating in Section 2.4.1:

"To address this, we double the assumed PSAP uncertainty of ~25% to 50% in the calculations of uncertainty."

*Optical and Aerodynamic sizes*
The authors state that their inlet samples particles with aerodynamic diameters less than 7 $\mu$m, and that the particle diameter for 50% sampling efficiency is 5 $\mu$m (lines 196-198). They go on to address this issue on lines 601-617 by using the AERONET size distributions to estimate the fraction of aerosol extinction that occurs at AERONET diameters of less than 5 $\mu$m. By doing this, the authors are assuming that aerodynamic diameter is equal to volume equivalent diameter, which is not generally the case (unless the aerosols are spherical and the aerosol density equals 1). Since: $D_{vol} \alpha D_{aer}/p^{1/2}$ , the authors should use a much smaller cutoff diameter than 5 m in their equation on line 606. For instance, if one assumes  = 2, then Dvol = 3.53 for spherical aerosols; if  = 2:7, then Dvol = 3.04. These smaller cutoff values will decrease their estimated extinction fractions of the particles entering their inlets. However, the authors correctly argue on lines 612-617 that under sampling large particles probably lowers the in situ SSA, and that correcting this artifact will probably make the discrepancy between AERONET and in situ worse. Nonetheless, some recognition of the optical/aerodynamic size difference will indicate to readers that you considered this issue.

This is a good point and we've updated the paragraph in section 3.1.2 (lines 601-617 in original submission) to reflect this. We suggest that the inlet cutoff would be closer to 3 or 4 um. A quick survey of the literature (e.g., Kannosto et al., 2008; Topping et al., 2011; Zhang et al., 2016) suggests a value of 1.5 g/cm3 is probably a reasonable value for ambient aerosol density which would lead to a volume equivalent size cut of 4.1 um (density of ammonium sulfate is 1.77, density of secondary organics is often assumed to be 1.4, water is 1). If we assumed all of the aerosol was AmSulf, the size cut would be 3.75 um. The values we reported in the initial submission were a rough calculation and actually correspond to diameter<4um due to the rather coarse AERONET size bins. We've changed the numbers in the paragraph to reflect a cut size of 3um as a worst case scenario for this effect. The mean extinction fraction shifts down by 1-2% to 0.88+/-0.09 for SGP and 0.93+/-0.07 for BND.

*Tables and Figures*
I like Table 1, but the authors need to include the number of AERONET retrievals and the number of in situ flights that are used to compute these statistics. Standard deviations of the means (SDOM) would be nice, too, so that readers can quickly see that the difference between the averages are statistically significant.

Standard deviations are already included in Table 1. The first line in each row is the median, the second line in each row is the mean and standard deviation of the mean. We've added a final row with the number of retrievals and flights. The number of retrievals and flights is the same for all 3 parameters as we wanted the values to be for the same data points - the values in Table 1 represent the blue points in figures 3-5.
We've also added a Table 1b now which includes the values for the purple points (i.e., AERONET AOD440 > 0.2)

I am having difficulty understanding Table 2. You list 56 retrievals with Level 2 AOD + almucantar at BND, but only 6 retrievals with Level 2 + almucantar + AOD440 > 0:2. The 2[nd] number should be greater than the first number, right?
 (The first number only includes AOD > 0.4, but the 2nd number includes all retrievals with AOD > 0.2). Unless... do you mean "size distribution retrieval" instead of "almucantar retrieval?" The almucantar products include size, AAOD, SSA, and complex refractive index; size distributions are the only Level 2 almucantar products available at AOD(440) < 0.4.

You've kind of answered your question – but clearly we need to be more clear. There are three sections to the table: (1) number of profile flights (one row); (2) AERONET statistics for AOD for various constraints (three rows); and (3) AERONET statistics for AAOD for various constraints (three rows). For the specific values you ask about, the '56' includes observations with AOD<0.4, i.e., it includes V2 Level 2 values where there was a successful almucantar retrieval (of any property) – for these, no AOD threshold is applied. In contrast, the second number '6' represents how many of the 56 points correspond to AOD values >0.2.
To address this we've add further demarcations in the table splitting the 3 sections. We've also added a footnote stating that an almucantar retrieval does not necessarily imply an AAOD retrieval. Finally we've updated the numbers in Table 2 both to reflect the fact that we removed 3 flights at each location due to the potential for aerosol aloft and also so they represent the number of flight matches (rather than overall number of retrievals as there are variable numbers of retrievals/flight)

The legend in Figure 1 is confusing. The legend contains two solid black lines (Direct RF BCFF and Global), but there is only one solid black line in the figure. The legend also contains two variations of dashed black lines (Land and Sea), but there are no black dashed lines in the figure. The figure contains two red dashed lines that are not shown in the legend.
We're sorry if this was confusing – the idea was that the colors indicate the variable ("AOD", "direct RF, all comp" and "direct RF, BCFF", while the line style represented whether it was global, or just over land or sea.  We've remade the legend to make this more clear.

The authors need to explain that "r" represents auto-correlation in Figure 2, as it is not obvious from the caption.
We've augmented the caption as suggested. There is now a caption sentence that says:
"The value r(k) on the y-axis represents the autocorrelation at lag time 'k'."

Also, presumably the correlograms correspond to scattering and absorption coefficients measured with the airborne nephelometer and PSAP; that should also be stated in the caption.
Figure 2 – as stated in the caption - represents the continuous surface measurements at each site not the airborne measurements – it's difficult to look at autocorrelations with non-continuous data – AERONET has lots of gaps due clouds and nighttime and the in-situ profiles are even more 'gappy' than AERONET.

Line-by-line Details
The authors lament on line 27 that the terminology for absorbing aerosols are imprecise, but Andreae and Galencsér (2006) provide precise definitions for soot, soot carbon, brown carbon, light absorbing carbon, elemental carbon, apparent elemental carbon, black carbon, and equivalent black carbon. It would be nice if the community embraced this paper as the "go-to" article for carbonaceous aerosol definitions.
The article cited (Petzold et al 2013) expands upon the important work of Andreae and Gelencsér (2006) and makes specific recommendations for terminology for carbonaceous aerosol as a function of measurement technique. We've added the Andreae and Gelencser reference.

Line 77-78: the authors state ". . . by invalidating low AOD cases, the AAOD values that are retained in the AERONET Level-2 data may be biased high." This is not quite correct. It would be more accurate to state that ". . . the averaged AAODs" or ". . . the climatological AAODs" are biased high.
Changed the sentence to read: "…by excluding low AOD cases, the climatological statistics of AAOD derived from the AERONET Level-2 data may be biased high."
We've also changed a similar sentence that occurs later in the paper (lines 817-818 original submission)

Lines 82, 93: Authors mention several Aerocom models, but do not tell us which ones or which Aerocom experiments they are pulling the data from.

The model information was in the caption of Figure 1 but we've now added the model names to the text and additionally noted that the model simulations were from the AeroCom Pha II control experiments.

Line 185: How much horizontal distance is covered in these 5- and 10-minute flight legs? This is important for readers to understand the auto-correlation within the profiles.

Good point – we've added this to the text. Airplane speed was ~50 m/s resulting in the 10 min upper level legs being approximately 30 km long and the 5 min lower level legs approximately half that (15km) length.

Lines 192-194 states that there are 253 complete profiles at BND and 132 complete profiles at SGP. However, only 24 and 14 flights were used in the analysis of Figures 3-6. I think that it is important to mention this here, because this description sounds like you have a much larger dataset available than you really do. This might be a good opportunity to mention that the stringent AERONET cloud screening procedure drastically reduces the number of possible comparisons, and this must also be considered by modelers when they utilize AERONET retrievals.

The intention was to first separately describe the in-situ and AERONET measurements and then in Section 2.3 we describe the merging of the data sets which leads to reduced numbers of comparisons. We've added a sentence after lines 192-194 (original submission line numbers): "The number of flights that could be compared with AERONET measurements is significantly less than this, as discussed in Section 2.3 where the merging of the AERONET and in-situ data sets is described"

We've also added these sentences to the first paragraph in section 3.1.1:

"The low number of points on Figures 3-5 and in Table 1 indicate both the effects of AERONET stringent cloud screening routine and the constraints imposed by the almucantar retrievals. In addition to limiting the number of comparisons available in this study this limited data availability also has implications for modellers utilizing AERONET data – for example, Schutgens et al. (2016) has shown the importance of temporal collocation in measurement/model comparisons."

Lines 322-325: The authors state:

"Because the profiles are "stair-step" descents from 4600 m asl down to 450 m asl (e.g., see Figure 4 in Sheridan et al., 2012), matching with AERONET retrievals at the end of the profile means that the matches are more closely aligned with when the airplane is in the boundary layer and thus, typically, sampling the highest aerosol concentrations." This is inconsistent with other statements in the paper. The flight takes 2 hours and the authors are matching AERONET retrievals within +/- 3 hours of the end of the profile. Hence, the actual retrieval could occur up to 1 hour before the flight profile begins, and anytime during the flight. Thus, the AERONET retrievals are not necessarily "more closely aligned when the airplane is in the boundary layer.

We've added the following text:

"This way the maximum time difference between the boundary layer portion of the flight and the AERONET retrieval is 3 h; if we'd chosen to match based on the start of the flight the maximum time difference between the boundary layer measurements and the AERONET retrieval could be as large as 5 h."

Line 359: I don't understand how the authors can test $\gamma$ +/-1 standard deviation when they do not measure $\gamma$.
We've added the following text: "As described above, $\gamma$ was calculated from the climatological chemistry measurements made by the IMPROVE network (14 years of data, ~1700 data points at BND; 10 years of data, ~1000 data points at SGP) using the Quinn et al. (2005) parameterization. We calculated the mean and standard deviation of $\gamma$ based on those climatological chemistry measurements."

Lines 360-361: Authors estimate the uncertainty in AOD by varying by up to 2 S.D., but there are hundreds of sunphotometer AODs available during their flights (per figure 3). Why not just compare their in situ AODs to AERONET AODs?
Here we were trying estimate the uncertainty in the AOD specifically due to the relative humidity adjustment. We can't do this by comparing to the AERONET AODs as other factors may also contribute to discrepancies between insitu and AERONET AOD. We have not changed the text.

Line 379: The Lack 2008 citation is not included in the references.
Oops! Added.

Lines 434-435: Authors state that it is not possible to estimate the uncertainty in measured coatings due to coating effects. Consequently, they seem to be choosing zero uncertainty associated with coatings. Many readers will not agree with this choice.
As noted above we've redone the uncertainty calculation including an additional uncertainty to account for the unknown effect of the coating.

Line 516: Authors state that AERONET AOD tends to be higher than in situ AOD in Figure 3; however, the slope is only greater than 1 in one of those figures, although the offset is always positive. So AERONET > in situ AOD is not obvious, especially since many of the regression lines in Figure 3 fall below the 1:1 line. Additionally, AERONET median and average AODs are almost always less than the corresponding in situ AODs in Table 1, indicating that line 516 is not correct for the medians or the averages.
Table 1 has new numbers based on some other analysis and Figure 3-5 are different as well. We've let the sentence stand as it definitely relect the new analysis. Here's a table describing the AOD flight info for the revised paper (it's not included in the paper):

|  | BND | SGP |
|---|---|---|
| RED | #total points=629
**points above 1:1=441**
median aod ratio=1.99 | #total points=347
**points above 1:1=202**
median aod ratio=1.03 |
| BLUE | #total points=21
**points above 1:1 = 16**
median aod ratio=1.22 | #total points=11
**points above 1:1=6**
median aod ratio=1.01 |

Also, please note that the values in Table 1 in the manuscript have changed for two reasons: (1) we are now using the flight averaged AERONET values (2) we eliminated 3 flights from each comparison due to the potential for layers aloft identified using Raman lidar and/or shape of profiles (see details in other responses to reviewer comments).

Line 519: Should be changed to "The correlations improve when subsetted for the more restrictive Level-2 almucantar retrievals." Otherwise, it sounds like the almucantar scans are incorporated into the AOD measurements, which is not the case.
We changed the sentence to read:
The $R^2$ values increase when sub-setted for the more restrictive Level-2 almucantar retrievals.

Lines 541-543:
Authors state: "although the scatter in the relationships (particularly at BND) suggests that a multiplicative factor doesn't represent the relationship very well." I agree that the BND data looks quite scattered, but the SGP values do not look scattered. The R2 values are 0.76-0.77, which is excellent compared to the autocorrelation values for absorption provided in Fig 2 (rabs = 0.45-0.55). Additionally, the scatter shown for AERONET AAOD is deceiving. There are approximately 2 AERONET retrievals for each in situ flight. The in situ flight represents a 2-hour average, but each AERONET retrieval occurs over the period of a 10-minute almucantar scan. The AERONET scans could occur up to 6 hours apart, and up to 3 hours after the flight is complete. Finally, R2 = 0.3 implies that R = 0.55. Thus, the correlation in Figs 4 and Fig 2 at BND are not very different.
The SGP AAOD linear fits have a large y-intercept, which is why we stated that a multiplicative factor doesn't represent the relationship very well.  The BND data in addition to looking less linear than the SGP data also have a large y-intercept.
The reviewer is right that some of the scatter in Fig 4 could be due to our matching criteria and that we are comparing averages derived from 10min almucantar scans with 2h averaged flight profiles.  Both of these issues could contribute to the observed scatter in Figs 3-5 if there are changes in the airmass on short timescales (<1h).   But we thank the reviewer for pointing out that the correlations are not that dissimilar after all – we've now added the following text to the discussion of figure 4:
"Surprisingly, while the BND site has higher 3-hour autocorrelations for absorption than SGP (R = 0.75 for BND and R = 0.55 for SGP, per Figure 2), the results for BND in Figure 4 indicate less correlation than at SGP for absorption. Nonetheless, the correlation coefficients for BND in Figure 4 ($R^2$=0.49 (blue) and 0.37 (red) correspond to R = 0.70 (blue) and 0.61 (red)) are not that far from the 3 h auto-correlation of R=0.75 for absorption at BND in Figure 2."

Also just to comment on the timing of retrievals - at both sites the AERONET retrievals tended to occur during the flights.  At BND the retrievals generally occurred between 1-2 h before the end of the flight (i.e., at the start or during the flight), while at SGP the retrievals generally occurred 40 min before the end of the flight (i.e., around the time the plane was entering the boundary layer).  For flights with multiple retrievals the difference between first and last retrieval was <2h at SGP and typically 3h or less at BND (two flights had a ~5h range).  We haven't added this information to the text though.

Lines 574-576: Well... you don't really have enough data to draw this conclusion, in my opinion. Why not include the averages for the purple points in Table 1, though?

This comment was directed at this statement: "As with Figure 4, the purple points on Figure 5 indicate when the AOD440>0.2; there does not appear to be an improvement in the relationship between in-situ and AERONET SSA when only these purple points are considered."

We've changed the sentence to read: although there aren't enough points to draw a robust conclusion, there does not appear to be an improvement in the relationship between in-situ and AERONET SSA when only these purple points are considered."

We've also added Table 1b which includes the values for the purple points.

Line 579: "Figure 3 shows that the AERONET AOD may be slightly larger than the in-situ AOD." Again, this is inconsistent with Table 1.

We've rewritten the sentence to read:

"Figure 3 shows that the AERONET AOD is similar (SGP) to or be slightly larger (BND)  than the in-situ AOD,"

Line 644: It should be noted that 100 km is a long ways away! What kind of autocorrelations did Anderson (1998) find at those distances? That value should be noted here.

We've added the following text:

For non-plume data sets, Anderson et al. (2003) found autocorrelations $\geq$ 0.8 at 100 km (their figure 6). For plume-influenced data sets they found autocorrelations ~0.6.

Line 673: Authors state that SSA differences are greater than would be expected from random error, even when AOD > 0.4. Why not quantify that? That is, compute the mean and 2xSDOM of all independent measurements in that figure to see if the null hypothesis is valid; add this point with the resulting errorbar to the figure. It is important to isolate the independent measurements, though – multiple values that are obtained within 4-5 hours are not independent measurements (as you argued in Fig 2).

[Figure]

This is the plot the reviewer suggests.  We've put two points with bars representing 2*SD on the plot.  The plot was remade with flight average values for the BND and SGP flights so it doesn't include multiple points for a single flight.  The black diamond represents just the literature studies, while the square represents all the studies (lit+BND+SGP).  If we assume that ∆SSA is normally distributed, we can use the characteristics of the normal distribution to say where ∆SSA is likely to fall.  In the case of random error we would expect the values of ∆SSA to be evenly distributed above and below ∆SSA=0.  However, based on where the standard deviation lines cross the ∆SSA=0 line, for both cases (lit and lit+BND+SGP)  we ~80% of the ∆SSA points will be negative (while random error should lead to only ~50% of the points being negative).  We can also calculate the confidence interval that a value will fall within based on t-statistics.  The t-statistics suggest with 99.9% confidence that the ∆SSA literature values will fall in the interval 0.0 and -0.04 (i.e., the literature mean -0.02+/-0.02.)  Similarly for the all data confidence interval of 0 to -0.12 there's a greater than 99.9% confidence that the data will fall in that range.  We've added the following to the text:

"Figure 6 also shows the mean and 2*standard deviation of all of the points (black square and vertical lines) and just the literature value points (black diamond and vertical lines).  Based on the characteristics of a normal distribution the standard deviation lines suggest ~80% of the points will be negative – random error would suggest only 50% of the points should be negative."

Page 10+: Since the authors are discussing non-synchronized data here, I think that they should note that the models and the in situ flights include cloudy periods, whereas the AERONET data are stringently cloud-screened.
We've added the following text in the second paragraph of section 3.3 before the statistical data comparisons are discussed:
"It should be reiterated here that we are comparing asynchronous data  and that there are some additional differences amongst the data sets that need to be kept in mind:  the AERONET data are rigorously cloud-screened and only obtained during daytime; the in-situ measurements are also daytime-only and the airplane did not fly in-cloud due to FAA flight restrictions, but may have flown near clouds; and the model data include day and night with clouds and also represent values over a 1x1 degree grid."

Lines 830-837: This is a nice approach in some regions, but it is virtually impossible to capture seasonal variability over North America with this approach. For instance, from 4/1994 through 10/2012 the Cart Site had 3 lev2 retrievals in DJF, 22 Lev2 retrievals in MAM, 122 lev2 retrievals in JJA, and 51 retrievals in SON. Thus, results would be skewed to the high humidity summer periods.
We agree that using AERONET to describe seasonality over North America (and specifically over BND and SGP) is limited by frequent cloudiness and/or the cleaner, dryer conditions prevalent over the US Midwest, particularly in the cooler months.  We did note in lines 764-765 of the original submitted manuscript that 'During the cleanest months of the year (December-February) there are none to few Level-2 almucantar retrievals of SSA and AAOD at either BND or SGP.' We've expanded on that statement to mention that lower humidity during the winter also plays a role and pointed to the gray lines in Figure 7ab showing the lack of level2 almucantar retrievals in Jan, Feb, Dec (at BND) and Jan, Dec (SGP).

Lines 840-844: This passage needs some massaging, as it makes no sense to me.
This passage has been re-written and is (hopefully) clearer:

A similar, though statistical, approach was used in Bond et al.'s (2013) bounding BC paper in order to reduce uncertainty and better represent AERONET SSA and AAOD retrievals at low AOD. Bond et al. (2013) worked with AERONET monthly local statistics for the time period 2000-2010. Monthly values of AAOD and SSA at 550 nm were calculated from size distributions and refractive index when there were at least 10 valid inversion retrievals for that month at that site in the 2000-2010 period (most sites had more than 10 retrievals in a given month over the 11 year period). It was assumed in Bond et al. (2013), based on AERONET reported uncertainties, that the retrieved absorption-related values were more reliable at larger AOD and so they made some adjustments to account for this. For each site, AAOD and SSA values were binned as a function of AOD (there were five AOD bins, with each bin corresponding to 20% of the AOD probability distribution).  For lower AOD conditions, the calculated AAOD and SSA values were replaced by values obtained during larger AOD conditions for the same month as follows: (i) the SSA and AAOD values corresponding to $AOD_{550}$ of 0.25 were prescribed for all SSA and AAOD observations at lower AOD and (ii) for locations where all $AOD_{550}<0.25$, the average SSA and AAOD of the upper $20^{th}$ percentile of AOD observations at the site was prescribed for all lower AOD bins.  Finally, the average of all five bins was used to determine the overall monthly average.  In the case of AAOD the bin averages were simply averaged to get the monthly value while for SSA the AOD-weighted bin averages were averaged to get the monthly value.   Note: the $AOD_{550}=0.25$ cutoff point corresponds (approximately) to $AOD_{440}=0.35$ for smaller particles and $AOD_{440}=0.25$ when large particles are present.  This is less strict than the AERONET recommended constraint of $AOD_{440}>0.4$, but it had been suggested $AOD_{440}>0.4$ might be too restrictive (pers. comm., O. Dubovik).

Line 872: Emphasize that this comment only pertains to AOD < 0.2.
We've added the phrase "below $AOD_{440}=0.2$" to the end of the sentence

Line 1125: Müller (2012) is not included in the text. This is an important citation that needs to be presented in the text, as it discusses the difficulties associated with using in situ instruments to measure absorption.
Müller et al. (2012) is included in the text (line 127, original submission); however, Müller et al. (2011) was included in the citations but was not included in the text and should have been.
We've added the following sentence to the third paragraph of section 2.4.1:
"Müller et al. (2011) describe detailed experiments to characterize filter-based absorption instruments and describe some additional limitations of the instruments."

**Reviewer#2**

This paper deals with an important and challenging issue, certainly acceptable for ACPD. The authors argue that the aerosol absorption data most widely used in climate modeling is likely biased high at low AOD, based on coincident and climatological in situ data at two rural sites. I've included some notes below; in summary, there is a lot of good data presented here, but I think the estimates of uncertainty need to be tightened up in order to reach a strong conclusion. Also, evaluating AERONET SSA at AOD below the value they state as the lower limit of quality results is a key caveat, though I agree that the AERONET results are widely applied beyond their stated validity range. Note that this is actually my full review, so it can be considered as part of the formal review process rather than just as a "quick" review for ACPD posting.

We thank the reviewer for the 'quick' review and helpful comments. We've responded below to each of them.

Lines 73-79. This essentially makes the case for selection bias in the AERONET SSA and AAOD values by itself, though I don't think it negates the value of going further and comparing with in situ observations. If the in situ data can show that in general, SSA is lower when AOD is higher, that could make a useful contribution to the argument.

The in-situ data make a useful contribution to assessing AERONET SSA and AAOD regardless of the observed relationship between SSA and AOD. That said, in general at individual sites (at least in the US) the SSA seems to be lower when AOD is lower – for both in-situ and AERONET data.

Line 112. This is supported by the AERONET data themselves. AERONET does not offer global spatial coverage, but it does provide overwhelming evidence AOD_440 is generally <0.4 via direct-sun AOD measurements, which don't suffer from the uncertainties entailed in the model estimates.

We agree with the reviewer that the AERONET data would likely also support our assertion that AOD_440 is rarely greater than 0.4. But the point of our paper is not to quantify rigorously the global coverage of the AERONET Level 2.0 AAOD/SSA retrieval products, so we don't feel that an estimate of the global coverage based on AERONET data would significantly improve the paper.

Lines 242 to 258. As you know, in addition to collocation, the big challenges for this study are probably getting the total column data from the aircraft sampling right, and accounting for the difference between the properties of the ambient particle observed by AERONET and the dessicated ones measured in situ. Assuming that absorbing aerosol is hygrophobic seems a bit risky, especially for an SSA calculation, though this would be less of an issue for cases where the ambient RH is also low.

It is risky, but it is the standard assumption that is made (i.e., in every other direct comparison paper cited in Table 3 and 4), based on limited lab and field data about absorption hygroscopicity. Nonetheless, we also performed a sensitivity test where we assume that the absorption enhancement due to RH is the same as the hygroscopicity scattering enhancement. More details are included in response to the reviewer's comments related to lines 594-596 and we've added the following sentences to section 3.1.2:

"A sensitivity test was performed assuming absorption enhancement due to RH was the same as the hygroscopicity scattering enhancement, i.e., $\sigma_{ap}(RH_{amb})/\sigma_{ap}(RH_{dry})=a*(1-(RH_{amb}/100))^{-\gamma}$. While this is likely an extreme assumption, it had little effect on the comparisons of AOD, AAOD and SSA."

(Do Lines 332-334 raise another question about getting SSA right?)
Lines 332-334 are: "For SSA there appeared to be no correlation between AERONET retrievals and in-situ calculated values regardless of match window length (highest SSA correlation coefficient was 0.12, but most were less than 0.05 for both sites)." We've added the following sentence after that sentence:
"The poor correlations for SSA are not surprising given the uncertainties at low loading."

Ok. I see that you deal with these issues in Section 2.4.1. I'm thinking that the hygroscopicity issue might need a bit more consideration; there does not seem to be a conclusion about the uncertainty in SSA from the in situ observations, and it is not clear whether the general discussion derived from the literature is applicable to the aerosols observed over the AERONET sites in the current study.
We haven't responded specifically to this comment as it seems to summarize the previous several comments which we have responded to.

For the column AOD question, again the discussion does not seem to come to a real conclusion about the uncertainties. Having coincident lidar would help, and this might be available for at least some cases at one or both sites.
We've now looked at the Raman lidar best estimates of aerosol extinction profiles at SGP for the 14 flights with AERONET matches (there is no lidar data available from BND). We found three cases where there appeared to be an aerosol layer in the vicinity of the highest in-situ flight levels, but in each case the profile flight provided a hint of the presence of this layer. Looking at the actual shape of the in-situ profiles, these three flights exhibited a significant increase in measured loading at the highest flight levels. We've removed those flights from the comparisons reported here. There may still be aerosol above the height of the Raman lidar but we have no means for identifying it. Based on the criterion of observing a strong increase in aerosol loading at the highest flight levels, we also removed 3 flights from the set of BND profiles. We've added the following text:
"Although statistical profile results (e.g., Turner et al., 2001; Yu et al., 2010; Ma and Yu, 2014) suggest little contribution from high altitude aerosol layers in the region of these two sites, Schutgens et al. (2016) demonstrates the importance of considering the specifics rather than the statistical. We used the Raman lidar best estimate data product of extinction profiles at SGP to evaluate the presence of aerosol above the highest flight level at the site. For the SGP in-situ profiles that had matches with AERONET inversion retrievals, we identified three lidar profiles that exhibited aerosol layers at high altitudes, but in all three cases the presence of these layers was also hinted at by an increase in the aerosol loading at the highest flight levels of the in-situ measurement. Thus, we further screened in-situ/AERONET comparisons by removing flights at SGP and BND with significant increases in loading at the highest flight levels. There may still be aerosol layers above the level measured by the Raman lidar, but we have no means of assessing that. The AOD comparison presented in Figure 3 suggests we are unlikely to be missing significant aerosol at high altitudes."

Section 2.4.2. There are other possible factors to consider here. For example, the AERONET retrievals report only one pair of (real, imaginary) refractive index values. If there are two or more modes in the column, this assumption will skew the result. You mention the possible surface reflectance contribution to the AERONET AOD uncertainty; there is a paper assessing this which might be worth considering (Sinyuk et al., Remote Sensing Environment 2007, doi:10.1016/j.rse.2006.07.022).

We have no particular insight or expertise concerning the AERONET retrievals, and can only rely on the published uncertainty estimates. If the retrieval experts have not assessed the uncertainty associated with a particular assumption in the retrieval, then we are unable to include that uncertainty in our paper. However, the point the reviewer makes about AERONET retrieving a column RI is a good one and we've added the following to Section 2.4.2: "Another potential issue is that the AERONET retrievals report only one pair of (real, imaginary) refractive index values for the total size distribution (for each wavelength). If there are two or more modes in the column, this assumption may skew the resulting SSA and AAOD values, although the effect of such skewing would depend on the aerosol properties and cannot be assessed here. Potential impacts in the case of uneven mode absorption in the retrieved size distribution have been found to be minor since the retrieved size distribution is more linked to forward scattering than absorption (pers. comm., O. Dubovik)."

Then there is a question about whether the direct sun AOD measurements are used to obtain the extinction in the determination of aerosol absorption properties, or whether the scattering and extinction are both determined from the almucantar scan. In the latter case, the measurement uncertainty will be larger than 0.01 or 0.02, whereas in the former case, heterogeneity could affect the result, as the extinction and scattering data would be taken in different parts of the sky. Either way, the SSA result in most cases would be the small difference between two larger numbers, so accuracy could be an issue.

We are using the reported values of the aerosol absorption properties from the almucantar scans/inversion retrievals and we rely on the published uncertainty estimates for AERONET products. We had helpful discussions with several AERONET gurus (David Giles/Brent Holben) they provided comments to our discussion of the AERONET uncertainties (hence the mention of surface reflectance referred to in the previous comment!).

Lines 503-505. Perhaps the AOD comparisons address the total-column sampling question for the aircraft measurements, in addition to the uncertainty related to the hygroscopicity adjustment and possible large-particle under-sampling. Note that in general, a high correlation does not assure good quantitative values, as might be required for SSA assessment. So, quantitatively, how does this affect the uncertainty in subsequent SSA estimation?

We are not totally clear about what "this" refers to in the question. It could be "total-column sampling", or "hygroscopicity adjustment", or "possible undersampling". We can (and did) assess the uncertainty in the SSA derived from our in-situ measurements for all of these issues, but rely on published uncertainty estimates for AERONET products. However, we think the reviewer is referring to the effects of quantitation vs correlation.  Figure 3-5 (for 440 nm) include an indication of both the in-situ and AERONET uncertainties.  For AOD we see that those uncertainty estimates cross the 1:1 line for almost all cases (red or blue) and definitely for all the blue cases at both sites.  This suggests that the in-situ measurements provide a reasonable representation of the total column aerosol loading as represented  by AERONET and student t-tests at the 95% level support this.  In contrast, for AAOD and SSA the uncertainty bars don't cross the 1:1 line for any of the measurement comparison points at SGP and for only do so for a small subset of the comparison points at BND.  Student t-tests on the AAOD and SSA data suggest the AERONET and SSA values are different at the 95% level.  We've added comments about the uncertainty bars and student t-tests in the discussion of each figure.

Line 561. Again, it is not clear how much the measurement uncertainty contributes to the discrepancies between in situ and AERONET AAOD.
Uncertainty doesn't contribute to the discrepancies, but rather provides the means for assessing the significance of the discrepancy.   As we note in our response to the previous comment, the uncertainty bars for AAOD and SSA suggest that even taking into account the uncertainty estimates for the measurements there are very few points (and only at BND) that overlap the 1:1 line.  This suggests that there is a significant discrepancy between the in-situ and AERONET AAOD (and SSA) measurements that we don't see in the AOD comparison.

Line 567-568. Does this mean the in situ measurements are missing the extremes, either due to sampling, or to perhaps to conservative estimates of the hygroscopicity effect?
We don't think the in-situ measurements are missing the extremes.  The aircraft results are very consistent with the long-term surface measurements at both sites which show much less variability in SSA than is obtained from the AERONET retrievals (e.g., Sherman et al., 2015).  Figure 3 in Andrews et al. 2004 shows a comparison of the scattering at the lowest flight leg at SGP with the surface scattering measurements for a 2 year time period suggesting the aircraft is capturing the overall variability at the site…at least over the vertical range the aircraft samples at.  We've also updated figure 8 in the paper to show the surface SSA data adjusted to ambient conditions for better comparisons with the ambient SSA values from the airplane and AERONET.  The ambient-adjusted SSA from the continuous surface measurements (day/night, 1 min frequency, more than 15 years of data) shown in figure 8 is very similar to the SSA from the aircraft.

We've added the following sentence to the first paragraph of Section 2.1:
"Previous work has shown that the airplane measurements appear to capture the variability in aerosol properties observed by the long-term, continuous measurements at the surface (e.g., Figure 3 in Andrews et al., 2004)."

It is unclear whether assuming constant hygroscopicity fit parameters (that are used in conjunction with the variable ambient RH) will narrow or expand the variability of the calculated SSA.  The discussion of the 'SSA$_{hybrid}$' (SSA$_{hybrid}$=(AOD$_{AERONET}$–AAOD$_{PSAP}$)/AOD$_{AERONET}$) in the new last paragraph of Section 3.1.1 and now included on Figure 5 provides some additional thoughts on this.  (Calculation of SSA$_{hybrid}$ was proposed by Reviewer#1)

Lines 594-596. Right. But this does not address whether the underlying assumption that absorbing particles are non-hygroscopic is valid. If the absorbing species are OC rather than entirely BC, one might expect at least some hygroscopic growth is possible. And I think you concluded earlier that there must be something like OC, at least at one site.

As we've noted elsewhere in the manuscript there are VERY FEW studies (ambient or lab) investigating water uptake by absorbing aerosol and those that exist tend to suggest that water uptake is minimal.  We have no data to assess the underlying assumption that the absorbing particles are non-hygroscopic. Since we expect that the aerosols at both sites are likely to be well-aged and internally-mixed, it is possible that the absorbing particles are hygroscopic, but we don't know the extent to which it would affect the absorption coefficient.  At SGP, Sheridan et al (2001) showed that the aerosol hygroscopicity decreased in the presence of aerosol thought to contain dust or smoke.

There is organic at both sites – the IMPROVE measurements suggest 30%+/-13% OC at BND and 40%+/-14% OC at SGP for sub1um aerosol.  Parworth et al., 2015 suggests anywhere between 25-75% organic at SGP for non-refractory portion of the1um depending on season.  But the hygroscopicity and absorbing nature of that organic aerosol has not been assessed.  The parameterization for hygroscopicity that we use (from Quinn et al., 2005) was derived using tandem nephelometer measurements of hygroscopicity on ambient aerosol (i.e., both scattering and absorbing aerosol) and the measurements of aerosol chemistry (specifically organic carbon (OC) and sulfate (Sulf)).  The observed hygroscopicity (scattering as f(RH)/scattering_dry) was shown to decrease as the organic mass fraction (defined by Quinn as the ratio of OC/(OC+Sulf)) increased.  This is a simple parameterization and does not account for all the individual chemical species which may influence water uptake nor does it account for interaction between absorbing species and water.  We've added the following text to the sentence describing the parameterization to make this a little more clear:

 "Climatological IMPROVE network surface aerosol chemistry measurements of sulfate and organic carbon (Malm et al., 1994) were utilized to determine a value for the hygroscopic growth parameter 'γ' for each site based on the Quinn et al. (2005) parameterization which relates aerosol hygroscopicity to organic mass fraction."

We've also done a sensitivity test to see how figures 3-5 would change if we assumed that the absorption enhancement due to RH was the same as the scattering enhancement due to RH.  This assumption has little affect on the AOD comparison (in-situ absorption is only ~10% of in-situ extinction).  The slopes in the AAOD comparison decrease by ~30%, but the AERONET AAOD values are still predominantly and significantly above the 1:1 line (i.e., all points at SGP and all but 3 points at BND are above the 1:1 line).  At both sites the SSA values shift slightly closer to the 1:1 line; at BND 19 out of 21 AERONET SSA points are below the 1:1 line and at SGP all the SSA points are below the 1:1 line.  We've added the following sentences to the text: "A sensitivity test was performed assuming absorption enhancement due to RH was the same as the hygroscopicity scattering enhancement, i.e., $\sigma_{ap}(RH_{amb})/\sigma_{ap}(RH_{dry})=a*(1-(RH_{amb}/100))^{-\gamma}$. While this is likely an extreme assumption, it had little effect on the comparisons of AOD, AAOD and SSA."

Lines 614 to 617. Does this call into question whether the in situ measurements adequately sample the entire column observed by AERONET? I'm thinking Section 3.1.2 does not put to rest the question in the title of this section. So I'm uncertain whether you have established the conclusion stated in Lines 625-627, though I think AERONET might overestimate absorption in many cases, due to the way they relate the measured extinction and scattering in order to derive absorption.

*The 'this' referred to by the reviewer is:*
*"The in-situ measurements would need to preferentially under-sample absorbing aerosol relative to scattering aerosol in order to come into line with the AERONET observations."*
*Section title is: "3.1.2 How might AOD discrepancies affect SSA and AAOD comparisons"*
*Conclusion sentences (which actually start next section) are:*
*"Direct comparisons at BND and SGP suggest that AERONET retrievals underestimate SSA and, consequently, that AERONET overestimates AAOD relative to in-situ measurements of AAOD for the low AOD conditions typical at these two sites."*

We've tried to address the limitations of the in-situ measurements as best we can. Given that we do fairly well in the AOD comparison we don't think we are missing a significant amount of the aerosol. We can see two ways that the in-situ measurements would collect enough scattering aerosol to simulate the AERONET AOD but miss absorbing aerosol:
(1) not accounting properly for the effect of coatings (organic or water) on absorption enhancement which we've discussed in detail in the manuscript
and
(2) not sampling layers of predominantly absorbing aerosol below, between, and/or above the flight layers. These layers couldn't have much scattering associated with them or they would affect the AOD comparisons. Weigum et al (2012) do report on BC plumes over the remote Pacific although they don't comment on the aerosol scattering associated with these plumes and the levels of BC in the plumes they observed are significantly (factor of 10 or more) lower than what would be needed to bring the in-situ AAOD up to the level of the AERONET AAOD. We've added the following text:
"In summary, we can only see two ways that the in-situ measurements can sample aerosol efficiently enough to represent AERONET AOD fairly well but significantly underestimate AAOD and overestimate SSA: (1) not accounting properly for the effect of coatings (organic or water) on absorption enhancement which we've discussed in detail (e.g., see Section 2.4.1) and (2) not sampling layers of predominantly absorbing aerosol below, between, and/or above the flight layers. We suspect that the SSA required of such layers in order to explain the AAOD and SSA discrepancies is physically impossible."
Note: We've also changed the title of section 3.1.2 to:
"*How might in-situ hygroscopicity assumptions and under-sampling of the aerosol affect SSA and AAOD comparisons?* "

Lines 689-690. This might be stated differently, as it assumes no systematic underestimation of absorption for the in situ measurements.
We've re-written the entire paragraph to be a bit more even-handed:

"In summary, the literature survey featuring measurements across the globe for many aerosol types suggests that even at higher AOD conditions, direct comparisons of AERONET with in-situ aerosol profiles find that AERONET column SSA is consistently lower than the SSA obtained from in-situ measurements (although often within the combined uncertainty of the AERONET SSA retrieval and in-situ measurements).  If there was no consistent bias in the AERONET/in-situ comparison we would expect (AERONET_SSA – INSITU_SSA) to be evenly distributed around zero.  Instead, Figure 6, which summarizes the literature survey, suggests either that AERONET retrievals are biased towards too much absorption, or that in-situ, filter-based measurements of aerosol absorption are biased low. We note that the results from the literature indicate that the hypothesized low-bias in in-situ absorption is not associated with a single airplane's measurement system or the atmospheric conditions encountered in a single experiment. That leaves us with possible bias in the in-situ experimental methods (instrument issues (nephelometer, PSAP), treatment of f(RH), vertical coverage, sampling artifacts), all of which we have attempted to address above."

We've also come up with a different title and edited sentences throughout the manuscript that suggest the only bias may be with AERONET retrievals.

**Reviewer#3**

We appreciate the reviewers detailed reading and commenting on the manuscript and hope we have address the concerns raised.

General comments: The authors present data of situ measurements from aircraft profile flights from which calculations of AOD, single scattering albedo (SSA), and Absorption Aerosol Optical Depth (AAOD) are compared to remote sensing measurements (of AOD) and retrievals of SSA and AAOD from AERONET sun-sky radiometers. These comparisons are made for two sites in the USA and for primarily low AOD levels, mostly less than 0.25 at 440 nm. This is well below the AERONET recommendations for use of absorption parameters from their retrievals (>0.4 at 440 nm is recommended), and although the authors discuss this in the text this recommended low AOD threshold is conspicuously absent from both the Abstract and Conclusions sections (and this needs to be remedied).

We've re-written the abstract and conclusions to reflect these points.

The authors state that in prior publications "... the in-situ derived AOD values tend to be slightly lower than the AOD retrieved from remote sensing measurements." They fail to point out that the in situ measurements rarely if ever measure the total column AOD, which includes both mid- to upper-tropospheric aerosol plus stratospheric AOD. The authors should include some references and discussion on the AOD that is not measured by in situ instruments in the upper troposphere and stratosphere since the aircraft do not fly complete profiles from the surface on upward into the stratosphere.

Not covering the entire column is indeed a limitation of all aircraft measurements.  We've added some additional discussion of this issue in section 2.4.1.  Additionally, we've also now used SGP Raman lidar data and also assessed the shapes of the profiles to better account for aerosol above the highest flight level of the aircraft.  We've also added altitude ranges and information on how each campaign dealt with aerosol below (and above in the case of Magi et al. 2005) their flight profiles if that information was provided.  Please see our responses to the specific comments related to this issue below.

Discussion of the fact that the aircraft profiles presented (with 4.2 km above ground level as the maximum in situ sampling altitude) do not actually measure the total atmospheric column AOD needs to be included in this manuscript. Therefore differences in in situ versus AERONET AOD are indeed expected and the AOD would be expected to be somewhat higher for sunphotometer total column AOD than for in situ in most aircraft sampling strategies. Moreover lidar measurements sometimes show mid to upper altitude aerosol layers that this aircraft sampling strategy (max at 4.2 km agl) would not measure.

See our response to the previous comment.

We should note that the AERONET and in-situ AOD are in fair agreement, whereas the AAOD comparisons look much, much different. Suppose the AOD discrepancy were entirely due to particles above 4.2 km agl – what SSA would those particles need in order to eliminate the AAOD discrepancy? We suspect that the required SSA is physically impossible, which means that missing particles can't explain the AAOD discrepancy.

Additionally, Sunphotometers in general and AERONET instruments in particular measure AOD more directly than any other technique and as such these data are considered by the scientific community to be the gold standard of accurate AOD measurement for the total atmospheric column. AERONET measured AOD represent the ambient aerosol optical properties and do not have to be corrected for RH humidification growth effects, loss of large particle sampling, etc. as is required and/or discussed for in situ data utilized in this paper. Nyeki et al. (2012) found that AERONET measured AOD agrees very well with other well-calibrated sunphotometers. We apologize if any part of this paper came across as questioning AERONET AOD measurements. We recognize them as the gold standard for AOD and indeed our NOAA colleagues making solar radiation measurements have discussed this (e.g., Augustine et al., 2008)

At the Davos, Switzerland site the comparison of the time co-located and matched 500 nm AOD differences between AERONET and GAW-PFR from 2007 through 2010 resulted in a mean AOD difference of -0.0024 and a root-mean square error of 0.0071. These issues should be included in the discussion on AERONET data, and in the section on comparison of AOD from AERONET measurements to in situ estimates. Accuracy of AOD is very important in this paper as AAOD is derived from AOD values and the AOD values derived from aircraft profiles (after corrections to make ambient estimations) and also from models (such as within AEROCOM) can be either biased or have significant uncertainties.

First, we'd like to correct a possible misunderstanding by the reviewer. The in-situ AAOD values are NOT derived using the in-situ AOD values. The in-situ measurements include a separate measurement of aerosol absorption and that absorption is what we integrate over the vertical range to calculate the in-situ AAOD value. Figure 3 was included to show that we can use the in-situ measurements to estimate AOD reasonably well.

As we say above, we recognize that AERONET is a gold standard for AOD measurements and have already noted the standard reference for AERONET AOD uncertainties in the text (e.g., Eck et al., 1999) as advised by our communications with the NASA AERONET scientists. The uncertainty of 0.01 for AERONET AOD is the same as the uncertainty Nyeki et al. (2012) report for the PFRs: "The combined uncertainty related to instruments and retrieval algorithms is estimated to result in an AOD uncertainty <0.010 at $\lambda = 500$ nm.". The AERONET AOD uncertainties are certainly less than those for in-situ AOD. The uncertainties in other variables (e.g., SSA and AAOD from both AERONET and in-situ measurements) are the important ones to consider because they are much larger than the AERONET AOD uncertainties.

In case the title of section 3.1.2 was confusing to the reviewer we've changed it to:

"3.1.2. *How might in-situ hygroscopicity assumptions and under-sampling of the aerosol affect SSA and AAOD comparisons?*"

Furthermore, in order to better understand the comparisons of aircraft profiles to AERONET measurements a scatterplot of computed Extinction Angstrom Exponent (EAE; 440 - 675 nm) estimated from the aircraft data versus AERONET measured EAE needs to be added to Figure 3. This comparison of EAE is pertinent to the discussion in the current text of aircraft inlet sampling issues regarding possible large-sized particle losses.

Esteve et al. (2012) presents a plot of airplane column Angstrom exponent vs AERONET Angstrom exponent for BND which shows the AERONET Angstrom exponents to be consistently lower than the airplane column Angstrom exponents (med_aeronet=1.53, med_airplane=1.82. Andrews et al., (2011) provides a statistical comparison of Angstrom exponent (their figure 3) from the airplane and AERONET at SGP and there's a similar offset of ~0.3 between AERONET and in-situ Angstrom exponent with AERONET being lower. Delene&Ogren, 2002 (their Fig 9b) shows that a difference of 0.3 in SAE corresponds to a difference of about 0.05 in submicrometer scattering fraction. For BND, this means that the supermicrometer scattering fraction might drop from 0.26 to 0.20, i.e., a 25% loss of supermicron-mode scattering. But since supermicron scattering is only about 20-25% of the total, losing 25% of the supermicron-mode means only a 5% loss in total scattering. This indicates that possible losses of supermicrometer particles has a minor effect on the in-situ AOD. We've also used the AERONET size distributions (lines 604-617 original manuscript) to evaluate super micron particle undersampling – the AERONET size distribution analysis suggests a 5-10% loss of total extinction.

[Figure]

BND angstrom comparison                SGP angstrom comparison

We've included the Angstrom exponent plots here for the reviewer, but as versions of them appear in other papers we have not added another figure to this manuscript.

There is a lack of discussion in the paper of how the uncertainties of the in situ measurements change as aerosol concentrations decrease. All measurement methodologies suffer from issues related to a decrease in signal at low concentrations (relative to potential instrumental noise and offsets), therefore I think that discussion of how the in situ measurement uncertainty changes with aerosol concentration is a very important aspect that needs to be included in the manuscript. Since the paper focuses primarily on low AOD cases, this is a critical issue that is surprisingly neglected in the current manuscript.

While BND and SGP are termed low loading sites in terms of their AERONET AOD climatology, the boundary layer aerosol loading is not typically low enough to significantly impact the uncertainty in the in-situ measurements. The uncertainty as a function of loading and averaging time for the in-situ measurements has been discussed in detail in many previous publications (e.g., Table 2 in Sheridan et al 2002; Table 2 in Andrews et al., 2011; supplemental materials of Sherman et al., 2015). We already provided those references in the manuscript and have used their methodology to determine the uncertainty values reported here.  For example, we state in the first paragraph of the in-situ uncertainty discussion:

"Sheridan et al. (2002) calculated uncertainties in aerosol light scattering for the TSI nephelometer to be 7-13% for 10 min legs depending on amount of aerosol present – the higher uncertainty value applies to very low aerosol loadings (scattering < 1 Mm$^{-1}$)."

We've now added the following text to the in-situ uncertainty discussion:

"For the higher altitude flight segments the loading does tend to be quite a bit lower and thus has higher uncertainty but those upper-level segments contribute little to the overall AOD or AAOD. Because the flight column SSA is calculated using extinction-weighted SSA flight segments, segments with very low aerosol concentrations will have little impact on the column SSA derived from the flight measurements."

Additionally it is necessary to summarize in the text a description of the methodology used for computation of profile weighting of the in situ SSA estimates during each aircraft flight. Are these SSA values at each altitude weighted by the extinction coefficient at that altitude, thereby effectively giving higher weighting to the measurements at altitudes that had the highest aerosol concentrations? The AERONET retrievals of SSA are effective optical extinction weighted values for the total atmospheric column, therefore extinction weighting of the in situ data would be the most rigorous way to compare similar quantities.

We've updated the description of how flight column SSA was calculated:

"As described in Andrews et al. (2004), the in-situ column SSA (which is compared to the AERONET SSA value in section 3.1) was calculated for each flight level and then extinction-weighted and integrated to determine column SSA.  This results in SSA values which are virtually identical to SSA values calculated using: $SSA_{col,in\text{-}situ} = (AOD_{in\text{-}situ} - AAOD_{in\text{-}situ})/AOD_{in\text{-}situ}$) and effectively gives higher weighting to the SSA values at altitudes that had the highest aerosol concentrations."

The authors also need to show plots of the in situ aircraft measured/computed SSA altitude profiles to show how SSA varies as a function of altitude for several days of varying AOD magnitude. This is important as it can provide some needed information on how the in situ SSA measurement data look at very low concentrations, especially higher than 3 km above ground level on very low AOD days and also on some moderately high AOD days.

Examples of profiles of multiple variables including SSA are presented in Figure 2 of Andrews et al. (2004).  The AODs aren't noted in the text of Andrews et al (2004), but Figure 2a corresponds to an AOD_440 ~ 0.15 , Figure 2b corresponds to an AERONET AOD_440 of ~1.0 and Figure 2c corresponds to an AERONET AOD_440 of ~0.3.  Box-whisker statistics for various parameters (including SSA) for the flight profiles can be found Andrews et al. 2004, Andrews et al., 2011a and Sheridan et al 2012.

Additionally, profile plots of various parameters (including SSA) for each individual SGP flight can be found here:  http://www.esrl.noaa.gov/gmd/aero/net/iap/iap_profiles.html

And for the first two years of the BND flights can be found here:
https://www.esrl.noaa.gov/gmd/aero/net/aao/aao_prof2007.html

As these individual profile plots and statistics on the profiles are available in other locations, we have not included them here. We've added a sentence mentioning the availability of these plots in other locations:
"Profile statistics for various parameters including SSA are provided in Andrews et al. (2004, 2011a) and Sheridan et al. (2012). Individual flight profiles for various parameters are available online at: http://www.esrl.noaa.gov/gmd/aero/net/iap/iap_profiles.html (for SGP) and https://www.esrl.noaa.gov/gmd/aero/net/aao/aao_prof2007.html (for BND)."

In the abstract you state: "The tendency of AERONET inversions to overestimate absorption at low AOD values is generally consistent with other published comparisons." However the published comparisons between AERONET retrievals and in situ measured SSA shown in Figure 6 are not for low AOD (the AOD are moderate to high in the Figure) and also the SSA differences are generally within the combined uncertainty estimates of the two different techniques (see numerous additional comments on Figure 6 data below in 'Specific Comments'). Since it has never been established that the in situ measurements of SSA have no bias of their own then it is not possible to say that the AERONET measurements of SSA at moderate to high AOD are biased since there is no absolute benchmark for comparison purposes.
In order to more accurately reflect Figure 6, we've rephrased the sentence in the abstract to read: "The tendency of AERONET inversions to overestimate absorption at low AOD values relative to the in-situ measurements is generally consistent with other published comparisons across a range of locations, atmospheric conditions and AOD values." We've now noted in the abstract that the comparisons tend to fall within the reported uncertainty range. We've also rephrased the comments about bias to note that the in-situ measurements could be biased low. We feel it's important to note here that we do have absolute benchmarks for the accuracy of in-situ measured scattering ($CO_2$) and absorption (various, PAS uses molecular absorption or scattering, EXT-SCA uses physical length). AERONET's absorption products lack such absolute benchmarks. However, we do not have characterization of bias vs random error in those benchmarks and our instruments that are referenced to those benchmarks. So our end conclusions are (a) that either AERONET overestimates absorption or INSITU underestimates it, and (b) there is bias in one or the other or both, because the comparisons in Fig 6 are not symmetrical about the "no-error" line.

Additionally since you state that the science is unclear on absorption enhancement due to coated absorbing particles (Section 2.4.1, line 404-407) you need to give a detailed explanation as to how this unknown factor was incorporated into the uncertainty estimates you made for the in situ measured single scattering albedo (it seems to have been ignored in your estimates). The aircraft sampled aerosols are dried first and therefore true atmospheric ambient state aerosol optical properties are actually not measured directly during the profiles. This is important regarding your claim of relative bias of single scattering albedo from one measurement type versus another since you cannot rigorously state (or indirectly suggest) that the aircraft measurements of single scattering albedo are unbiased given that the ambient state optical properties of the aerosols were not directly measured by your in-situ instruments.

There are studies (Lack, Cappa) that suggest coatings cause the PSAP to overestimate absorption, and numerous studies that suggest that coatings enhance absorption of suspended particles. If those coatings are lost or evaporate in our sampling system, then we would expect PSAP to underestimate absorption. As a result, we cannot treat the effects of coatings as a clear bias, as they could enhance or reduce the absorption measured by the PSAP. We should also point out that the particles are not completely dried or dessicated by the sampling system on the airplane. The heater only supplies enough heat to reduce the RH to 40%.

Based on the recommendation of another reviewer we've doubled the PSAP uncertainty to account for the effect of coatings, since the coating enhancement is unknown. We've added the following sentence in Section 2.4.1 when discussing the absorption enhancement:
"To address this, we double the assumed PSAP uncertainty of ~25% to 50% in the calculations of uncertainty."

The climatological comparison of in situ and AERONET values in Figure 7 is a very important figure in this paper. This figure suggests that if AERONET data are wisely utilized (as done by Bond et al. (2013), for example) then the SSA differences between the two methodologies can be relatively small. The large differences in the time matched data at Bondville and SGP sites from in situ flights and the AERONET retrievals shown in previous figures (Figs. 4-6) are not nearly as evident in Figure 7, especially for the SGP site. There is a surprising lack of discussion of this apparent discrepancy between the matched aircraft profile/AERONET data and the 'climatological' comparisons and the reasons for it. There is also a surprising lack of emphasis on the SSA comparison results shown in Figure 7 in the Abstract and Conclusions given the importance of this result.
We agree with the Reviewer that Figure 7 shows good agreement of monthly medians of SSA between AERONET Level 2.0 SSA and INSITU measurements and that is already stated in the text. This comparison is subject to considerable sampling bias, however, as we note in the discussion of the figure that the AERONET Level 2.0 almucantar data are restricted to more polluted cases with AOD440>0.4. Directly comparing the climatological AERONET Level 2.0 SSA with INSITU measurements requires an implicit assumption that SSA does not show a systematic co-variance with AOD, which does not seem to be valid (e.g., for in situ data sets: Delene and Ogren, 2002; Andrews et al., 2013; Pandolfi et al., 2014; Sherman et al., 2015 and for North American AERONET data sets: Schafer et al. (2014; their figure 6) and our own analysis as described in the text (Figure 8 and lines 855-861 of original submitted manuscript)). As a consequence, the combined results of Figure 7 and Figure 8 do not suggest that a "wise" utilization of AERONET data can minimize the differences between the two methodologies if "wise" implies the Bond et al., 2013 methodology of using SSA from high loading events and applying it to low loading conditions. Our Figure 8 suggests that a global climatology based on SSA measured at high AOD will lead to an underestimate of the global average AAOD. We've added the following text to the discussion of Figure 8:
"This relationship implies that a global climatology based on SSA measured at high AOD will lead to an underestimate of the global average AAOD."

Additionally it is very interesting that in Figure 8 the in situ surface measurements of SSA agree quite well with AERONET retrieved SSA for both sites, with excellent agreement at SGP site and within uncertainty bounds for the BND site except for extremely low AOD of less than 0.05. We've replaced the dry surface measurements previously shown in figure 8 with those same surface measurements adjusted to ambient humidity using the hygroscopic growth parameterizations that were applied to the aircraft measurements. The surface ambient RH measurements used in the adjustment came from DOE/ARM at SGP (2m ambRH) and from NOAA/GMD at BND (10m ambRH). We did this so that shape of the three curves and the SSA values are more directly comparable. We've updated the figure caption and the paragraph describing figure 8 to reflect this change (lines 862-870 in original submission). We've also adding the following text:

"The AERONET SSA values are also lower than the surface in-situ SSA values – the surface in-situ SSA values adjusted to ambient conditions are quite similar to those obtained from the in-situ vertical profiles."

The authors have stated that the RH for the surface measurements are all <40%, although this is somewhat surprising given that the surface RH is typically >40% at this location, or perhaps measurements are never made when RH exceeds 40%? Or maybe the surface data that are shown are for the dried aerosol only? If so, you should apply the same humidification factors to the surface in situ data that you have applied to the aircraft profile data in this analysis to make the comparisons in Figure 8 consistent. A look at climatological data for Enid, Oklahoma and Ponca City, Oklahoma (same region as the SGP site) show daily average surface minimum RH of >40% and average Maximum RH of >75-80% for almost all days of the year. The surface in-situ measurements are made at RH<40% for consistency with the GAW program protocols. As we note in the manuscript (original submission, lines 867-870) "…adjustment of the surface measurements to ambient conditions would tend to shift the SSA values upward (assuming absorption is not affected) and the scattering values to the right but would not significantly change the shape of the curve". We've now provided the surface data adjusted to ambient conditions so the shapes of the three curves and the SSA values are more directly comparable (see response to previous comment).

The surface in situ measured SSA to AERONET retrieval comparison result may be particularly interesting since the aerosol concentrations are often highest near the surface and therefore the in situ measurements made at the surface should have less uncertainty than those made at high altitudes where the concentrations may be very low. The authors should also present a comparison of the SSA and Extinction Angstrom Exponent measurements made at the lowest flight altitudes during profiles to those made at the surface by similar in situ instrumentation to show how good the agreement is between these measurements and to prove that the aircraft inlet sampling issues mentioned in the manuscript do not result in significant measurement uncertainties. These comparisons are shown in Andrews et al (2004) and Sheridan et al (2012). Further, many of the sampling issues (RH adjustment, size cut, discrete flight levels) are discussed in detail in Esteve et al., (2012) as mentioned in the text (see for example lines 447-468 in the original submitted manuscript). For example, Sheridan et al (2012) shows plots of surface measurements versus lowest level flight leg at 157 m agl. Their plots represent 5-min AAO low-level flight segment averages over the BND site vs. two-hour BND surface data centered on the flyby time.  They show a slope of 0.87 for sub10um surface data vs the aircraft and a slope of 0.97 for sub1um surface data vs the aircraft.  This suggests that the airplane measurements are capturing virtually all of the submicron aerosol but could be missing 10-15% of the super micron aerosol For scattering Angstrom exponent and SSA the slopes are 0.92 and 0.99 respectively. The airplane scattering Angstrom exponents are actually slightly smaller than the surface scattering Angstrom exponents which is the opposite of what might be expected.  Andrews et al. (2004) show that SGP for an earlier version of the inlet with a 1um size cut the surface vs lowest level flight leg slopes were 1.02, 1.04 and 1.00 for sub-1um scattering, scattering Angstrom exponent and SSA.

Specific Comments:
Abstract: You also state: "We conclude that scaling modeled black carbon concentrations upwards to match AERONET retrievals of AAOD may lead to aerosol absorption overestimates in regions of low AOD." This statement is somewhat simplistic and mis-leading since it does not reflect the much better comparisons shown in Figure 7 for 'climatological' analyses. It also ignores the well thought out application of the use of AERONET retrieved SSA values as weighted by higher AOD observations and then applied to highly accurate AOD measurements at all AOD levels from AERONET, similar to the approach of Bond et al. (2013).
We've re-written the abstract significantly.

Introduction (lines 77-79): You state: "Moreover, by invalidating low AOD cases, the AAOD values that are retained in AERONET Level-2 data may be biased high." Again, it is misleading and simplistic to suggest that careful investigators would take the AAOD values from only Level 2 data and assume that they can be utilized as is. Many researchers have already utilized a much more intelligent approach: first estimate SSA at higher AOD from AERONET, and then apply those values to ALL levels of AOD (see Bond et al., 2013). I suggest that you remove or modify this sentence.
This sentence has been modified as suggested by Reviewer#1.
Changed the sentence to read: "…by excluding low AOD cases, the climatological statistics of AAOD derived from the AERONET Level-2 data may be biased high."
We've also changed a similar sentence that occurs later in the paper (lines 817-818, original submission).

Introduction (lines 142-143): I assume you mean Dubovik et al. (2000). Dubovik et al. (2002) is not in the reference list.
Correct.  Fixed.

Section 2.1 (lines 185-187): Please elaborate what you mean by improving measurement statistics here. It would seem that the aircraft instruments 10-minute sampling rate at higher altitudes is an attempt to overcome issues associated with low aerosol concentrations and associated limits of instrumental sensitivity. Therefore on very low AOD days it would seem that an even longer time interval than 10 minutes would be justified. Please elaborate on the sampling strategy and state whether it was modified for very low aerosol concentrations (very low AOD days).

The sampling strategy was the same regardless of loading.  It is described in Andrews et al (2004), Andrews et al (2011) and Sheridan et al (2012).  We updated the sentence about improving statistics to read:

"…in order to improve measurement statistics at the typically cleaner higher altitude flight levels."

We've also added the following sentence in the first paragraph of section 2.1 (actually now the second paragraph – we split the first paragraph into two):

"The pilot flew within the constraints provided (specifically-defined stairstep profile, vary the time of day, cross wind, over the instrumented field site, during daylight and not within clouds) but without day-to-day scheduling input from scientists."

Section 2.1 (line 194): "Only complete profiles were used in this analysis." Please state here that complete profiles as made by the aircraft do not equate to complete atmospheric profiles. None of the aerosol from 4.2 km agl through the stratosphere is sampled in the flights. Especially for very some low AOD days and (and also for some moderate-high AOD days in summer with strong convection) it is expected that a significant amount of the AOD actually occurs above 4.2 km agl. These upper aerosol layers that are often seen in lidar data may have different optical properties than lower altitude aerosols.

We've clarified that sentence and added a second sentence:

"Only complete profiles (all 10 flight levels) were used in this analysis. As is obvious from the vertical range of the flight levels, complete in-situ profiles do not equate to complete atmospheric profiles – this is discussed more in the in-situ uncertainties discussion (Section 2.4.1).

In the in-situ uncertainties section (section 2.4.1) we now discuss in greater detail the fact that the aircraft does not cover all the way up to the stratosphere.  There is no lidar data available for BND, but we did retrieve the Raman lidar best estimate data product for SGP for the direct flights and compared the lidar extinction to the extinction obtained from the in-situ profiles. There were three SGP flights we removed from the comparison based on that analysis.  We also took a harder look at the BND profiles and removed profiles that appeared to have increasing extinction at the highest flight levels as this was the smoking gun in the lidar comparisons for SGP.  This is now discussed in more detail in the text of section 2.4.1.

Section 2.1 (line 216-219): Discuss how assuming a constant hygroscopic growth parameter would cause uncertainties when seasonal variation in aerosol type exits. Especially in spring, aerosol type may include biomass burning (crop waste or grass burning) and also dust from the Great Plains region (see Ginoux et al. (2010)), plus pollen from grass and trees.

It turns out that we'd been exploring this concept – we'd forgotten that we'd turned off the hygroscopicity adjustment for one BND flight because the hygroscopicity adjustment resulted in the flight's ambient in-situ AOD being ~2 times higher than the AERONET AOD (the value w/o hygroscopicity correction was within 0.01 of the AERONET AOD).  This is the very high AOD point for BND (Blue point on fig 3, now labeled BB, with AERONET AOD440~0.5) and was associated with smoke from wildfires in Canada being transported to the US Midwest (Flight date: June 28, 2006).

We now use the same hygroscopicity adjustment for that flight as we do for all the other flights, but we've labeled the point BB for biomass burning in Figs 3-5. We've added the following text to the manuscript about the issues with assuming a constant hygroscopic growth factor:

"While Equation 1 takes into account differences in hygroscopic growth due to RH for each segment of each flight, it does not account for compositional changes that might affect the scattering enhancement due to hygroscopicity.    For aerosol events such as biomass burning and dust episodes with significantly different composition than the 'normal' aerosol we would expect to over-predict the aerosol hygroscopicity relative to the normal aerosol.  Sheridan et al., (2001) showed that the SGP surface aerosol had lower hygroscopicity when it was influenced by dust or smoke."

We've also added this to the discussion of Figure 3:

"One thing to note on Figure 3a is the blue point marked BB (the BB stands for biomass burning).  This measurement occurred on June 28, 2006 and appears to have been strongly affected by forest fire smoke transported from Canada.  We applied the same hygroscopicity adjustment to the measurements of this flight as we did to all of the BND flights and, in this BB case, the hygroscopicity correction was the primary reason the in-situ AOD value is significantly higher than the AERONET AOD value.  This point would lie much closer to the 1:1 line if the in-situ BB data were assumed to be hygrophobic.  Previous work at the surface site at SGP has shown that dust and smoke aerosol types tend to exhibit lower hygroscopicity than the background aerosol normally observed at the site (Sheridan et al., 2001).  This BB point provides an extreme example of the downside of using a constant hygroscopic growth parameter as a function of RH, although without additional information about the aerosol for each profile it is difficult to do otherwise.    The light blue dotted line on Figure 3 represents the relationship between AERONET and in-situ data if the BB point is excluded."

Section 2.1 (line 267): Please change "column average properties" here to "flight profile average properties" to accurately reflect the fact that the aircraft does not measure the total atmospheric column, as AERONET does.
Done

Section 2.2 (line 290-292): Please add 'calibrations' before 'corrections' as the consistent high accuracy calibration of AOD and sky radiances are the basis for what makes AERONET data so valuable.
Done

Section 2.2 (line 294): Please add that the AERONET data is Version 2 data, since the Version 3 database will be available in the near future.
Done – we've also mentioned that the Version 3 data are coming in the same paragraph.

Section 2.4.1 (line 356-358): Please discuss whether you accounted for soil dust and biomass burning aerosols in the 'aerosol chemistry' mentioned here.

Our hygroscopicity relationship accounts for the hygroscopic growth based on the 'typical' aerosol chemistry - we did not specifically account for soil dust and biomass. We see little indication that the comparison flights were influenced by BB or dust (with the exception of one flight at BND which we now note in the text). We utilized the parameterization by Quinn et al. (2005) which uses the organic mass fraction (defined in Quinn et al as OC/(OC+sulfate) where OC= organic carbon concentration and sulfate = sulfate concentration ) to estimate hygrosocopicity. She developed this parameterization based on chemistry and hygroscopicity measurements at several sites, including sites impacted by dust and biomass burning. We've add some more details about this, in response to the reviewer's previous comment on this topic in both Section 2.1 and in the discussion of Figure 3. (See our response to previous related comment.)

Section 2.4.1 (line 361-364): In your discussion of RH levels during the profile flights please include some mention of the higher RH (RH halos) that typically exist in the vicinity of non-precipitating cumulus clouds that are imbedded within the aerosol layer < 4 km. Higher RH near cumulus clouds and higher AOD in the near Cu cloud environment (likely due to combined humidification, cloud processing of aerosols and rapid gas-to-particle conversions) were observed by Jeong and Li (2010) at the SGP site utilizing both AERONET data and in situ aircraft data. If you only flew aircraft profiles on cloudless time periods or avoided flying near clouds then this needs to be documented in the manuscript, as the sampling could possibly be skewed to specific meteorological and/or cloudiness conditions.

Thanks for bringing the Jeong and Li (2010) paper to our attention. I've also passed it on to the DOE Arm Aerial Facility manager as they try to keep track of papers using data from the IAP aircraft (e.g., Schmid et al., BAMS, 2014). We would not have expected to have AERONET retrievals available for comparison with the aircraft data under such conditions due to the rigorous cloud screening the AERONET Level 2.0 data undergoes. Jeong and Li (2010) made use of earlier measurements made by the same SGP aircraft flying the same profiles (albeit with a 1um inlet and max altitude of 3659 asl). Both the BND and SGP aircraft were operated under visual flight regulations and could not fly in clouds – they would skip a flight level if there was a cloud on that level and we did not use any flights that had missing flight levels in this analysis. We've now specifically mentioned this by adding the following sentence to the first paragraph of section 2:

"The flights at both sites were subject to 'visual flight regulations' which means they took place during daylight hours and the plane did not fly in-cloud."

We've also added another paragraph to the discussion of in-situ uncertainties and cited the Jeong and Li (2010) paper in there. Here is the text we've added:

"Jeong and Li (2010) have noted that the presence of nearby clouds may influence AOD values. They've investigated the effect of high RH-halos embedded in aerosol layers that typically exist in the vicinity of non-precipitating cumulus clouds. If the AERONET retrieval went through such a halo it could result in an increased AOD due to the combined effects of hygroscopic growth, cloud processing of aerosols and rapid gas-to-particle conversions. If the aircraft also flew through this RH-halo then the effect would also be accounted for in the RH-corrected in-situ measurements. However, if the high RH layer was between two flight levels then the aircraft measurements would not account for it.  Addressing this effect is outside the scope of this paper."

Section 2.4.1 (line 401-402): Although biomass burning does not have a consistent influence at these sites it is episodic, therefore did you exclude these biomass burning aerosol episodes from your data analysis? If so how did you identify the biomass burning episodes?
No data were excluded due to type of aerosol (biomass burning or otherwise).  We did try to identify points that were affected by biomass burning and we note in section 3.2 line 705-706 that the BND point with AOD~0.4 represents a day we believe was affected by biomass burning.

Section 2.4.1 (line 443-445): Please state here that the uncertainty estimate for the in situ SSA of 0.04 is a lower bound since it does not take into account the effect of particle coatings on aerosols since the aerosols are modified (dried) before the measurements are taken, plus some fraction of the coarse mode particles are not sampled.
Based on the recommendation of another reviewer we've doubled the PSAP uncertainty to account for the effect of coatings, since the coating enhancement is unknown. We've added the following sentence in Section 2.4.1 when discussing the absorption enhancement:
"To address this, we double the assumed PSAP uncertainty of ~25% to 50% in the calculations of uncertainty."

Section 2.4.1 (line 451-454): You state here that 15% of the aerosol in the column is not sampled below the lowest flight altitude (150 m agl) of the aircraft for in situ measurements, and it can also be inferred that possibly another 15% or more is not sampled above the highest flight altitude on very low AOD days or high AOD days with layering from convective vertical aerosol transport. Therefore it is likely that 30% of the aerosol in the total atmospheric column is not sampled by your aircraft vertical profiles. This issue needs some discussion in the text and also should be factored into your uncertainties of in situ measured SSA (or clearly state that it has been ignored).
We've significantly augmented, rearranged, and rewritten this discussion of potentially missed aerosol below and above the aircraft as described below.

We're not sure that the reviewer's suggestion that the aircraft is likely missing 30% of the aerosol is reasonable.  We based our comment that the aircraft could be missing 15% below the lowest flight level on previously published results for the comparing the lowest level leg (LLL) with the surface (S) measurements (e.g., Sheridan et al., 2012; Andrews et al., 2004; 2011; Esteve et al., 2012).  We've now gone back and looked at the lowest level leg/surface comparison for just the flights included in the direct comparisons reported on here.   At BND the relationship is LLL=1.0*S-0.99, R2=0.99 suggesting the lowest level leg and surface are seeing virtually identical aerosol.  At SGP the relationship is LLL=1.17*S+0.43, R2=0.96, so for these particular SGP flights the airplane is actually seeing ~17% higher aerosol than is observed at the surface.  The implication is that at SGP we may be over-estimating  by applying the lowest level leg value down to the surface in order to obtain the column values.  We've added the following text:

"We've looked at the surface/lowest flight leg relationship specifically for the flights with matching AERONET retrievals studied here. We found that at BND the surface and lowest level flight aerosol measurements were virtually identical. At SGP the lowest level leg actually measured slightly higher aerosol loading than was observed at the surface, which could lead to an overestimate of the aerosol optical depth in that layer, depending on the shape of the profile."

Obviously if there are layers above the highest aircraft flight level they wouldn't be sampled and that will negatively impact the AOD and AAOD comparisons. Depending on the layer loading that impact could be significant. It is however unclear to us how the reviewer can infer that the aircraft might be missing 15% or more above the highest flight on very low AOD days. Turner et al. (2001) segregated lidar aerosol extinction profiles at SGP by season and loading. Their results (their Figure 1) suggest that, for the vast majority of cases observed at SGP, 5% or less of the extinction will be found above 4 km. For low AOD cases ($AOD_{355}<0.3$) their mean extinction profiles suggest little to no aerosol extinction between 4-7km.

A 30% upward adjustment of the in-situ measurements would worsen the AOD comparisons shown in Figure 3 but not greatly improve the AAOD comparisons shown in Figure 4.

However, to further address this concern, we've now looked at the Raman lidar best estimates of aerosol extinction profiles at SGP for the 14 flights with AERONET matches (there is no lidar data available from BND). We found three cases where there appeared to be an aerosol layer in the vicinity of the highest in-situ flight levels, but in each case the profile flight provided a hint of the presence of this layer. Looking at the actual shape of the in-situ profiles, these three flights exhibited a significant increase in measured loading at the highest flight levels. We've removed those flights from the comparisons reported here. There may still be aerosol above the height of the Raman lidar but we have no means for identifying it. Based on the criterion of observing a strong increase in aerosol loading at the highest flight levels, we also removed 3 flights from the set of BND profiles. We've added the following text:

"Although statistical profile results (e.g., Turner et al., 2001; Yu et al., 2010; Ma and Yu, 2014) suggest little contribution from high altitude aerosol layers in the region of these two sites, Schutgens et al. (2016) demonstrates the importance of considering the specifics rather than the statistical. We used the Raman lidar best estimate data product of extinction profiles at SGP to evaluate the presence of aerosol above the highest flight level at the site. For the SGP in-situ profiles that had matches with AERONET inversion retrievals, we identified three lidar profiles that exhibited aerosol layers at high altitudes, but in all three cases the presence of these layers was also hinted at by an increase in the aerosol loading at the highest flight levels of the in-situ measurement. Thus, we further screened in-situ/AERONET comparisons by removing flights at SGP and BND with significant increases in loading at the highest flight levels. There may still be aerosol layers above the level measured by the Raman lidar, but we have no means of assessing that. The AOD comparison presented in Figure 3 suggests we are unlikely to be missing significant aerosol at high altitudes."

We've also added the following three sentences to the section.
"Missing aerosol above and below an aircraft profile is a potential issue in all aircraft/column comparisons."

And

"Turner et al. (2001) segregated lidar aerosol extinction profiles at SGP by season and loading. Turner et al.'s results (their Figure 1) suggest that for the vast majority of cases observed at SGP, 5% or less of the extinction will be found above 4 km.  For low AOD cases (AOD$_{355}$<0.3) their mean extinction profiles suggest no aerosol extinction between 4-7km."

And

"Regionally, seasonal average profiles from CALIPSO also suggest there is minimal aerosol above the flight's highest level (Ma and Yu, 2014; Yu et al., 2010)."

And

"Andrews et al. (2004) also assumed assumed an AOD contribution of 0.005 from stratospheric aerosol which was not done here."

Section 2.4.1 (line 464-467): Please elaborate here on whether the estimates of the percentage of aerosols above the highest flight altitudes as analyzed by Andrews et al. (2004) were comparisons made for all AOD levels and seasons. It would not be surprising for a greater percentage of AOD above flight altitudes to occur in summer when convection is stronger (transporting boundary layer aerosols upwards), or also in all seasons when AOD is very low since there is always some background midtropospheric to stratospheric AOD present which constitutes a greater percentage of total AOD when AOD magnitudes are very low.

The estimates in Andrews et al., 2004 were made by matching Raman lidar observations with each individual flight.  The flights discussed in Andrews et al., 2004 covered all seasons and loadings.  We've now utilized the Raman lidar data at SGP to further evaluate the potential high altitude contribution of aerosol as described in our response to the previous comment.

Figure 3: In Figure 3, please explain how you can have 3 observation points of AERONET measured AOD at 675 nm ranging from 0.5 to 1.0 at INSITU AOD of 0.15 when there does not seem to be any corresponding data at 440 nm in the plot above it. This does not seem possible, and should be explained in the text.

This is in reference to the BND plots.  These 3 points match up with the flight on DOY 187, 2006 (July 6, 2006) for a flight ending at 187.80191.  The AERONET values at 675 nm are: 0.493, 0.824, and 0.993 and the corresponding values at 440 nm are: 0.888, 1.464, 1.754.  The two points greater than 1 are off the scale of the 440nm plot (Figure 3a) as it only goes up to 1. The 3$^{rd}$ 440 nm point (AERONET AOD value 0.888, insitu AOD value 0.32) is (barely) visible under the blue linear fit equation. I've added the following text to the caption:

"Note: two BND direct sun AOD440 points corresponding to the two highest AOD675 points in the figure below are off the scale of the plot and not shown.  The third high AOD440 point is partly obscured by the legend."

Section 3 (line 528-530): Please also add to this paragraph that fact that the in situ under sampling of the total atmospheric column AOD is due to the restricted altitudes of the flight profile measurements (150 meters to 4200 meters).

We've added the following text in this section:

"Some of the discrepancy between the in-situ and the AERONET values may also be due to the limited vertical range covered by the airplane (150 – 4200 m asl)."

We've also included the reported altitude ranges and additional altitude information for all flights in Tables 3 and 4 (flight ranges are in column 2, additional altitude related info is in the comments column):

*Schafer 2014*: 250-5000 m (doesn't say if agl or asl) for column comparison flights average altitude range is 367-3339 m.  They required flights to be less than 500 m and greater than 1500 m to obtain adequate representation of column.

*Magi 2005:* 170-1500 m agl

*Mallet 2005:* 100-2900 m (doesn't say asl or agl)

*Leahy 2007:* 100-5320 m asl (that's min and max over 5 flights – no flights covered that entire range).  They used AATS to account for aerosol above plane and extrapolated down to acct for aerosol below plane. (Altitude range obtained from flight info in Magi et al, 2003)

*Haywood 2003:* 330-3420 m agl, extrapolated down to ground acct for aerosol below plane

*Osborne 2008:* 100-5000 m (doesn't say agl or asl) (that's min and max over 4 flights – no flights covered that entire range).

*Johnson 2009:* 150-3000 m (doesn't say agl or asl)

*Corrigan 2008:* 0-3200 m asl

Section 3.1.2 (line 604-607): Please note that the in situ instrument known cutoff of 5 micron for particle diameter for the aircraft sampling would also contribute to an under sampling of total column AOD, in addition to the incomplete altitudinal atmospheric profile for the total column AOD.

The first sentence of this paragraph has been adjusted to read:

'The other likely candidate to explain the in-situ AOD being slightly lower than the AERONET AOD is aircraft under-sampling of super-micron aerosol due to the 5 $\mu$m inlet cutoff'.

Figure 6 (numerous comments follow regarding some of the referenced data sets plotted in the Figure, especially note the issues regarding aircraft sampling and also the fact that some papers published Version 1 data that were biased due to inaccurate surface albedo assumptions, versus current Version 2 data that became available in 2006):

We utilized Version 2 data for all studies that used/reported Version 1 data.  That is noted in the comments column of Table 4 for the relevant papers – that's what the note 'Used AERONET 2.0' was supposed to indicate.  I imagine that could be confused with level 2.0 data so we've changed 'Used AERONET 2.0' to 'Used V2 AERONET Level 2.0'.  We've also added the following sentence to the end of the first paragraph in Section 3.2:

"Please note that some of the earlier studies shown in Figure 6 and described in Table 4 used values from Version 1 AERONET data.  Where that was the case, we retrieved Version 2 AERONET data from the AERONET website and those Version 2 data are what is depicted in Figure 6.  The comments section of Table 4 mentions the cases where this was done."

Osborne et al. [2008] compared three cases of aircraft flights (on three different days) over the same site during the same experiment with the same instruments and aircraft but found that the aircraft in situ measured SSA values ranged from 0.04 to 0.07 higher than the AERONET version 2 retrievals. However, for all three of these cases the aircraft measured Angstrom exponents were found to be about 0.40 lower than the AERONET measured values. This discrepancy in AE suggests that the aircraft may have sampled a different fine and coarse mode fraction mixture than the column integrated value measured by AERONET, and the higher SSA in conjunction with lower AE measured by the aircraft is consistent with this possibility. In fact, for the linear fit of SSA versus AE for all aircraft data from DABEX, reported in the work of Johnson et al. [2008], a difference of 0.40 in AE corresponds to a difference in SSA of about 0.06, almost the same value of the bias reported in Osborne et al. [2008].

It's already noted in Table 4 that there was a large discrepancy between the aircraft and AERONET AOD comparison and that the aircraft may have over-sampled large particles (or over corrected for large particles).

Johnson et al. [2009] compared in situ measured aerosol optical properties from an aircraft vertical profile flight over the Banizoumbou (Niger) AERONET site on 19 January 2006. This was a mixed aerosol case with Angstrom exponent (450–700 nm) of approx. 0.8–0.9 and high 550 nm AOD of approx. 0.75, where a shallow dust layer up to 1 km altitude was overlain by a layer of predominantly fine mode smoke. Both aircraft and AERONET measurements of column integrated AOD at 550 nm and of AE were in good agreement for this case, with dAOD = 0.08 (INSITU was 7% higher) and dAE = 0.06, suggesting that both were sampling the same aerosol mixture. The aircraft measured column mean SSA at 550 nm (from PSAP and nephelometer) was 0.87, in good agreement with the AERONET retrieval of 0.85 (interpolated to 550 nm).

These are the values reflected in Figure 6.

Magi et al. (2005; JAS) Note: Version 1 retrievals were 0.015 lower than V2 retrievals on this day at 1310 UTC at COVE site: From the paper: "Ground-based retrievals of SSA were obtained by the Aerosol Robotic Network (AERONET) sun photometers (e.g., Dubovik et al. 2000) during the CLAMS field campaign from a site known as the Clouds and the Earth's Radiant Energy System (CERES) Ocean Validation Experiment (COVE; 36.98 N, 75.78 W). The vertical profiles were often spatially located close to COVE. The mean value of SSA at 550 nm from AERONET retrieval data (processed to remove clouds and manually quality assured) is 0.94 +- 0.03. Therefore, the mean value of SSA retrieved from AERONET agrees with mean value of SSA derived from our in situ airborne measurements (0.96 +- 0.03) to within one standard deviation. On 17 July 2001, measurements were made from the UW aircraft and the COVE site that were both temporally (the aircraft vertical profile was from 1304–1337 UTC and the AERONET retrieval was at 1310 UTC) and spatially (the aircraft was ; 2.5 km from COVE) collocated. The mean value of SSA calculated from the airborne in situ measurements made in polluted layers during this vertical profile was 0.97 +-0.02; the corresponding column-averaged value of SSA for accumulation mode particles retrieved from the AERONET data was 0.90 +- 0.03 (VERSION 1 data). Particle losses in the sampling system for the in situ instruments could have contributed to an underestimate of the absorbing component of the aerosol. Spatial variability may have played a role as well."

As stated above and in the comments section of Table 4, we retrieved the Version 2 AERONET AOD440 values from the AERONET website.

Mallet et al. (2005): V2 almost same as V1 at 0.932 at 550 nm at 6 UTC: AERONET retrieval (at 0600 UTC ; on June 25, 2001) of the single scattering albedo at Avignon indicated a coherent value (SSA 0.93 at 550 nm) compared to the one obtained from optical measurements for flight 41 (0515–0537 UTC, SSA 0.94 in the PBL).

As stated above and in the comments section of Table 4, we retrieved the Version 2 AERONET AOD440 values from the AERONET website.

Haywood et al., (2003): Comparison of aerosol size distributions, radiative properties, and optical depths determined by aircraft observations and Sun photometers during SAFARI 2000 V2 SSA = 0.84, 0.83, 0.81, 0.80 for 440, 675, 870 and 1020 nm V1 SSA = 0.88, 0.87, 0.84, 0.82 for 440, 675, 870 and 1020 nm The corresponding SSA for the mean size distribution used in the calculations derived from the PCASP distributions and from the nephelometer and PSAP on the C-130 is 0.90, 0.87, 0.85, and 0.82 (Table 1).

We thank the reviewer for pointing out our error – in Figure 6 we used the the V1 SSA value for AERONET of 0.88 when we should have used the V2 SSA value to be consistent with the other comparisons. The pink triangle for this campaign shifts to -0.06 (instead of -0.02). We have updated Figure 6 and the comments column of Table 4 accordingly.

Figure 7 caption: One detail regarding the caption of Figure 7 that is misleading is the Level 2 AOD shown in the plot. Please note that Level 2 AOD exists for all AOD levels not just for AERONET almucantar retrievals for which AOD is >0.4 at 440 nm. Level 2 almucantar retrievals of size distributions are made for all AOD levels, but refractive indices are only given for AOD>0.4 at 440 nm. Therefore the authors need to clearly describe and accurately label in the figure caption that this data as only associated with AERONET almucantar retrievals for which AOD>0.4 and therefore have error bars on SSA of 0.03.

The caption as originally submitted says "AERONET AOD medians are for observations with Level-2 almucantar retrievals, with corresponding AAOD and SSA retrievals at Level-1.5 (black) or Level-2 (gray). AERONET 2.0 values are biased high by definition, because of the $AOD_{440}$>0.4 constraint."

We've added in the words 'Level-2 almucantar' and 'AOD and AAOD' in the following caption sentence to further clarify:

AERONET Level-2 almucantar AOD and AAOD values are biased high by definition, because of the $AOD_{440}$>0.4 constraint.

The actual complete Level 2 AOD data set (for all AOD levels) shows monthly means that are significantly higher in summer with many more days of data sampled and many more partly cloudy to mostly cloudy days sampled also (see Jeong et al., 2010; JGR for a discussion of higher AOD in the near cloud environments at the SGP site).

Figure 7 shows monthly medians not means. Below, in the other comment related to this point, I show a version of the AOD plots from Figure 7 that also includes the direct sky AOD medians – they lie directly on top of the 1.5* median values.

Section 3.2 (line 692-695): Your statements here assume that the in situ determinations of SSA are un-biased (despite the fact that ambient aerosol properties are not actually measured). This has not been proven in the paper, especially since the in situ data have to be corrected for humidification effects, the total column aerosol is not sampled, and the effects of aerosol coatings are not accounted for (therefore blindly assumed have no effect**). Please revise or eliminate these sentences.

We've rewritten the paragraph as follows:

"In summary, the literature survey featuring measurements across the globe for many aerosol types suggests that even at higher AOD conditions, direct comparisons of AERONET with in-situ aerosol profiles find that AERONET column SSA is consistently lower than the SSA obtained from in-situ measurements (although often within the uncertainty of the AERONET SSA retrieval and in-situ measurements). If there was no consistent bias in the AERONET/in-situ comparison we would expect (AERONET_SSA – INSITU_SSA) to be evenly distributed around zero. Instead, Figure 6, which summarizes the literature survey, suggests either that AERONET retrievals are biased towards too much absorption, or that in-situ, filter-based measurements of aerosol absorption are biased low. We note that the results from the literature indicate that the hypothesized low-bias in in-situ absorption is not associated with a single airplane's measurement system or the atmospheric conditions encountered in a single experiment. That leaves us with possible bias in the in-situ experimental methods (instrument issues (nephelometer, PSAP), treatment of f(RH), vertical coverage, sampling artifacts), all of which we have attempted to address above."

Section 3.3 (line 754-768): Please note that AOD sampled by AERONET in the Level 2 dataset (not just for the subset that have L2 retrievals) includes many more days of data than the in situ flights, and is therefore a much more statistically robust data sample. Please note that the Level 2 AOD climatology for the SGP site (average of 13-19 years per month) shows significantly higher AOD (440 nm) than shown for L1.5* in Figure 7, For example for the SGP site the August monthly mean AOD is 0.272 and the September monthly mean is 0.215 at 440 nm. Similarly for the BND site the L2 monthly means of AOD(440 nm) for June, July, August, and September are 0.282, 0.329, 0.343 and 0.283 respectively (computed from 15-17 years of data per month). These monthly means are significantly higher than the AERONET values shown in Figure 7, since the data in Fig 7 are only AOD associated with the Dubovik and King algorithm retrievals.

The plots in the manuscript show medians, not means. Below I've pasted the AOD portion of Figure 7 that also includes the medians for the version 2 Level 2 direct sky AOD measurements (in mustard). The direct sky medians lie pretty much directly on top of the 1.5* median AOD values (black lines). We have not added the direct sky AOD line to the plot in the manuscript. This sentence (lines 739-740 of original manuscript) still stands:

"The AERONET Level-1.5* AOD monthly medians are representative of the direct sun AERONET Level-2 AOD climatology at the two sites."

We've clarified this sentence in the following paragraph by adding the phrase 'direct sky':

"…AERONET Level-1.5* retrievals (recall that the AERONET 1.5* AOD is representative of the overall direct-sky AERONET AOD climatology at each site)…"

[Figure]

| BND monthly median AOD values | SGP monthly median AOD values | LEGEND
AERONET 1.5* AOD
Version 2, level 2 AOD for successful Almucantar scans with AAOD and SSA retrievals (i.e., has the AOD440<0.4 restriction)
INSITU AOD
AEROCOM (phase II) AOD
Version 2, level 2 direct sky AOD |

Section 3.3 (line 781-783): You state: "The AERONET 1.5* SSA values tend to be quite a bit lower than the other data sets at both sites, which is why the AERONET 1.5* AAOD values tend to be higher (recall that for AERONET data AAOD is calculated using AAOD=(1-SSA)*AOD)." No, this is not really accurate, since as shown on Figure 7, at the SGP site the agreement between the AERONET L1.5* data and in situ measurements of SSA are well within the uncertainty of the measurements for all months (and you have not proved that the in situ is not biased). Please revise this sentence to reflect this fact as presented by the data shown in Figure 7b.
We've revised this sentence to read: "The AERONET 1.5* SSA values tend to be quite a bit lower at BND, and somewhat lower at SGP which is why the AERONET 1.5* AAOD values tend to be higher (recall that for AERONET data AAOD is calculated using AAOD=(1-SSA)*AOD).

Section 3.4 (line 875-879): Again, you have omitted the fact that for the SGP site the agreement between the in situ estimations of SSA and the AERONET retrievals of SSA are within the uncertainty levels of these data sets over the entire range of AOD shown in Figure 8.
Note: we've remade Figure 8 so that the surface in-situ SSA values are also now at ambient conditions.  We've added the following sentences to the discussion of Figure 8:
"It should however be noted that despite the discrepancy between in-situ and AERONET SSA values, Figure 8 shows that the SSA values for all three sets of measurements at SGP are within the reported AERONET SSA uncertainty range of 0.05-0.07 for $AOD_{440}<0.2$ across the narrow and low AOD range shown in the figure.  At BND the SSA values are within the AERONET SSA uncertainty range down to $AOD_{440}\sim0.1$."

The way the paper is written there seems to be a consistent attempt to steer the reader to the conclusion that the AERONET retrievals are biased low despite significant uncertainties in the in situ determinations and despite the fact that the in situ instruments do not measure ambient aerosol properties directly without corrections. Therefore it is not proven that the in situ determinations of SSA are unbiased themselves, so the text and title require rewriting to acknowledge this.

New title: "Comparison of AOD, AAOD and column single scattering albedo from AERONET retrievals and in-situ profiling measurements"

We've gone rewritten the abstract, text and conclusions to emphasize that (a) there is a systematic difference in the comparisons that would suggest 
[revised manuscript text omitted]

---

## Author Response (AR2)

**General comments:**

The authors have extensively revised the manuscript and have responded satisfactorily to many of my comments/suggestions. However, some important issues remain which need to be resolved before publication.

We thank the reviewer for really getting after us on the uncertainty analysis. It's been helpful (and educational) to continue to dig into this and we think it's improved the paper. We've now added a figure (new Figure 2) showing the SSA uncertainty dependence on AOD (see response to second general comment below).

Especially regarding Figure 6: Please explain how you estimated the error bar in yellow shading in Figure 6. You had stated previously in this revised manuscript that the in-situ uncertainty in SSA was 0.06 (and gave no AOD dependence for this) while for AERONET it is 0.03 for AOD(440)>0.2 and 0.05-0.07 for lower AOD values (Table 2 of Dubovik et al., 2000). How did you get SSA uncertainty in Figure 6 of 0.05 for AOD>0.20 and 0.08 for lower AOD? In the first draft that I reviewed the in-situ SSA uncertainty was estimated to be 0.04, yet the yellow shaded uncertainty limits in Figure 6 were the same as in the revised manuscript. It appears that the uncertainty shading in Figure 6 needs to be updated.

The yellow shading in Figure 6 (now Figure 7) indicates the combined standard uncertainty of the AERONET and in-situ SSA values. The combined uncertainty is not additive, rather we've utilized the following equation (BIPM, 2008, their eqn 16) to estimate the uncertainty in the difference of the two SSA values based on the reported uncertainties for each:

$Unc^2 = (SSA_{is})^2*(unc.SSA_{is})^2 + (SSA_A)^2*(unc.SSA_A)^2 + 2*SSA_{is}*SSA_A*unc.SSA_{is}*unc.SSA_A*R_{A,is}$

where the subscript 'is' indicates the in-situ data and 'A' indicates AERONET data and R is the covariance between AERONET and in-situ SSA for all the studies (0.1). We used average SSA values for in-situ and AERONET (0.95 and 0.89, respectively) and SSA uncertainties for both AERONET and in-situ of 0.06 for AOD<0.2 and SSA uncertainty of 0.03 for AOD>0.2. The shaded area for the AOD<0.2 cases stays the same (±0.08) while the shaded area for AOD>0.2 decreases to ±0.04. We've updated the plot to reflect this.

The text now has this additional information in the discussion of the figure and the figure caption was slightly modified:

*"..., although most of the values are within the combined standard uncertainty of the AERONET and in-situ values indicated by the shading (see BIPM, 2008, their equation 16 for how the combined standard uncertainty was calculated)."*

This reference was added.
BIPM, "Evaluation of measurement data – Guide to the expression of uncertainty in measurement," Joint committee for guides in metrology (JCGM): 100, http://www.iso.org/sites/JCGM/GUM-JCGM100.htm, 2008.

Some attempt should be made to make an estimate of in-situ uncertainty for extremely low concentrations which occur in winter at mid latitudes (such as at SGP and Bondville), for total column AOD values ranging from ~0.02 to 0.08 at 440 nm (Figure 8). It is not realistic to think that the uncertainty of the in-situ measured/derived SSA will not increase when these very low aerosol loadings occur at all flight levels, especially since the flight segments at lower levels are presumably still 5 minutes in duration. Concentrations at all altitudes on very low AOD days (<0.08 at 440 nm) would likely be similar or lower than high altitude concentrations on moderate to high AOD days, therefore the overall uncertainty of the in-situ data would likely be significantly greater for these low AOD days. The paper still lacks convincing discussion/analysis regarding in-situ SSA uncertainty on the lowest AOD days (occurring typically in the winter season), as it must be higher uncertainty than on moderate to high AOD days when aerosol concentrations at flight altitudes in the boundary layer can be an order of magnitude (or more) higher. (see Lines 399-400 & Lines 528- 532 in the revised MS for this topic/issue).

We've added the following plot to the paper that shows the calculated uncertainty of the in-situ SSA as a function of AOD.  The points on the plot represent all three wavelengths (red/green/blue for the visible) for the in-situ instruments.  The larger purple points represent the in-situ 'column' SSA uncertainty as a function of the 'column' in-situ AOD (column is in quotes as the airplane doesn't sample above 4700 m and only samples discretely below that).  The small black points represent the SSA uncertainty for each individual flight layer as a function of that layer's AOD.  For comparison the orange line represents the AERONET uncertainty values reported in Dubovik et al. (2000, their table 2). They give a range of 0.05-0.07, so we used 0.06 to represent the AERONET SSA uncertainty for AOD<0.2.  This figure shows that the in-situ SSA uncertainty of 0.06 is a worst case value, the median in-situ 'column' SSA uncertainty for AOD<0.2 is 0.03.  This 0.03 uncertainty is actually consistent with equation 2 in the paper if we use an SSA value of 0.95, which is more representative of the SSA values obtained from the in-situ measurements.

On the AERONET/in-situ comparison plot (formerly figure 5, now figure 6) we left the uncertainty range as 0.06 to represent the worst case scenario.

[Figure]

**Specific Comments:**

Lines 347-348: Regarding the AERONET Version 3 data, maybe say at the 'time of writing this manuscript' here.
**done**

Lines 354-355: This should be "if there was a corresponding Level 2 almucantar retrieval available, but AOD (440)<0.4."
**done**

Lines 427-429: Please include the reference of Eck et al. (2014; ACP) here, as this paper specifically discusses the phenomenon you are discussing here, of the measurement of higher AOD in the near vicinity of cumulus clouds. This paper also shows that both lidar and in-situ measurements found similar aerosol increases as AERONET measured AOD near to cumulus clouds during this field campaign in Maryland.
**done**

Line 638: Direct sun AOD are more accurately called measurements, not retrievals (the AERONET retrievals are made with the Dubovik and King algorithm from almucantar sky radiance and spectral AOD input).
Changed 'retrieval' to 'measurement'

Lines 674- 675: That is not possible, as there is always aerosol above 4.2 km asl, such as mid- to upper troposphere plus stratospheric aerosol. Perhaps the word 'significant' needs to be added here, since the exact amount in AOD above the 4.2 km level is often unknown.
Added word significant

Lines 761- 762: Perhaps putting both Hybrid and AERONET on the y-axis label would make this plot (Figure 5) easier to interpret.
Thanks for pointing this out! We've changed the axis.

Lines 806- 808: It should be mentioned here in the text that for mineral dust there is significant absorption at 440 nm in the coarse mode from iron oxide content.
The reviewer makes several comments about the importance of mineral dust absorption, suggesting that it could perhaps be a significant contributor to the discrepancies between in-situ and AERONET AAOD and SSA observed here, due to the aircraft inlet 50% sampling cutoff excluding particles with aerodynamic diameter > 5um (equivalent to ~3um once the particle density is accounted for).  We respectfully disagree with this suggestion for the following reasons.
   (1) While it is true that that dust absorption is dependent on iron oxide content particularly at lower wavelengths, our plots at 675 nm (where dust is less absorbing) don't look markedly different from the plots at 440 nm (where dust is more absorbing) – see figures 4cd and 5cd in the original manuscript (now figures 5cd and 6cd).  We state that this is why the 675 nm plots are included on lines 289-292.  We briefly discuss the implications of these plots in the paragraph starting at line 720 in the first revision of the manuscript (now line 725).

(2) There's variability in how much iron oxide there is in any given dust sample. Recent work by Engelbrecht et al. (2016) show that most dusts and clays have single scattering albedos >0.9 at 405 nm. This suggests that dust will have similar absorption properties to the aerosol we are observing at these two sites. We've added a reference to Engelbrecht in the paragraph discussing the comparisons at 675 nm.

(3) In the paragraph starting on line 792 in the revised manuscript we determine we may be missing ~10% of the aerosol in the in-situ measurements based on the AERONET size distribution retrievals. These particles would have to be very absorbing and not very scattering to account for absorption discrepancies of a factor of ~2.

Engelbrecht et al., "Technical note: Mineralogical, chemical, morphological, and optical interrelationships of mineral dust re-suspensions," Atmos. Chem. Phys., 16, 10809-10830, 2016. http://www.atmos-chem-phys.net/16/10809/2016/

Lines 817- 824: Please add here the underestimation of coarse mode absorption at 440 nm due to dust particles that are under-sampled in the in-situ measurements.
See comment above.

Lines 894-898: This should be stated as greater absorption than in-situ here (rather than over-estimation) since the in-situ data is not recognized (by the scientific community) as truth or gold standard for SSA.
Rephrased sentence:
*Most of the SSA comparisons in Table 4 found fairly good agreement between AERONET and in-situ AOD, implying that the discrepancy is associated with the absorption values rather than the scattering values (since scattering is typically 90% of extinction).*

Lines 898-900: In the interest of completeness and unbiased reporting, the number of cases where agreement is excellent (within 0.02), given the in-situ uncertainty of 0.06, should be mentioned here in the text.
We added the following sentence in the previous paragraph – it seemed to fit better there:
*"Of the 63 cases depicted in Figure 7, 16 cases (~25%) of the AERONET/in-situ comparisons were within 0.02."*

Line 907: Please change 'often' to 'mostly' here since most data shown in Figure 6 for AOD >0.2 at 440 nm are within the yellow shaded uncertainty bounds.
done

Lines 970 - 971: Not really true, since the almucantar associated measurements (L1.5*) of AOD are biased towards days with lower cloud cover due to sky radiance screening for clouds away from the solar disc. Additionally almucantar retrievals are only valid in the relatively early morning and late afternoon-evening in summer at mid-latitude (SZA>50, which occurs all day in winter) thereby missing the most active fair weather cumulus development times of day in summer. AOD measurements associated with more cloudy conditions are often higher due to increased particle hygroscopic growth, cloud processing of particles in aqueous phase chemistry, gas-to particle conversion that occurs more rapidly in cloud droplets, and convergence of aerosols associated with larger cloud systems (see Jeong and Li (2010) and Eck et al. (2014)).

We respectfully disagree with this comment – the plot below is the same one we included in our previous response to the reviewers. The mustard colored line represents the AERONET level 2 direct sun climatology. The black line (which is mostly hidden by the mustard colored line) represents the AERONET level 1.5* AOD climatology. We stand by our statement that:
*"The AERONET Level-1.5* AOD monthly medians are representative of the direct sun AERONET Level-2 AOD climatology at the two sites."*

[Figure]

| BND monthly median AOD values | SGP monthly median AOD values | LEGEND
• AERONET 1.5* AOD
• AERONET Version 2, level 2 direct sky AOD
• Version 2, level 2 AOD for successful Almucantar scans with AAOD and SSA retrievals (i.e., has the AOD440<0.4 restriction)
• INSITU AOD
• AEROCOM (phase II) AOD |

Lines 983 - 984: This statement is not rigorously true, as noted in the previous comment.
See response to previous comment.

Lines 986 - 987: These are very large and significant differences ("by up to 50%") between in-situ and AERONET measured AOD. Please note this in the paper and attempt an explanation for the significant low summer season bias of the in-situ AOD measurements.

We agree that it's a pretty significant difference, but remember this figure is comparing climatologies not time-matched comparisons. The AERONET data at BND cover the time period 1996-2011 (but is missing June 2007-April 2008 due to AERONET instrument malfunctions) while the airplane covers several hundred ~3h flights between June 2006-Sept 2009. While we can compare the climatological values in these plots, that was not the primary point of presenting them. Rather, our goal for presenting the climatological information was threefold:

(1) Show that we see similar seasonal patterns amongst different methods of obtaining monthly AOD (in-situ, AERONET level 1.5 (also direct sun level 2.0 as shown above) and model). Our first sentence discussing these figures is "*At both sites, the climatological seasonal patterns for AOD (i.e., high in summer, low in winter) are similar for the three data sets: in-situ measurements, AERONET Level-1.5* retrievals (recall*

*that the AERONET 1.5\* AOD is representative of the overall AERONET AOD climatology at each site) and AeroCom model output.*"

(2) Indicate where there are limitations in the AERONET retrieval climatologies at these two sites (e.g., winter)

(3) Demonstrate that while the level 1.5 values for AOD were at least in the ballpark there could be large discrepancies if just the level 2 almucantar retrievals were used to represent the seasonality AOD and AAOD, while the opposite is true for SSA. (Level 2.0 retrievals of SSA are more representative of the climatological data (in-situ and model) than level 1.5\* SSA.)

We've added the following sentences:

"While 50% discrepancy between the AERONET and in-situ climatology may appear significant, it's important to remember that these data sets do not represent the same period of time or measurement conditions (e.g., time of day, cloud cover, aerosol events, ambient humidity, etc). Schutgens et al. (2016) shows there can be large differences when comparing values obtained with different samplings (more than 100% for AOD), particularly when there are high levels of variability in the data."

Lines 1107 - 1113: It would be useful to note that the fine mode volume median radius (and also fine mode effective radius) decrease as AOD decreases at all of these AERONET sites, thereby suggesting one physical mechanism for the lower SSA at lower AOD levels. Sub-micron sized particles scatter less efficiently when they are smaller, therefore even if the particle composition remains constant the smaller sized particles at lower AOD would result in lower SSA. Additionally the coarse mode fraction of AOD increases at lower AOD likely due to a relatively constant background level of coarse mode particles while the fine mode often shows much larger seasonal increases due to stagnation, humidification, cumulus cloud interaction in the summer (opposite seasonality for Fresno due to fog in the winter). The lower Angstrom Exponents seen in low AOD months at most of these sites is indicative of greater coarse mode fraction of AOD when AOD is lower. Since the in-situ instruments under-sample the coarse mode particles this can potentially be a source of bias in the in-situ determination of SSA (440 nm) at lower concentration levels (lower AOD days), since iron oxide absorption in coarse mode dust may be missed. The SGP and Bondville sites are located in rural areas that would be expected to have varying amounts of mineral dust loading, depending on regional soil moisture conditions and transport from drier regions to the west (direction of prevailing winds in many seasons).

This depends on the size distribution of the sub-micron particles relative to the size corresponding to the first peak in the Mie scattering efficiency curve. The Reviewer is correct that particles smaller than the size of the Mie peak will scatter less efficiently when they are smaller, but particles that are larger than the size of the Mie peak will scatter more efficiently when they become smaller (until they shrink below the Mie peak). The Reviewer's assertion may or may not be correct, depending on the size distribution and magnitude of the size change. We have not changed the text. Additionally, as we've noted in the text and in response to other comments (e.g., our response to the reviewers comment about lines 806-808) related to dust absorption, we think it's very unlikely that dust is a dominant culprit in the discrepancy between in-situ and AERONET absorption and SSA at these two sites.

Lines 1180-1182: Suggest changing to "…typically well within the reported uncertainty bounds especially in light of the in-situ value of 0.06"

We've updated our uncertainty analysis and included a plot of the uncertainties in SSA for the in-situ measurements (new figure 3). Because of this we think that the statement is fine as stated.

Lines 1188-1191: It should also be mentioned that a fraction of the 440 nm absorption from coarse mode dust may be underestimated from in-situ measurements due to the large particle under-sampling bias.

See our thoughts about coarse mode dust effects in our response to the reviewers comments about Lines 806-808. We have not added a dust caveat.

[revised manuscript text omitted]